# THE ROLE OF COVERAGE IN ONLINE REINFORCEMENT LEARNING

**Tengyang Xie**[*]
UIUC
tx10@illinois.edu

**Dylan J. Foster**[*]
Microsoft Research
dylanfoster@microsoft.com

**Yu Bai**
Salesforce Research
yu.bai@salesforce.com

**Nan Jiang**
UIUC
nanjiang@illinois.edu

**Sham M. Kakade**
Harvard University
sham@seas.harvard.edu

## ABSTRACT

*Coverage conditions*—which assert that the data logging distribution adequately covers the state space—play a fundamental role in determining the sample complexity of offline reinforcement learning. While such conditions might seem irrelevant to online reinforcement learning at first glance, we establish a new connection by showing—somewhat surprisingly—that the mere *existence* of a data distribution with good coverage can enable sample-efficient online RL. Concretely, we show that *coverability*—that is, existence of a data distribution that satisfies a ubiquitous coverage condition called concentrability—can be viewed as a structural property of the underlying MDP, and can be exploited by standard algorithms for sample-efficient exploration, even when the agent does not know said distribution. We complement this result by proving that several weaker notions of coverage, despite being sufficient for offline RL, are insufficient for online RL. We also show that existing complexity measures for online RL, including Bellman rank and Bellman-Eluder dimension, fail to optimally capture coverability, and propose a new complexity measure, the *sequential extrapolation coefficient*, to provide a unification.

## 1 INTRODUCTION

The last decade has seen development of reinforcement learning algorithms with strong empirical performance in domains including robotics (Kober et al., 2013; Lillicrap et al., 2015), dialogue systems (Li et al., 2016), and personalization (Agarwal et al., 2016; Tewari and Murphy, 2017). While there is great interest in applying these techniques to real-world decision making applications, the number of samples (steps of interaction) required to do so is often prohibitive, with state-of-the-art algorithms requiring millions of samples to reach human-level performance in challenging domains. Developing algorithms with improved sample efficiency, which entails efficiently generalizing across high-dimensional states and actions while taking advantage of problem structure as modeled practitioners, remains a major challenge.

Investigation into design and analysis of algorithms for sample-efficient reinforcement learning has largely focused on two distinct problem formulations:

- *Online reinforcement learning*, where the learner can repeatedly interact with the environment by executing a policy and observing the resulting trajectory.

- *Offline reinforcement learning*, where the learner has access to logged transitions and rewards gathered from a fixed behavioral policy (e.g., historical data or expert demonstrations), but cannot directly interact with the underlying environment.

While these formulations share a common goal (learning a near-optimal policy), the algorithms used to achieve this goal and conditions under which it can be achieved are seemingly quite different. Focusing on value function approximation, sample-efficient algorithms for online reinforcement learning require both (a) *representation conditions*, which assert that the function approximator is flexible enough to represent value functions for the underlying MDP (optimal or otherwise), and (b) *exploration conditions*

---

[*]Equal contribution

(or, structural conditions) which limit the amount of exploration required to learn a near-optimal policy—typically by enabling extrapolation across states or limiting the number of effective state distributions (Russo and Van Roy, 2013; Jiang et al., 2017; Sun et al., 2019; Wang et al., 2020b; Du et al., 2021; Jin et al., 2021a; Foster et al., 2021). Algorithms for offline reinforcement learning typically require similar representation conditions. However, since data is collected passively from a fixed logging policy/distribution rather than actively, the exploration conditions used in online RL are replaced with *coverage conditions*, which assert that the data collection distribution provides sufficient coverage over the state space (Antos et al., 2008; Chen and Jiang, 2019; Xie and Jiang, 2020; 2021; Jin et al., 2021b; Rashidinejad et al., 2021; Foster et al., 2022; Zhan et al., 2022). The aim for both lines of research (online and offline) is to identify the weakest possible conditions under which learning is possible, and design algorithms that take advantage of these conditions. The two lines have largely evolved in parallel, and it is natural to wonder whether there are deeper connections. Since the conditions for sample-efficient online RL and offline RL mainly differ via exploration versus coverage, this leads us to ask:

> *If an MDP admits a data distribution with favorable coverage for offline RL, what does this imply about our ability to perform online RL efficiently?*

Beyond intrinsic theoretical value, this question is motivated by the observation that many real-world applications lie on a spectrum between online and offline. It is common for the learner to have access to logged/offline data, yet also have the ability to actively interact with the underlying environment, possibly subject to limitations such as an exploration budget (Kalashnikov et al., 2018). Building a theory of real-world RL that can lead to algorithm design insights for such settings requires understanding the interplay between online and offline RL.

## 1.1 OUR RESULTS

We investigate connections between coverage conditions in offline RL and exploration in online RL by focusing on the *concentrability* coefficient, the most ubiquitous notion of coverage in offline RL. Concentrability quantifies the extent to which the data collection distribution uniformly covers the state-action distribution induced by any policy. We introduce a new structural property, *coverability*, which reflects the best concentrability coefficient that can be achieved by *any data distribution*, possibly designed by an oracle with knowledge of the underlying MDP. Our main results are as follows:

1. We show (Section 3) that coverability (that is, mere existence of a distribution with good concentrability) is sufficient for sample-efficient online exploration, even when the learner has no prior knowledge of this distribution. This result requires no additional assumptions on the underlying MDP beyond standard Bellman completeness, and—perhaps surprisingly—is achieved using standard algorithms (Jin et al., 2021a), albeit with analysis ideas that go beyond existing techniques.

2. We show (Section 4) that several weaker notions of coverage in offline RL, including single-policy concentrability (Jin et al., 2021b; Rashidinejad et al., 2021) and conditions based on Bellman residuals (Chen and Jiang, 2019; Xie et al., 2021a), are *insufficient* for sample-efficient online exploration. This shows that in general, coverage in offline reinforcement learning and exploration in online RL not compatible, and highlights the need for additional investigation going forward.

Our results serve as a starting point for systematic study of connections between online and offline learnability in RL. To this end, we provide several secondary results:

1. We show (Section 5) that existing complexity measures for online RL, including Bellman rank and Bellman-Eluder dimension, do not optimally capture coverability, and provide a new complexity measure, the sequential extrapolation coefficient, which unifies these notions.

2. We establish (Appendix C) connections between coverability and reinforcement learning with exogenous noise, with applications to learning in exogenous block MDPs (Efroni et al., 2021; 2022a).

3. We give algorithms for reward-free exploration (Jin et al., 2020a; Chen et al., 2022) under coverability (Appendix G).

While our results primarily concern analysis of existing algorithms rather than algorithm design, they highlight a number of exciting directions for future research, and we are optimistic that the notion of coverability can guide the design of practical algorithms going forward.

**Notation.** For an integer $n \in \mathbb{N}$, we let $[n]$ denote the set $\{1,...,n\}$. For a set $\mathcal{X}$, we let $\Delta(\mathcal{X})$ denote the set of all probability distributions over $\mathcal{X}$. We adopt standard big-oh notation, and write $f = \widetilde{O}(g)$ to denote that $f = O(g \cdot \max\{1, \text{polylog}(g)\})$ and $a \lesssim b$ as shorthand for $a = O(b)$.

## 2 BACKGROUND: ONLINE/OFFLINE RL, COVERAGE, AND COVERABILITY

**Markov decision processes.** We consider an episodic reinforcement learning setting. Formally, a Markov decision process $M = (\mathcal{X}, \mathcal{A}, P, R, H, x_1)$ consists of a (potentially large) state space $\mathcal{X}$, action space $\mathcal{A}$, horizon $H$, probability transition function $P = \{P_h\}_{h=1}^{H}$, where $P_h : \mathcal{X} \times \mathcal{A} \to \Delta(\mathcal{X})$, reward function $R = \{R_h\}_{h=1}^{H}$, where $R_h : \mathcal{X} \times \mathcal{A} \to [0,1]$, and deterministic initial state $x_1 \in \mathcal{X}$.[1] A (randomized) policy is a sequence of per-timestep functions $\pi = \{\pi_h : \mathcal{X} \to \Delta(\mathcal{A})\}_{h=1}^{H}$. The policy induces a distribution over trajectories $(x_1, a_1, r_1), \ldots, (x_H, a_H, r_H)$ via the following process. For $h = 1, \ldots, H$: $a_h \sim \pi(\cdot \mid x_h)$, $r_h = R_h(x_h, a_h)$, and $x_{h+1} \sim P_h(\cdot \mid x_h, a_h)$. For notational convenience, we use $x_{H+1}$ to denote a deterministic terminal state with zero reward. We let $\mathbb{E}^\pi[\cdot]$ and $\mathbb{P}^\pi[\cdot]$ denote expectation and probability under this process, respectively.

The $Q$-function for policy $\pi$ is $Q_h^\pi(x,a) := \mathbb{E}^\pi\left[\sum_{h'=h}^{H} r_{h'} \mid x_h = x, a_h = a\right]$, the value function for $\pi$ is $V_h^\pi(x) := \mathbb{E}_{a \sim \pi_h(\cdot \mid x)}[Q_h^\pi(x,a)]$, and the expected reward for $\pi$ is $J(\pi) := V_1^\pi(x_1)$. We let $\pi^\star$ denote the optimal (deterministic) policy, which maximizes $Q_h^\pi(x,a)$ for all $(x,a) \in \mathcal{X} \times \mathcal{A}$ simultaneously; we define $V_h^\star = V_h^{\pi^\star}$ and $Q_h^\star = Q_h^{\pi^\star}$. We define the occupancy measure for policy $\pi$ via $d_h^\pi(x,a) := \mathbb{P}^\pi[x_h = x, a_h = a]$ and $d_h^\pi(x) := \mathbb{P}^\pi[x_h = x]$. We let $\mathcal{T}_h$ denote the Bellman operator for layer $h$, defined via $[\mathcal{T}_h f](x,a) = R_h(x,a) + \mathbb{E}_{x' \sim P_h(x,a)}[\max_{a'} f(x',a')]$ for $f : \mathcal{X} \times \mathcal{A} \to \mathbb{R}$. We also assume that rewards are normalized such that $\sum_{h \in [H]} r_h \in [0,1]$. To simplify technical presentation, we assume that $\mathcal{X}$ and $\mathcal{A}$ are countable; we anticipate that this assumption can be removed.

**Online Reinforcement Learning.** Our main results concern online reinforcement learning in an episodic framework, where the learner repeatedly interacts with an unknown MDP by executing a policy and observing the resulting trajectory, with the goal of maximizing total reward.

Formally, the protocol proceeds in $T$ rounds, where at each round $t = 1, \ldots, T$, the learner: i) Selects a policy $\pi^{(t)} = \left\{\pi_h^{(t)}\right\}_{h \in [H]}$ to execute in the (unknown) underlying MDP $M^\star$; ii) Observe the resulting trajectory $(x_1^{(t)}, a_1^{(t)}, r_1^{(t)}), \ldots, (x_H^{(t)}, a_H^{(t)}, r_H^{(t)})$. The learner's goal is to minimize their cumulative regret, defined via $\mathsf{Reg} := \sum_{t \in [T]} J(\pi^\star) - J(\pi^{(t)})$.

To achieve sample-efficient online reinforcement learning guarantees that do not depend on the size of the state space, one typically appeals to *value function approximation* methods that take advantage of a function class $\mathcal{F} \subset (\mathcal{X} \times \mathcal{A} \to \mathbb{R})$ that attempts to model the value functions for the underlying MDP $M^\star$ (optimal or otherwise). An active line of research provides structural conditions under which such approaches succeed (Russo and Van Roy, 2013; Jiang et al., 2017; Sun et al., 2019; Wang et al., 2020b; Du et al., 2021; Jin et al., 2021a; Foster et al., 2021), based on assumptions that control the interplay between the function approximator $\mathcal{F}$ and the dynamics of the MDP $M^\star$. These results require (i) *representation conditions*, which require that $\mathcal{F}$ is flexible enough to model value functions of interest (e.g., $Q^\star \in \mathcal{F}$ or $\mathcal{T}_h \mathcal{F}_{h+1} \subseteq \mathcal{F}_h$) and (ii) *exploration conditions*, which either explicitly or implicitly limit the amount of exploration required for a deliberate algorithm to learn a near-optimal policy. This is typically accomplished by either enabling extrapolation from states already visited, or by limiting the number of effective state distributions that can be encountered.

**Offline Reinforcement Learning and Coverage Conditions.** Our aim is to investigate parallels between online and offline reinforcement learning. In offline reinforcement learning, the learner cannot actively execute policies in the underlying MDP $M^\star$. Instead, for each layer $h$, they receive a dataset $D_h$ of $n$ tuples $(x_h, a_h, r_h, x_{h+1})$ with $r_h = R_h(x_h, a_h)$, $x_{h+1} \sim P_h(\cdot \mid x_h, a_h)$, and $(x_h, a_h) \sim \mu_h$ i.i.d., where $\mu_h \in \Delta(\mathcal{X} \times \mathcal{A})$ is the *data collection distribution*; we define $\mu = \{\mu_h\}_{h \in [H]}$. The goal of the learner is to use this data to learn an $\varepsilon$-optimal policy $\widehat{\pi}$, that is: $J(\pi^\star) - J(\widehat{\pi}) \leq \varepsilon$.

Algorithms for offline reinforcement learning require representation conditions similar to those required for online RL. However, since it is not possible to actively explore the underlying MDP, one dispenses with exploration conditions and instead considers *coverage conditions*, which require that each data distribution $\mu_h$ sufficiently covers the state space. As an example, consider *Fitted Q-Iteration* (FQI), one of the most well-studied offline reinforcement learning algorithms (Munos, 2007; Munos and Szepesvári, 2008; Chen and Jiang, 2019). The algorithm, which uses least-squares to approximate Bellman backups, is known to succeed under (i) a representation condition known as *Bellman completeness* (or "completeness"), which requires that $\mathcal{T}_h f \in \mathcal{F}_h$ for all $f \in \mathcal{F}_{h+1}$, and (ii) a coverage condition called *concentrability*. To state, the result, recall that $\|x\|_\infty := \max_i |x_i|$ for $x \in \mathbb{R}^d$.

---

[1]While our results assume that the initial state is fixed for simplicity, this assumption is straightforward to relax.

**Definition 1** (Concentrability). *The concentrability coefficient for a data distribution $\mu = \{\mu_h\}_{h=1}^H$ and policy class $\Pi$ is given by $C_{\mathsf{conc}}(\mu) := \sup_{\pi \in \Pi, h \in [H]} \|d_h^\pi / \mu_h\|_\infty$.*

Concentrability requires that the data distribution uniformly covers all possible induced state distributions. With concentrability[2] and completeness, FQI can learn an $\varepsilon$-optimal policy using $\mathrm{poly}(C_{\mathsf{conc}}(\mu), \log|\mathcal{F}|, H, \varepsilon^{-1})$ samples. Importantly, this result scales only with the concentrability coefficient $C_{\mathsf{conc}}(\mu)$ and the capacity $\log|\mathcal{F}|$ for the function class, and has no explicit dependence on the size of the state space. There is a vast literature which provides algorithms with similar, often more refined guarantees (Chen and Jiang, 2019; Xie and Jiang, 2020; 2021; Jin et al., 2021b; Rashidinejad et al., 2021; Foster et al., 2022; Zhan et al., 2022).

**The Coverability Coefficient.** Having seen that access to a data distribution $\mu$ with low concentrability $C_{\mathsf{conc}}(\mu)$ is sufficient for sample-efficient offline RL, we now ask what existence of such a distribution implies about our ability to perform online RL. To this end, we introduce a new structural parameter, the *coverability coefficient*, whose value reflects the best concentrability coefficient that can be achieved with oracle knowledge of the underlying MDP $M^\star$.

**Definition 2** (Coverability). *The coverability coefficient $C_{\mathsf{cov}} > 0$ for a policy class $\Pi$ is given by $C_{\mathsf{cov}} := \inf_{\mu_1, \ldots, \mu_H \in \Delta(\mathcal{X} \times \mathcal{A})} \{C_{\mathsf{conc}}(\mu)\}$.*

Coverability is an intrinsic structural property of the MDP $M^\star$ which implicitly restricts the complexity of the set of possible state distributions. While it is always the case that $C_{\mathsf{cov}} \leq |\mathcal{X}| \cdot |\mathcal{A}|$, the coefficient can be significantly smaller (in particular, independent of $|\mathcal{X}|$) for benign MDPs such as block MDPs and MDPs with low-rank structure (Chen and Jiang, 2019, Prop 5); see Appendix C for details.

With this definition in mind, we ask: *If the MDP $M^\star$ satisfies low coverability, is sample-efficient online reinforcement learning possible?* Note that if the learner were given access to data from the distribution $\mu$ that achieves the value of $C_{\mathsf{cov}}$, it would be possible to simply appeal to offline RL methods such as FQI, but since the learner has no prior knowledge of $\mu$, this question is non-trivial, and requires deliberate exploration.

## 3 COVERABILITY IMPLIES SAMPLE-EFFICIENT ONLINE EXPLORATION

We now present our main result, which shows that low coverability is sufficient for sample-efficient online exploration. We first describe the algorithm and regret bound, then sketch the proof and give intuition (Section 3.1). We conclude (Section 3.2) by applying the main result to give regret bounds for learning in *exogenous block MDPs* (Efroni et al., 2021), highlighting structural properties of coverability.

**Function approximation.** We work with a value function class $\mathcal{F} = \mathcal{F}_1 \times \cdots \times \mathcal{F}_H$, where $\mathcal{F}_h \subset (\mathcal{X} \times \mathcal{A} \to [0,1])$, with the goal of modeling value functions for the underlying MDP. We adopt the convention that $f_{H+1} = 0$, and for each $f \in \mathcal{F}$, we let $\pi_f$ denote the greedy policy with $\pi_{f,h}(x) := \mathrm{argmax}_{a \in \mathcal{A}} f_h(x,a)$, and we use $f_h(x, \pi_h) := \mathbb{E}_{a \sim \pi_h(\cdot|x)}[f_h(x,a)]$ for any $\pi_h$. We take our policy class to be the induced class $\Pi := \{\pi_f \mid f \in \mathcal{F}\}$ for the remainder of the paper unless otherwise stated. We make the following standard completeness assumption, which requires that the value function class is closed under Bellman backups (Wang et al., 2020b; Jin et al., 2020b; Wang et al., 2021b; Jin et al., 2021a).

**Assumption 1** (Completeness). *For all $h \in [H]$, we have $\mathcal{T}_h f_{h+1} \in \mathcal{F}_h$ for all $f_{h+1} \in \mathcal{F}_{h+1}$.*

Completeness implies that $\mathcal{F}$ is *realizable* (that is, $Q^\star \in \mathcal{F}$), but is a stronger assumption in general. We assume for simplicity that $|\mathcal{F}| < \infty$, and our results scale with $\log|\mathcal{F}|$; this can be extended to infinite classes via covering numbers using a standard analysis.

**Algorithm and main result.** Our result is based on a new analysis of the GOLF algorithm of Jin et al. (2021a), which is presented in Algorithm 1 of Appendix D for completeness. GOLF is based on the principle of optimism in the face of uncertainty. At each round, the algorithm restricts to a confidence set $\mathcal{F}^{(t)} \subseteq \mathcal{F}$ with the property that $Q^\star \in \mathcal{F}^{(t)}$, and chooses $\pi^{(t)} = \pi_{f^{(t)}}$ based on the value function $f^{(t)} \in \mathcal{F}^{(t)}$ with the most optimistic estimate $f_1(x_1, \pi_{f,1}(x_1))$ for the total reward. The confidence sets $\mathcal{F}^{(t)}$ are based on an empirical proxy to squared Bellman error, and are constructed in a *global* fashion that entails optimizing over $f_h$ for all layers $h \in [H]$ simultaneously (Zanette et al., 2020a).

---

[2]Specifically, FQI requires concentrability with $\Pi$ chosen to be the set of all admissible policies (see, e.g., Chen and Jiang, 2019). Other algorithms (Xie and Jiang, 2020) can leverage concentrability w.r.t smaller policy classes.

Note that while GOLF was originally introduced to provide regret bounds based on the notion of Bellman-Eluder dimension, we show (Section 5) that coverability cannot be (optimally) captured by this complexity measure, necessitating a new analysis. Our main result, Theorem 1, shows that GOLF attains low regret for online reinforcement learning whenever the coverability coefficient is small.

**Theorem 1** (Coverability implies sample-efficient online RL). *Under Assumption 1, there exists an absolute constant $c$ such that for any $\delta \in (0,1]$ and $T \in \mathbb{N}_+$, if we choose $\beta = c \cdot \log(TH|\mathcal{F}|/\delta)$ in Algorithm 1, then with probability at least $1-\delta$, we have $\mathsf{Reg} \leq O(H\sqrt{C_{\mathsf{cov}}T\log(TH|\mathcal{F}|/\delta)}\log(T))$, where $C_{\mathsf{cov}}$ is the coverability coefficient (Definition 2).*

Beyond the coverability parameter $C_{\mathsf{cov}}$, the regret bound in Theorem 1 depends only on standard problem parameters (the horizon $H$ and function class capacity $\log|\mathcal{F}|$). Hence, this result shows that coverability, along with completeness, is sufficient for sample-efficient online RL. Additional features of Theorem 1 are as follows.

- While coverability implies that there exists a distribution $\mu$ for which the concentrability coefficient $C_{\mathsf{conc}}$ is bounded, Algorithm 1 has no prior knowledge of this distribution. We find the fact that the GOLF algorithm—which does not involve explicitly searching such a distribution—succeeds under this condition to be somewhat surprising (recall that given sample access to $\mu$, one can simply run FQI). Our proof shows that despite the fact that GOLF does not explicitly reason about $\mu$, coverability implicitly restricts the set of possible state distributions, and limits the extent to which the algorithm can be "surprised" by substantially new distributions. We anticipate that this analysis will find broader use.

- Ignoring factors logarithmic in $T$, $H$, and $\delta^{-1}$, the regret bound in Theorem 1 scales as $H\sqrt{C_{\mathsf{cov}}T\log|\mathcal{F}|}$, which is optimal for contextual bandits (where $C_{\mathsf{cov}} = |\mathcal{A}|$ and $H = 2$),[3] and hence cannot be improved in general (Agarwal et al., 2012). The dependence on $H$ matches the regret bound for GOLF based on Bellman-Eluder dimension (Jin et al., 2021a).

- GOLF uses confidence sets based on squared Bellman error, but there are similar algorithms which instead work with average Bellman error (Jiang et al., 2017; Du et al., 2021) and, as a result, require only realizability rather than completeness (Assumption 1). While existing complexity measures such as Bellman rank and Bellman-Eluder dimension can be used to analyze both types of algorithm, and our results critically use the non-negativity of squared Bellman error, which facilitates certain "change-of-measure" arguments. Consequently, it is unclear whether the completeness assumption can be removed (i.e., whether coverability and realizability alone suffice for sample-efficient online RL).

On the algorithmic side, our results give guarantees for PAC RL via online-to-batch conversion, which we state here for completeness. We also provide an extension to reward-free exploration in Appendix G.

**Corollary 2.** *Under Assumption 1, there exists an absolute constant $c$ such that for any $\delta \in (0,1]$ and $T \in \mathbb{N}_+$, if we choose $\beta = c \cdot \log(TH|\mathcal{F}|/\delta)$ in Algorithm 1, then with probability at least $1-\delta$, the policy $\bar{\pi}$ output by Algorithm 1 has[4] $J(\pi^\star) - J(\bar{\pi}) \leq O\big(H\sqrt{C_{\mathsf{cov}}\log(TH|\mathcal{F}|/\delta)\log(T)/T}\big)$.*

### 3.1 PROOF SKETCH FOR THEOREM 1: WHY IS COVERABILITY SUFFICIENT?

We now sketch the main ideas behind the proof of Theorem 1, highlighting the role of coverability in limiting the complexity of exploration.

**Regret decomposition and change of measure.** For each $t$, we define $\delta_h^{(t)}(\cdot,\cdot) := f_h^{(t)}(\cdot,\cdot) - (\mathcal{T}_h f_{h+1}^{(t)})(\cdot,\cdot)$, which may be viewed as a "test function" at level $h$ induced by $f^{(t)} \in \mathcal{F}$. We adopt the shorthand $d_h^{(t)} \equiv d_h^{\pi^{(t)}}$, and we define $\widetilde{d}_h^{(t)}(x,a) := \sum_{i=1}^{t-1} d_h^{(i)}(x,a)$ as the cumulative historical visitation for rounds prior to step $t$.

A standard regret decomposition for optimistic algorithms (Lemma 13) allows us to relate regret to the average Bellman error under the learner's sequence of policies:

$$\mathsf{Reg} \leq \sum_{t\in[T]} \left( f_1^{(t)}(x_1, \pi_{f_1^{(t)},1}(x_1)) - J(\pi^{(t)}) \right) = \sum_{t\in[T]}\sum_{h\in[H]} \mathbb{E}_{d_h^{(t)}} \big[ \underbrace{f_h^{(t)}(x,a) - (\mathcal{T}_h f_{h+1}^{(t)})(x,a)}_{=:\delta_h^{(t)}(x,a)} \big]. \quad (1)$$

---

[3] We require $H = 2$ to apply the result to contextual bandits due to assuming the deterministic starting state.

[4] $\bar{\pi}$ is the non-Markov policy obtained by sampling $t \sim [T]$ and playing $\pi^{(t)}$.

Fix $h \in [H]$. We use a change-of-measure argument to relate the on-policy *average* Bellman error $\mathbb{E}_{(x,a) \sim d_h^{(t)}}[\delta_h^{(t)}(x,a)]$ to the in-sample *squared* Bellman error under $\widetilde{d}_h^{(t)}$, writing Eq. (1) as

$$\sum_{t \in [T]} \sum_{x,a} d_h^{(t)}(x,a) \left( \frac{\widetilde{d}_h^{(t)}(x,a)}{\widetilde{d}_h^{(t)}(x,a)} \right)^{1/2} \delta_h^{(t)}(x,a) \leq \underbrace{\sqrt{\sum_{t \in [T]} \sum_{x,a} \frac{\left(d_h^{(t)}(x,a)\right)^2}{\widetilde{d}_h^{(t)}(x,a)}}}_{\text{(I): extrapolation error}} \cdot \underbrace{\sqrt{\sum_{t \in [T]} \sum_{x,a} \widetilde{d}_h^{(t)}(x,a) \left(\delta_h^{(t)}(x,a)\right)^2}}_{\text{(II): in-sample } \textit{squared} \text{ Bellman error}},$$

where the inequality is an application of Cauchy-Schwarz. As an immediate consequence of the confidence set construction in Eq. (3), completeness, and a standard concentration argument (Lemma 12 in Appendix D), we can bound the in-sample error by $\text{(II)} \leq O\left(\sqrt{\beta T}\right)$.

**Bounding the extrapolation error using coverability.** To proceed, we show that the extrapolation error (I) is controlled by coverability. We have:

$$\sum_{t \in [T]} \sum_{x,a} \frac{\left(d_h^{(t)}(x,a)\right)^2}{\widetilde{d}_h^{(t)}(x,a)} \leq \sum_{t \in [T]} \sum_{x,a} \max_{t' \in [T]} d_h^{(t')}(x,a) \cdot \frac{d_h^{(t)}(x,a)}{\widetilde{d}_h^{(t)}(x,a)} \leq \underbrace{\left( \max_{x,a} \sum_{t \in [T]} \frac{d_h^{(t)}(x,a)}{\widetilde{d}_h^{(t)}(x,a)} \right)}_{\overset{\text{(a)}}{\lesssim} O(\log(T)) \text{ by Lemma 15}} \cdot \underbrace{\left( \sum_{x,a} \max_{t \in [T]} d_h^{(t)}(x,a) \right)}_{\overset{\text{(b)}}{\leq} C_{\text{cov}} \text{ by Lemma 14}}.$$

Here, the inequality (a) uses a *scalar* variant of the elliptic potential lemma (Lemma 15; cf. Lattimore and Szepesvári (2020)), which we apply on a *per-state basis*.[5] The inequality (b) uses a key result (Lemma 14 in Appendix D), which shows that coverability is equivalent to a quantity we term *cumulative reachability*, defined via $\sum_{(x,a) \in \mathcal{X} \times \mathcal{A}} \sup_{\pi \in \Pi} d_h^\pi(x, a)$. Cumulative reachability reflects the variation in visitation probabilities for policies in the class $\Pi$, and boundedness of this quantity (which occurs when state-action pairs visited by policies in $\Pi$ have large overlap) implies that the contributions from potentials for different state-action pairs average out. See Figure 1 for an illustration.

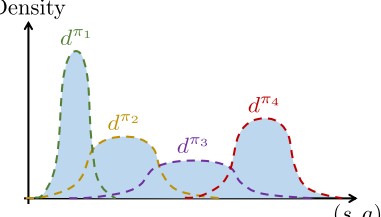

Figure 1: An example of *coverability* $\iff$ *cumulative reachability* (which is equal to the total area of the shaded region without double-counting overlaps. $\Pi = \{\pi_1, \pi_2, \pi_3, \pi_4\}$, dashed curves is $d^\pi$).

To conclude, we substitute the preceding bounds into the term (I), which gives $\text{Reg} \leq \sum_{h=1}^H \mathbb{E}_{(x,a) \sim d_h^{(t)}}[\delta_h^{(t)}(x,a)] \leq O\left(H\sqrt{C_{\text{cov}} \cdot \beta T \log(T)}\right)$.

Note that to obtain the expression in term (I), our proof critically uses that the confidence set construction provides a bound on the *squared Bellman error* $\mathbb{E}_{(x,a) \sim \widetilde{d}_h^{(t)}}[\delta_h^{(t)}(x,a)^2]$ in the change of measure argument. This contrasts with existing works on online RL with general function approximation (e.g., Jiang et al., 2017; Jin et al., 2021a; Du et al., 2021), which typically move from average Bellman error to squared Bellman error as a lossy step, and only work with squared Bellman error because it permits simpler construction of confidence sets. Confidence sets based on average Bellman error will lead to a larger notion of extrapolation error which cannot be controlled using coverability (cf. Section 5).

### 3.2 RICH OBSERVATIONS AND EXOGENOUS NOISE: APPLICATION TO BLOCK MDPs

As an application of Theorem 1, we consider the problem of reinforcement learning in *Exogenous Block MDPs* (Ex-BMDPs), a problem which has received extensive recent interest (Efroni et al., 2021; 2022a;b; Lamb et al., 2022). Recall that the block MDP (Jiang et al., 2017; Du et al., 2019; Misra et al., 2020) is a model in which the ("observed") state space $\mathcal{X}$ is large/high-dimensional, but can be mapped by an (unknown) decoder $\phi^\star$ to a small *latent* state space which governs the dynamics. Exogenous block MDPs generalize this model further by factorizing the latent state space into small controllable ("endogenous") component $\mathcal{S}$ and a large irrelevant ("exogenous") component $\Xi$, which may be temporally correlated.

The main challenge of learning in block MDPs is that the decoder $\phi^\star$ is not known to the learner in advance. Indeed, given access to the decoder, one can obtain regret $\text{poly}(H, |\mathcal{S}|, |\mathcal{A}|) \cdot \sqrt{T}$ by applying

---

[5]Applying this result formally requires a separate argument to handle early rounds in which pairs $(x,a)$ have been visited very little; this is given in Appendix D.

tabular reinforcement learning algorithms to the latent state space. In light of this, the aim of the Ex-BMDP setting is to obtain sample complexity guarantees that are independent of the size of the observed state space $|\mathcal{X}|$ and exogenous state space $|\Xi|$, and scale as $\text{poly}(|\mathcal{S}|,|\mathcal{A}|,H,\log|\mathcal{F}|)$, where $\mathcal{F}$ is an appropriate class of function approximators (typically either a value function class $\mathcal{F}$ or a class of decoders $\Phi$ that attempts to model $\phi^\star$ directly).

We show (Proposition 8 in Appendix C) that for any Ex-BMDP, one has $C_{\text{cov}} \leq |\mathcal{S}| \cdot |\mathcal{A}|$, which—through Theorem 1—implies that GOLF attains $\text{Reg} \leq O\big(H\sqrt{|\mathcal{S}||\mathcal{A}|T\log(TH|\mathcal{F}|/\delta)}\log(T)\big)$ whenever Assumption 1 holds; critically, this result scales only with the cardinality $|\mathcal{S}|$ for the endogenous latent state space, and with the capacity $\log|\mathcal{F}|$ for the value function class. It is the first result for this setting that allows for stochastic latent dynamics and emission process, albeit with the extra assumption of completeness. Existing algorithms either require that the endogenous latent dynamics $P^{\text{endo}}$ are deterministic (Efroni et al., 2021) or allow for stochastic dynamics but heavily restrict the observation process (Efroni et al., 2022a), and existing complexity measures such as Bellman Rank and Bellman-Eluder dimension can be arbitrarily large for this setting (see discussion in Section 5). See Appendix C for details and discussion.

## 4    ARE WEAKER NOTIONS OF COVERAGE SUFFICIENT?

In Section 3, we showed that existence of a distribution with good concentrability (coverability) is sufficient for sample-efficient online RL. However, while concentrability is the most ubiquitous coverage condition in offline RL, there are several weaker notions of coverage which also lead to sample-efficient offline RL algorithms. In this section, we show that analogues of coverability based on these conditions, *single-policy concentrability* and *generalized concentrability* for Bellman residuals, do not suffice for sample-efficient online RL. This indicates that in general, the interplay between offline coverage and online exploration is nuanced.

**Single-policy concentrability.**    *Single-policy concentrability* is a widely used coverage assumption in offline RL which weakens concentrability by requiring only that the state distribution induced by $\pi^\star$ is covered by the offline data distribution $\mu$, as opposed to requiring coverage for all policies (Jin et al., 2021b; Rashidinejad et al., 2021).

**Definition 3** (Single-policy concentrability). *The single-policy concentrability coefficient for a data distribution* $\mu = \{\mu_h\}_{h=1}^{H}$ *is given by* $C_{\text{conc}}^\star(\mu) := \big\| d_h^{\pi^\star}/\mu_h \big\|_\infty$.

For offline RL, algorithms based on pessimism provide sample guarantee complexity guarantees that scale with $C_{\text{conc}}^\star(\mu)$ (Jin et al., 2021b; Rashidinejad et al., 2021). However, for the online setting, it is trivial to show that an analogous notion of "single-policy coverability" (i.e., existence of a distribution with good single-policy coverability) is not sufficient for sample-efficient learning, since for any MDP, one can take $\mu = d^{\pi^\star}$ to attain $C_{\text{conc}}^\star(\mu) = 1$. This suggests that any notion of coverage that suffices for online RL must be more uniform in nature.

**Generalized concentrability for Bellman residuals.**    Another approach to weaker coverage in offline RL is to relax concentrability by only requiring coverage with respect to the Bellman residuals for value functions in $\mathcal{F}$ (Chen and Jiang, 2019; Xie et al., 2021a; Cheng et al., 2022); the following definition adapts this notion to the finite-horizon setting.

**Definition 4** (Generalized concentrability). *We define the generalized concentrability coefficient* $\mathfrak{C}_{\text{conc}}(\mu,\mathcal{F})$ *for a policy class $\Pi$ and value function class $\mathcal{F}$ as the least constant $C > 0$ such that the offline data distribution $\mu = \{\mu_h\}_{h=1}^{H}$ satisfies that for all $f \in \mathcal{F}$ and $\pi \in \Pi$, $\sum_{h \in [H]} \mathbb{E}_{d_h^\pi}\big[(f_h(s_h,a_h) - (\mathcal{T}_h f_{h+1})(s_h,a_h))^2\big] \leq C \cdot \sum_{h \in [H]} \mathbb{E}_{\mu_h}\big[(f_h(s_h,a_h) - (\mathcal{T}_h f_{h+1})(s_h,a_h))^2\big]$.*

Note that $\mathfrak{C}_{\text{conc}}(\mu,\mathcal{F}) \leq C_{\text{conc}}(\mu)$ (in particular, they coincide if one chooses $\mathcal{F}$ to be the set of all functions over $\mathcal{X} \times \mathcal{A}$) but in general $\mathfrak{C}_{\text{conc}}(\mu,\mathcal{F})$ can be much smaller. For example, in the linear Bellman-complete setting, it is possible to bound $\mathfrak{C}_{\text{conc}}(\mu,\mathcal{F})$ in terms of feature coverage conditions (Wang et al., 2021a; Zanette et al., 2021). Using offline data from $\mu$, sample complexity guarantees that scale with $\mathfrak{C}_{\text{conc}}(\mu,\mathcal{F})$ can be obtained under Assumption 1 via MSBO (see, e.g., Xie and Jiang, 2020, Section 5) or by running a "one-step" variant of GOLF (Algorithm 1); we provide this result (Proposition 16) in Appendix E for completeness. Given that this notion leads to positive results for offline RL, it is natural to consider a generalized notion of coverability based upon it.

**Definition 5** (Generalized coverability). *We define the generalized coverability coefficient for a policy class $\Pi$ value function class $\mathcal{F}$ and as $\mathfrak{C}_{\mathsf{cov}}(\mathcal{F}) = \inf_{\mu_1,...,\mu_H \in \Delta(\mathcal{X} \times \mathcal{A})} \{\mathfrak{C}_{\mathsf{conc}}(\mu, \mathcal{F})\}$.*

Unfortunately, we show that this condition does not suffice for sample-efficient online RL, even when the number of actions is constant and Assumption 1 is satisfied.

**Theorem 3.** *For any $X, H, C \in \mathbb{N}$, there exists a family of MDPs with $|\mathcal{X}| = X$, $|\mathcal{A}| = 2$ and horizon $H$ and a function class $\mathcal{F}$ with $\log|\mathcal{F}| \leq H \log(2|\mathcal{X}|)$ such that: i) Assumption 1 (completeness) is satisfied for $\mathcal{F}$ and we have $\mathfrak{C}_{\mathsf{cov}}(\mathcal{F}) \leq C$ and ii) Any online RL algorithm that returns a $0.1$-optimal policy with probability $0.9$ requires at least $\Omega\left(\min\{X, 2^{\Omega(H)}, 2^{\Omega(C)}\}\right)$ trajectories.*

Theorem 3 highlights that in general, notions of coverage that suffice for offline RL—even those that are uniform in nature—can fail to lead to useful structural conditions for online RL. Briefly, the issue is that bounding regret for online RL entails controlling the extent to which a deliberate algorithm that has observed state distributions $d_h^{(1)},...,d_h^{(t-1)}$ can be "surprised" by a substantially new state distribution $d_h^{(t)}$; here, surprise is typically measure in terms of Bellman residual. The proof of Theorem 3 shows that existence of a distribution with good coverage with respect to Bellman residuals does suffice to provide meaningful control of distribution shift. We caution, however, that the lower bound construction makes use of the fact that Definition 5 requires coverage only *on average* across layers, and it is unclear whether a similar lower bound holds under uniform coverage across layers. Developing a more unified and fine-grained understanding of what coverage conditions lead to efficient exploration is an important question for future research.

## 5 A NEW STRUCTURAL CONDITION FOR SAMPLE-EFFICIENT ONLINE RL

Having shown that coverability facilitates sample-efficient online RL, an immediate question is whether this structural condition is related to existing complexity measures such as Bellman-Eluder dimension (Jin et al., 2021a) and Bellman/Bilinear rank (Jiang et al., 2017; Du et al., 2021), which attempt to unify existing approaches to sample-efficient RL. We now show that these complexity measures are insufficient to capture coverability, then provide a new complexity measure, the Sequential Extrapolation Coefficient, which bridges the gap.

### 5.1 INSUFFICIENCY OF EXISTING COMPLEXITY MEASURES

Bellman-Eluder dimension (Jin et al., 2021a) and Bellman/Bilinear rank (Jiang et al., 2017; Du et al., 2021) can fail to capture coverability for two reasons: (i) insufficiency of average Bellman error (as opposed to squared Bellman error), and (ii) incorrect dependence on scale. To highlight these issues, we focus on $Q$-type Bellman-Eluder dimension (Jin et al., 2021a), which subsumes Bellman rank.[6] See Appendix F for discussion of other complexity measures. Let $\mathfrak{D}_h^\Pi := \{d_h^\pi : \pi \in \Pi\}$ and $\mathcal{F}_h - \mathcal{T}_h \mathcal{F}_{h+1} := \{f_h - \mathcal{T}_h f_{h+1} : f \in \mathcal{F}\}$. Following Jin et al. (2021a), we define the ($Q$-type) Bellman-Eluder dimension as follows.

**Definition 6** (Bellman-Eluder dimension). *The Bellman-Eluder dimension $\mathsf{dim}_{\mathsf{BE}}(\mathcal{F}, \Pi, \varepsilon, h)$ for the layer $h$ is the largest $d \in \mathbb{N}$, such that there exist sequences $\{d_h^{(1)}, d_h^{(2)}, ..., d_h^{(d)}\} \subseteq \mathfrak{D}_h^\Pi$ and $\{\delta_h^{(1)}, ..., \delta_h^{(d)}\} \subseteq \mathcal{F}_h - \mathcal{T}_h \mathcal{F}_{h+1}$ such that for all $t \in [d]$, $|\mathbb{E}_{d_h^{(t)}}[\delta_h^{(t)}]| > \varepsilon^{(t)}$, and $\sqrt{\sum_{i=1}^{t-1} \left(\mathbb{E}_{d_h^{(i)}}[\delta_h^{(t)}]\right)^2} \leq \varepsilon^{(t)}$, for $\varepsilon^{(1)}, ..., \varepsilon^{(d)} \geq \varepsilon$. We define $\mathsf{dim}_{\mathsf{BE}}(\mathcal{F}, \Pi, \varepsilon) = \max_{h \in [H]} \mathsf{dim}_{\mathsf{BE}}(\mathcal{F}, \Pi, \varepsilon, h)$.*

**Issue #1: Insufficiency of average (vs. squared) Bellman error.** The Bellman-Eluder dimension reflects the length of the longest consecutive sequence of value function pairs for which we can be "surprised" by a large Bellman residual for a new policy if the value function has low Bellman residual on all preceding policies. Note that via Definition 6, the Bellman-Eluder dimension measures the size of the surprise and the error on preceding points via *average* Bellman error (e.g., $\mathbb{E}_{d_h^{(i)}}[\delta_h^{(t)}]$). On the other hand, the proof of Theorem 1 critically uses *squared* Bellman error $\mathbb{E}_{d_h^{(i)}}[(\delta_h^{(t)})^2]$ bound regret by coverability; this is because the (point-wise) nonnegativity of squared Bellman error facilitates change-of-measure in a similar fashion to offline reinforcement learning. The following result shows that this issue is fundamental, and Bellman-Eluder dimension can be exponential large relative to the regret bound in Theorem 1.

**Proposition 4.** *For any $d \in \mathbb{N}$, there exists an MDP $M$ with $H = 2$ and $|\mathcal{A}| = 2$, policy class $\Pi$ with $|\Pi| = d$, and value function class $\mathcal{F}$ with $|\mathcal{F}| = d$ satisfying completeness, such that $C_{\mathsf{cov}} = O(1)$, but the Bellman-Eluder dimension has $\mathsf{dim}_{\mathsf{BE}}(\mathcal{F}, \Pi, \varepsilon) = \Omega(\min\{|\mathcal{F}|, |\Pi|\}) = \Omega(d)$ for any $\varepsilon \leq 1/2$.*

---

[6]$Q$-type and $V$-type are similar, but define the Bellman residual with respect to different action distributions.

The lower bound in Proposition 4 is realized by an exogenous block MDP (Appendix C), with $d$ representing the number of *exogenous* states. The result gives an exponential separation between what can be achieved using Bellman-Eluder dimension and coverability, because GOLF attains $\mathsf{Reg} \leq \widetilde{O}\big(\sqrt{T\log(d)}\big)$ (cf. Corollary 9), yet we have $\dim_{\mathsf{BE}}(\mathcal{F},\Pi,1/2)=\Omega(d)$. The construction, which is based on Efroni et al. (2022b, Section B.1), critically leverages cancellations in the average Bellman error; these cancellations are ruled out by squared Bellman error, which is why Theorem 1 gives a regret bound that scales only *logarithmically* in $d$. Bilinear rank (Du et al., 2021) and $V$-type Bellman rank suffer from similar drawbacks; see Appendix F for further discussion.

**Issue #2: Incorrect dependence on scale.** In light of the previous example, a seemingly reasonable fix is to adapt the Bellman-Eluder dimension to consider squared Bellman error rather than average Bellman error (i.e., use $\sqrt{\sum_{i=1}^{t-1}\big(\mathbb{E}_{d_h^{(i)}}[(\delta_h^{(t)})^2]\big)} \leq \varepsilon^{(t)}$ in Definition 6). We show (Appendix F.1) that while it is possible to bound this modified Bellman-Eluder dimension in terms of the coverability parameter, the dependence on the scale parameter $\varepsilon$ is *polynomial*, and it is not possible to derive regret bounds better than $T^{2/3}$ under coverability with this approach. Informally, the issue is *scale*: Bellman-eluder dimension only checks whether the average Bellman error violates the threshold $\varepsilon$, and does not consider how far the error violates the threshold (e.g., $|\mathbb{E}_{d_h^{(t)}}[\delta_h^{(t)}]| > \varepsilon$ and $|\mathbb{E}_{d_h^{(t)}}[\delta_h^{(t)}]| > 1$ are counted the same).

## 5.2 THE SEQUENTIAL EXTRAPOLATION COEFFICIENT

To address the issues above, we introduce a new complexity measure, the Sequential Extrapolation Coefficient (SEC), which i) leads to regret bounds via GOLF and ii) subsumes both coverability and the Bellman-Eluder dimension. Conceptually, the Sequential Extrapolation Coefficient should be thought of as a minimal abstraction of the main ingredient in regret bounds based on GOLF and other optimistic algorithms: extrapolation from in-sample error to on-policy error. We begin by stating a variant of the Sequential Extrapolation Coefficient for abstract function classes, then specialize it to RL.

**Definition 7** (Sequential Extrapolation Coefficient). *Let $\mathcal{Z}$ be an abstract set. Given a test function class $\Psi \subset (\mathcal{Z} \to \mathbb{R})$ and distribution class $\mathfrak{D} \subset \Delta(\mathcal{Z})$, the sequential extrapolation coefficient for length $T$ is given by*

$$\mathsf{SEC}(\Psi,\mathfrak{D},T) := \sup_{\psi^{(1)},\dots,\psi^{(T)} \in \Psi} \sup_{d^{(1)},\dots,d^{(T)} \in \mathfrak{D}} \left\{ \sum_{t \in [T]} \frac{\mathbb{E}_{d^{(t)}}[\psi^{(t)}]^2}{1 \vee \sum_{i=1}^{t-1} \mathbb{E}_{d^{(i)}}[(\psi^{(t)})^2]} \right\}.$$

To apply the Sequential Extrapolation Coefficient to RL, we use Bellman residuals for $\mathcal{F}$ as test functions and consider state-action distributions induced by policies in $\Pi$.

**Definition 8** (SEC for RL). *We define $\mathsf{SEC}_{\mathsf{RL}}(\mathcal{F},\Pi,T) := \max_{h \in [H]} \mathsf{SEC}(\mathcal{F}_h - \mathcal{T}_h \mathcal{F}_{h+1}, \mathfrak{D}_h^\Pi, T)$.*

The following result, which is a near-immediate consequence of the definition, shows that the Sequential Extrapolation Coefficient leads to regret bounds via GOLF; recall that $\Pi = \{\pi_f \mid f \in \mathcal{F}\}$ is the set of greedy policies induced by $\mathcal{F}$.

**Theorem 5.** *Under Assumption 1, there exists an absolute constant $c$ such that for any $\delta \in (0,1]$ and $T \in \mathbb{N}_+$, if we choose $\beta = c \cdot \log(TH|\mathcal{F}|/\delta)$ in Algorithm 1, then with probability at least $1-\delta$, we have $\mathsf{Reg} \leq O\big(H\sqrt{\mathsf{SEC}_{\mathsf{RL}}(\mathcal{F},\Pi,T) \cdot T \cdot \log(TH|\mathcal{F}|/\delta)}\big)$.*

We defer the proof of Theorem 5 to Appendix F, and conclude by showing that the Sequential Extrapolation Coefficient subsumes coverability $C_{\mathsf{cov}}$ (Definition 2) and Bellman-Eluder dimension.

**Proposition 6** (Coverability $\implies$ SEC). $\mathsf{SEC}_{\mathsf{RL}}(\mathcal{F},\Pi,T) \leq O(C_{\mathsf{cov}} \cdot \log(T))$.

**Proposition 7** (Bellman-Eluder dim. $\implies$ SEC). $\mathsf{SEC}_{\mathsf{RL}}(\mathcal{F},\Pi,T) \leq O(\dim_{\mathsf{BE}}(\mathcal{F},\Pi,\sqrt{1/T}) \cdot \log(T))$.

The Sequential Extrapolation Coefficient can likely be generalized further along many directions (e.g., by allowing for different test functions in the vein of Du et al. (2021)). Further unifying these notions is an interesting question for future research; see Appendix F.3 for further discussion.

## 6 CONCLUSION

This paper initiates the systematic study of parallels between online and offline learnability in reinforcement learning and uncovers surprising new connections. The possible future directions include general theories under weaker notions of coverability or approximation conditions (see Appendix A for open problems) as well as the connection to the practical algorithm design.

ACKNOWLEDGEMENTS

Nan Jiang acknowledges funding support from ARL Cooperative Agreement W911NF-17-2-0196, NSF IIS-2112471, NSF CAREER award, and Adobe Data Science Research Award. Sham Kakade acknowledges funding from the Office of Naval Research under award N00014-22-1-2377 and the National Science Foundation Grant under award #CCF-1703574.

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

# APPENDIX

## A  DISCUSSION AND OPEN PROBLEMS

Toward a general theory beyond this paper, we highlight some exciting and challenging open problems for future research.

- *Linear feature coverage.* Our results in Section 4 show that the generalized coverability condition (Definition 4), which exploits the structure of the value function class $\mathcal{F}$, is not sufficient for online exploration. For the special case of linear functions ($\mathcal{F} = \{(x,a) \mapsto \langle \phi(x,a), \theta \rangle \mid \theta \in \Theta \subset \mathbb{R}^d\}$) a natural strengthening of this condition (Wang et al., 2021a; Zanette et al., 2021) is to assert the existence of a data distribution $\mu = \{\mu_h\}_{h=1}^H$ such that $\mathbb{E}_{d_h^\pi}[\phi(x_h,a_h)\phi(x_h,a_h)^\top] \preceq C \cdot \mathbb{E}_{\mu_h}[\phi(x_h,a_h)\phi(x_h,a_h)^\top]$ for some coverage parameter $C$. Is this condition (or a variant) sufficient for sample-efficient online exploration?

- *Further conditions from offline RL.* There are many conditions used to provide sample-efficient learning guarantees in offline RL beyond those considered in this paper, including (i) pushforward concentrability (Munos, 2003; Xie and Jiang, 2021), (ii) $L_p$ variants of concentrability (Farahmand et al., 2010; Xie and Jiang, 2020), and (iii) weight function realizability (Xie and Jiang, 2020; Jiang and Huang, 2020; Zhan et al., 2022). Which of these conditions can be adapted for online exploration, and to what extent?

## B    ADDITIONAL RELATED WORK

In this section we briefly highlight some relevant related work not otherwise discussed.

**Online RL with access to offline data.**   A separate line of work develops algorithms for online reinforcement learning that assume additional access to offline data gathered with a known data distribution $\mu$ or known exploratory policy (Abbasi-Yadkori et al., 2019; Xie et al., 2021b). These results are complementary to our own, since we assume only that a good exploratory distribution exists, but do not assume that such a distribution is known to the learner.

**Further structural conditions for online RL.**   While we have already discussed connections to Bellman Rank, Bilinear Classes, and Bellman-Eluder Dimension, another more general complexity measure is the *Decision-Estimation Coefficient* (Foster et al., 2021). One can show that the Decision-Estimation Coefficient is bounded by coverability, but to apply the algorithm in Foster et al. (2021), one must assume access to a realizable *model class* $\mathcal{M}$, which leads to regret bounds that scale with $\log|\mathcal{M}|$ rather than $\log|\mathcal{F}|$.

**Instance-dependent algorithms.**   Wagenmaker et al. (2022) provide instance-dependent guarantees for tabular PAC-RL which scale with a quantity called *gap-visitation complexity*. It is possible to bound the gap-visitation complexity in terms of coverability, but the lower-order sample complexity terms in this result have explicit dependence on the number of states, which our results avoid. For future work, it would be interesting to understand deeper connections between coverability and instance-dependent complexity measures (Wagenmaker et al., 2022; Wagenmaker and Jamieson, 2022; Dong and Ma, 2022). See also Wagenmaker and Jamieson (2022), which provides similar guarantees for linear MDPs.

## C    APPLICATION TO EXOGENOUS BLOCK MDPS

As an application of Theorem 1, we consider the problem of reinforcement learning in *Exogenous Block MDPs* (Ex-BMDPs). Following Efroni et al. (2021), an Ex-BMDP $M = (\mathcal{X}, \mathcal{A}, P, R, H, x_1)$ is defined by an (unobserved) *latent state space*, which consists of an *endogenous* state $s_h \in \mathcal{S}$ and *exogenous* state $\xi_h \in \Xi$, and an *observation process* which generates the observed state $x_h$. We first describe the dynamics for the latent space. Given initial endogenous and exogenous states $s_1 \in \mathcal{S}$ and $\xi_1 \in \Xi$, the latent states evolve via

$$s_{h+1} \sim P_h^{\mathrm{endo}}(s_h, a_h), \quad \text{and} \quad \xi_{h+1} \sim P_h^{\mathrm{exo}}(\xi_h);$$

that is while both states evolve in a temporally correlated fashion, only the endogenous state $s_h$ evolves as a function of the agent's action. The latent state $(s_h, \xi_h)$ is not observed. Instead, we observe

$$x_h \sim q_h(s_h, \xi_h),$$

where $q_h : \mathcal{S} \times \Xi \to \Delta(\mathcal{X})$ is an *emission distribution* with the property that $\mathrm{supp}(q_h(s, \xi)) \cap \mathrm{supp}(q_h(s', \xi')) = \varnothing$ if $(s, \xi) \neq (s', \xi')$. This property (*decodability*) ensures that there exists a unique mapping $\phi_h^\star : \mathcal{X} \to \mathcal{S}$ that maps the observed state $x_h$ to the corresponding endogenous latent state $s_h$. We assume that $R_h(x, a) = R_h(\phi_h^\star(x), a)$, which implies that optimal policy $\pi^\star$ depends only on the endogenous latent state, i.e. $\pi_h^\star(x) = \pi_h^\star(\phi_h^\star(x))$.

The main challenge of learning in block MDPs is that the decoder $\phi^\star$ is not known to the learner in advance. Indeed, given access to the decoder, one can obtain regret $\mathrm{poly}(H, |\mathcal{S}|, |\mathcal{A}|) \cdot \sqrt{T}$ by applying tabular reinforcement learning algorithms to the latent state space. In light of this, the aim of the Ex-BMDP setting is to obtain sample complexity guarantees that are independent of the size of the observed state space $|\mathcal{X}|$ and exogenous state space $|\Xi|$, and scale as $\mathrm{poly}(|\mathcal{S}|, |\mathcal{A}|, H, \log|\mathcal{F}|)$, where $\mathcal{F}$ is an appropriate class of function approximators (typically either a value function class $\mathcal{F}$ or a class of decoders $\Phi$ that attempts to model $\phi^\star$ directly).

Ex-BMDPs present substantial additional difficulties compared to classical block MDPs because we aim to avoid dependence on the size $|\Xi|$ of the exogenous latent state space. Here, the main challenge is that executing policies $\pi$ whose actions depend on $\xi_h$ can lead to spurious correlations between endogenous exogenous states. In spite of this apparent difficulty, we show that the coverability coefficient for this setting is always bounded by the number of *endogenous states*.

**Proposition 8.** *For any Ex-BMDP, $C_{\mathsf{cov}} \leq |\mathcal{S}| \cdot |\mathcal{A}|$.*

This bound is a consequence of a structural result from Efroni et al. (2021), which shows that for any $(s,a) \in \mathcal{S} \times \mathcal{A}$, all $x \in \mathcal{X}$ with $\phi^\star(x) = s$ admit a common policy that maximizes $d_h^\pi(x,a)$, and this policy is *endogenous*, i.e., only depends on the endogenous state $s_h = \phi_h^\star(x_h)$. As a corollary, we obtain the following regret bound.

**Corollary 9.** *For the Ex-BMDP setting, under Assumption 1, Algorithm 1 ensures that with probability at least $1 - \delta$,*

$$\mathsf{Reg} \leq O\big(H\sqrt{|\mathcal{S}||\mathcal{A}|T\log(^{TH|\mathcal{F}|}/\delta)}\log(T)\big).$$

Critically, this result scales only with the cardinality $|\mathcal{S}|$ for the endogenous latent state space, and with the capacity $\log|\mathcal{F}|$ for the value function class.

Let us briefly compare to prior work. For general Ex-BMDPs, existing complexity measures such as Bellman Rank and Bellman-Eluder dimension can be arbitrarily large (see discussion in Section 5). Existing algorithms either require that the endogenous latent dynamics $P^{\mathrm{endo}}$ are deterministic (Efroni et al., 2021) or allow for stochastic dynamics but heavily restrict the observation process (Efroni et al., 2022a). Corollary 9 is the first result for this setting that allows for stochastic latent dynamics and emission process, albeit with the extra assumption of completeness. This result is best thought of as a "luckiness" guarantee in the sense that it is unclear how to construct a value function class that is complete for every problem instance,[7] but the algorithm will succeed whenever $\mathcal{F}$ does happen to be complete for a given instance. Understanding whether general Ex-BMDPs are learnable without completeness is an interesting question for future work, and we are hopeful that the perspective of coverability will lead to further insights for this setting.

## C.1 INVARIANCE OF COVERABILITY

Proposition 8 is a consequence of two general *invariance* properties of coverability, which show that $C_{\mathsf{cov}}$ is unaffected by the following augmentations to the underlying MDP: (i) addition of rich observations, and (ii) addition of exogenous noise.

The first property shows that for a given MDP $M$, creating a new block MDP $M'$ by equipping $M$ with a decodable emission process (so that $M$ acts as a *latent MDP*), does not increase coverability.

**Proposition 10** (Invariance to rich observations). *Let an MDP $M = (\mathcal{S},\mathcal{A},P,R,H,s_1)$. Let $M' = (\mathcal{X},\mathcal{A},P',R',H,x_1)$ be the MDP defined implicitly by the following process. For each $h \in [H]$:*

- $s_{h+1} \sim P_h(s_h,a_h)$ *and* $r_h = R_h(s_h,a_h)$. *Here,* $s_h$ *is unobserved, and may be thought of as a latent state.*

- $x_h \sim q_h(s_h)$, *where* $q_h : \mathcal{S} \to \Delta(\mathcal{X})$ *is an* emission distribution *with the property that* $\mathrm{supp}(q_h(s)) \cap \mathrm{supp}(q_h(s')) = \varnothing$ *for* $s \neq s'$.

*Then, writing $C_{\mathsf{cov}}(M)$ to make the dependence on $M$ explicit, we have*

$$C_{\mathsf{cov}}(M') \leq C_{\mathsf{cov}}(M).$$

The second result shows that coverability is also preserved if we expand the state space to include temporally correlated exogenous state whose evolution does not depend on the agent's actions.

**Proposition 11** (Invariance to exogenous noise). *Let an MDP $M = (\mathcal{S},\mathcal{A},P,R,H,s_1)$, conditional distribution $P^{\mathrm{exo}} : \Xi \to \Delta(\Xi)$, and $\xi_1 \in \Xi$ be given, where $\Xi$ is an abstract set. Let $\mathcal{X} := \mathcal{S} \times \Xi$, and let $M' = (\mathcal{X},\mathcal{A},P',R',H,x_1)$ be the MDP with state $x_h = (s_h,\xi_h)$ defined implicitly by the following process. For each $h \in [H]$:*

- $s_{h+1} \sim P_h(s_h,a_h)$, $r_h = R_h(s_h,a_h)$.

- $\xi_{h+1} \sim P_h^{\mathrm{exo}}(\xi_h)$.

*Then we have*

$$C_{\mathsf{cov}}(M') \leq C_{\mathsf{cov}}(M).$$

---

[7]For example, it is not clear how to construct a complete value function class given access to a class of decoders $\Phi$ that contains $\phi^\star$.

This result is non-trivial because policies that act based on the endogenous state $s_h$ and $\xi_h$ can cause these processes to become coupled (Efroni et al., 2021), but holds nonetheless.

Proposition 8 can be deduced by combining Propositions 10 and 11 with the observation that any tabular (finite-state/action) MDP with $S$ states and $A$ actions has $C_{\text{cov}} \leq SA$. However, Propositions 10 and 11 yield more general results, since they imply that starting with any (potentially non-tabular) class of MDPs $\mathcal{M}$ with low coverability and augmenting it with rich observations and exogenous noise preserves coverability.

## C.2 PROOFS

**Proof of Proposition 8.** Let $h \in [H]$ be fixed. Let $z_h := (s_h, \xi_h)$. For each $z = (s, \xi) \in \mathcal{S} \times \Xi$, let $d_h^\pi(z) := \mathbb{P}^\pi(z_h = z)$. Proposition 4 of Efroni et al. (2021) shows that for all $z = (s, \xi)$, if we define $\pi_s = \arg\max_{\pi \in \Pi} \mathbb{P}^\pi(s_h = s)$, then

$$\max_{\pi \in \Pi} d_h^\pi(z) = d_h^{\pi_s}(z). \qquad (2)$$

That is, $\pi_s$ maximizes $\mathbb{P}^\pi(z_h = (s, \xi))$ for all $\xi \in \Xi$ simultaneously. With this in mind, let us define

$$\mu_h(x, a) = \frac{1}{|\mathcal{S}||\mathcal{A}|} \sum_{s \in \mathcal{S}} d_h^{\pi_s}(x).$$

We proceed to bound the concentrability coefficient for $\mu$. Fix $\pi \in \Pi$ and $x \in \mathcal{X}$, and let $z = (s, \xi) \in \mathcal{S} \times \Xi$ be the unique latent state such that $x \in \text{supp}(q_h(s, \xi))$. We first observe that

$$\frac{d_h^\pi(x, a)}{\mu_h(x, a)} \leq |\mathcal{S}||\mathcal{A}| \cdot \frac{d_h^\pi(x)}{d_h^{\pi_s}(x)}.$$

Next, since $x_h \sim q_h(z_h)$, we have

$$\frac{d_h^\pi(x)}{d_h^{\pi_s}(x)} = \frac{q_h(x \,|\, z) d_h^\pi(z)}{q_h(x \,|\, z) d_h^{\pi_s}(z)} = \frac{d_h^\pi(z)}{d_h^{\pi_s}(z)}.$$

Finally, by Eq. (2), we have

$$\frac{d_h^\pi(z)}{d_h^{\pi_s}(z)} \leq \frac{\max_\pi d_h^\pi(z)}{d_h^{\pi_s}(z)} = \frac{d_h^{\pi_s}(z)}{d_h^{\pi_s}(z)} = 1.$$

Since this holds for all $x \in \mathcal{X}$ simultaneously, this choice for $\mu_h$ certifies that that $C_{\text{cov}} \leq |\mathcal{S}||\mathcal{A}|$. $\qquad \square$

**Proof of Proposition 10.** Let $\Pi$ denote the space of all randomized policies acting on the latent state space $\mathcal{S}$, and let $\Pi'$ denote the space of all randomized policies acting on the observed state space $\mathcal{X}$. Let $\mathbb{P}^\pi$ denote distribution over trajectories in $M$ induced by $\pi \in \Pi$, and let $\mathbb{Q}^{\pi'}$ denote the distribution over trajectories in $M$ induced by $\pi' \in \Pi'$.

Fix $h \in [H]$, and let $\mu_h \in \Delta(\mathcal{S} \times \mathcal{A})$ witness the coverability coefficient for $M$. Define

$$\mu_h'(x, a) = q_h(x \,|\, \phi^\star(x)) \mu_h(\phi^\star(x), a),$$

where $\phi_h^\star : \mathcal{X} \to \mathcal{S}$ is the decoder that maps $x \in \mathcal{X}$ to the unique state $s \in \mathcal{S}$ such that $x \in \text{supp}(q_h(s))$. For any $\pi' \in \Pi'$ and $(x, a) \in \mathcal{X} \times \mathcal{A}$, letting $s = \phi_h^\star(x)$, we have

$$\frac{d_h^{\pi'}(x, a)}{\mu_h'(x, a)} = \frac{q_h(x \,|\, s) \mathbb{Q}^{\pi'}(s_h = s, a_h = a)}{q_h(x \,|\, s) \mu_h(s, a)} = \frac{\mathbb{Q}^{\pi'}(s_h = s, a_h = a)}{\mu_h(s, a)} \leq \frac{\max_{\pi' \in \Pi} \mathbb{Q}^{\pi'}(s_h = s, a_h = a)}{\mu_h(s, a)}.$$

Finally, because the observation process is decodable, we have $\max_{\pi' \in \Pi} \mathbb{Q}^{\pi'}(s_h = s, a_h = a) = \max_{\pi \in \Pi} \mathbb{P}^\pi(s_h = s, a_h = a)$, and

$$\frac{\max_{\pi \in \Pi} \mathbb{P}^\pi(s_h = s, a_h = a)}{\mu_h(s, a)} \leq C_{\text{cov}}(M).$$

$\square$

**Proof of Proposition 11.** Let $\Pi$ denote the space of all randomized policies acting on the latent state space $\mathcal{S}$, and let $\Pi'$ denote the space of all randomized policies acting on the observed state space $\mathcal{X}$. Let $\mathbb{P}^\pi$ denote distribution over trajectories in $M$ induced by $\pi \in \Pi$, and let $\mathbb{Q}^{\pi'}$ denote the distribution over trajectories in $M$ induced by $\pi' \in \Pi'$.

Fix $h \in [H]$, and let $\mu_h \in \Delta(\mathcal{S} \times \mathcal{A})$ witness the coverability coefficient for $M$. For $x = (s, \xi) \in \mathcal{S} \times \Xi$, let
$$\mu'_h(x, a) = \mathbb{Q}(\xi_h = \xi)\mu_h(s, a),$$
where $\mathbb{Q}(\xi_h = \xi)$ is the marginal probability of the event that $\xi_h = \xi$ in $M'$, which does not depend on the policy under consideration.

For any $\pi' \in \Pi$ and $(s, \xi, a) \in \mathcal{S} \times \Xi \times \mathcal{A}$, we have
$$\frac{d_h^{\pi'}(x, a)}{\mu'_h(x, a)} = \frac{\mathbb{Q}^{\pi'}(s_h = s, \xi_h = \xi, a_h = a)}{\mathbb{Q}(\xi_h = \xi)\mu_h(s, a)} \le \frac{\max_{\pi' \in \Pi'} \mathbb{Q}^{\pi'}(s_h = s, \xi_h = \xi, a_h = a)}{\mathbb{Q}(\xi_h = \xi)\mu_h(s, a)}.$$

From Propositions 3 and 4 of Efroni et al. (2021), we have $\max_{\pi' \in \Pi'} \mathbb{Q}^{\pi'}(s_h = s, \xi_h = \xi, a_h = a) = \mathbb{Q}(\xi_h = \xi) \cdot \max_{\pi' \in \Pi'} \mathbb{Q}^{\pi'}(s_h = s, a_h = a) = \mathbb{Q}(\xi_h = \xi) \cdot \max_{\pi \in \Pi} \mathbb{P}^\pi(s_h = s, a_h = a)$, so that
$$\frac{\max_{\pi' \in \Pi'} \mathbb{Q}^{\pi'}(s_h = s, \xi_h = \xi, a_h = a)}{\mathbb{Q}(\xi_h = \xi)\mu_h(s, a)} = \frac{\mathbb{Q}(\xi_h = \xi)\max_{\pi \in \Pi} \mathbb{P}^\pi(s_h = s, a_h = a)}{\mathbb{Q}(\xi_h = \xi)\mu_h(s, a)}$$
$$= \frac{\max_{\pi \in \Pi} \mathbb{P}^\pi(s_h = s, a_h = a)}{\mu_h(s, a)} \le C_{\mathsf{cov}}(M).$$

$\square$

# D  PROOFS AND ADDITIONAL DETAILS FROM SECTION 3

## D.1  GOLF ALGORITHM AND PROOFS FROM SECTION 3

---
**Algorithm 1** GOLF (Jin et al., 2021a)

---
**input:** Function class $\mathcal{F}$, confidence width $\beta > 0$.
**initialize:** $\mathcal{F}^{(0)} \leftarrow \mathcal{F}, \mathcal{D}_h^{(0)} \leftarrow \varnothing \ \forall h \in [H]$.
1:  **for** episode $t = 1, 2, ..., T$ **do**
2:    Select policy $\pi^{(t)} \leftarrow \pi_{f^{(t)}}$, where $f^{(t)} := \mathrm{argmax}_{f \in \mathcal{F}^{(t-1)}} f(x_1, \pi_{f,1}(x_1))$.
3:    Execute $\pi^{(t)}$ for one episode and obtain trajectory $(x_1^{(t)}, a_1^{(t)}, r_1^{(t)}), ..., (x_H^{(t)}, a_H^{(t)}, r_H^{(t)})$.
4:    Update dataset: $\mathcal{D}_h^{(t)} \leftarrow \mathcal{D}_h^{(t-1)} \cup \left\{ \left( x_h^{(t)}, a_h^{(t)}, x_{h+1}^{(t)} \right) \right\} \ \forall h \in [H]$.
5:    Compute confidence set:
$$\mathcal{F}^{(t)} \leftarrow \left\{ f \in \mathcal{F} : \mathcal{L}_h^{(t)}(f_h, f_{h+1}) - \min_{f'_h \in \mathcal{F}_h} \mathcal{L}_h^{(t)}(f'_h, f_{h+1}) \le \beta \ \forall h \in [H] \right\}, \quad (3)$$
$$\text{where} \quad \mathcal{L}_h^{(t)}(f, f') := \sum_{(x, a, r, x') \in \mathcal{D}_h^{(t)}} \left( f(x, a) - r - \max_{a' \in \mathcal{A}} f'(x', a') \right)^2, \ \forall f, f' \in \mathcal{F}.$$

6:  Output $\bar{\pi} = \mathsf{unif}(\pi^{(1:T)})$. `// For PAC guarantee only.`

---

**Lemma 12** (Jin et al. (2021a, Lemmas 39 and 40)). *Suppose Assumption 1 holds. Then if $\beta > 0$ is selected as in Theorem 1, then with probability at least $1 - \delta$, for all $t \in [T]$, Algorithm 1 satisfies*

*1.* $Q^\star \in \mathcal{F}^{(t)}$.

*2.* $\sum_{i < t} \mathbb{E}_{(x, a) \sim d_h^{(i)}} \left[ (f_h(x, a) - [\mathcal{T}_h f_{h+1}](x, a))^2 \right] \le O(\beta)$ *for all* $f \in \mathcal{F}^{(t)}$.

**Lemma 13** (Jiang et al. (2017, Lemma 1)). *For any value function $f = (f_1, ..., f_H)$,*
$$f_1(x_1, \pi_{f,1}(x_1)) - J(\pi_f) = \sum_{h=1}^H \mathbb{E}_{(x, a) \sim d_h^{\pi_f}} [f_h(x, a) - (\mathcal{T}_h f_{h+1})(x, a)].$$

**Lemma 14** (Equivalence of coverability and cumulative reachability). *The following definition is equivalent to Definition 2:*

$$C_{\mathsf{cov}} := \max_{h \in [H]} \sum_{(x,a) \in \mathcal{X} \times \mathcal{A}} \sup_{\pi \in \Pi} d_h^\pi(x,a).$$

**Proof of Lemma 14.** We relate coverability and cumulative reachability for each choice for $h \in [H]$.

*Coverability bounds cumulative reachability.* It follows immediately from the definition of coverability that if $\mu_h \in \Delta(\mathcal{X} \times \mathcal{A})$ realizes the value of $C_{\mathsf{cov}}$, then

$$
\begin{aligned}
\sum_{(x,a) \in \mathcal{X} \times \mathcal{A}} \max_{\pi \in \Pi} d_h^\pi(x,a) &= \sum_{(x,a) \in \mathcal{X} \times \mathcal{A}} \frac{\max_{\pi \in \Pi} d_h^\pi(x,a)}{\mu_h(x,a)} \mu_h(x,a) \\
&\leq \sum_{(x,a) \in \mathcal{X} \times \mathcal{A}} C_{\mathsf{cov}} \cdot \mu_h(x,a) \qquad \text{(by Definition 2)} \\
&= C_{\mathsf{cov}}.
\end{aligned}
$$

*Cumulative reachability bounds coverability.* Define $\mu_h(x,a) \propto \max_{\pi \in \Pi} d_h^\pi(x,a)$. Then for any $\pi \in \Pi$ and any $(x,a) \in \mathcal{X} \times \mathcal{A}$, we have

$$
\begin{aligned}
\frac{d_h^\pi(x,a)}{\mu_h(x,a)} &= \frac{d_h^\pi(x,a)}{\max_{\pi'' \in \Pi} d_h^{\pi''}(x,a) / \sum_{(x',a') \in \mathcal{X} \times \mathcal{A}} \max_{\pi' \in \Pi} d_h^{\pi'}(x',a')} \\
&\leq \sum_{(x',a') \in \mathcal{X} \times \mathcal{A}} \max_{\pi' \in \Pi} d_h^{\pi'}(x',a').
\end{aligned}
$$

This completes the proof. $\qquad\square$

**Proof of Theorem 1.** Equipped with Lemma 14, we prove Theorem 1.

**Preliminaries.** For each $t$, we define $\delta_h^{(t)}(\cdot,\cdot) := f_h^{(t)}(\cdot,\cdot) - (\mathcal{T}_h f_{h+1}^{(t)})(\cdot,\cdot)$, which may be viewed as a "test function" at level $h$ induced by $f^{(t)} \in \mathcal{F}$. We adopt the shorthand $d_h^{(t)} \equiv d_h^{\pi^{(t)}}$, and we define

$$\widetilde{d}_h^{(t)}(x,a) := \sum_{i=1}^{t-1} d_h^{(i)}(x,a), \quad \text{and} \quad \mu_h^\star := \operatorname*{argmin}_{\mu_h \in \Delta(\mathcal{X} \times \mathcal{A})} \sup_{\pi \in \Pi} \left\| \frac{d_h^\pi}{\mu_h} \right\|_\infty. \tag{4}$$

That is, $\widetilde{d}_h^{(t)}$ unnormalized average of all state visitations encountered prior to step $t$, and $\mu_h^\star$ is the distribution that attains the value of $C_{\mathsf{cov}}$ for layer $h$.[8] Throughout the proof, we perform a slight abuse of notation and write $\mathbb{E}_{\widetilde{d}_h^{(t)}}[f] := \sum_{i=1}^{t-1} \mathbb{E}_{d_h^{(i)}}[f]$ for any function $f : \mathcal{X} \times \mathcal{A} \to \mathbb{R}$.

**Regret decomposition.** As a consequence of completeness (Assumption 1) and the construction of $\mathcal{F}^{(t)}$, a standard concentration argument (Lemma 12) guarantees that with probability at least $1 - \delta$, for all $t \in [T]$:

$$\text{(i) } Q^\star \in \mathcal{F}^{(t)}, \quad \text{and} \quad \text{(ii) } \sum_{x,a} \widetilde{d}_h^{(t)}(x,a) \big(\delta_h^{(t)}(x,a)\big)^2 \leq O(\beta). \tag{5}$$

We condition on this event going forward. Since $Q^\star \in \mathcal{F}^{(t)}$, we are guaranteed that $f^{(t)}$ is optimistic (i.e., $f_1^{(t)}(x_1, \pi_{f^{(t)},1}(x_1)) \geq Q_1^\star(x_1, \pi_{f^\star,1}(x_1))$), and a regret decomposition for optimistic algorithms (Lemma 13) allows us to relate regret to the average Bellman error under the learner's sequence of policies:

$$\mathsf{Reg} \leq \sum_{t=1}^T \Big( f_1^{(t)}(x_1, \pi_{f_1^{(t)},1}(x_1)) - J(\pi^{(t)}) \Big) = \sum_{t=1}^T \sum_{h=1}^H \mathbb{E}_{(x,a) \sim d_h^{(t)}} \Big[ \underbrace{f_h^{(t)}(x,a) - (\mathcal{T}_h f_{h+1}^{(t)})(x,a)}_{=: \delta_h^{(t)}(x,a)} \Big].$$

---

[8]If the minimum in Eq. (4) is not obtained, we can repeat the argument that follows for each element of a limit sequence attaining the infimum.

To proceed, we use a change of measure argument to relate the on-policy *average* Bellman error $\mathbb{E}_{(x,a)\sim d_h^{(t)}}[\delta_h^{(t)}(x,a)]$ appearing above to the in-sample *squared* Bellman error $\mathbb{E}_{(x,a)\sim \widetilde{d}_h^{(t)}}[\delta_h^{(t)}(x,a)^2]$; the latter is small as a consequence of Eq. (5). Unfortunately, naive attempts at applying change-of-measure fail because during the initial rounds of exploration, the on-policy and in-sample visitation probabilities can be very different, making it impossible to relate the two quantities (i.e., any natural notion of extrapolation error will be arbitrarily large).

To address this issue, we introduce the notion of a "burn-in" phase for each state-action pair $(x,a)\in \mathcal{X}\times\mathcal{A}$ by defining

$$\tau_h(x,a) = \min\left\{t\,\middle|\,\widetilde{d}_h^{(t)}(x,a)\geq C_{\mathsf{cov}}\cdot\mu_h^{\star}(x,a)\right\},$$

which captures the earliest time at which $(x,a)$ has been explored sufficiently; we refer to $t<\tau_h(x,a)$ as the burn-in phase for $(x,a)$.

Going forward, let $h\in[H]$ be fixed. We decompose regret into contributions from the burn-in phase for each state-action pair, and contributions from pairs which have been explored sufficiently and reached a stable phase "stable phase".

$$\underbrace{\sum_{t=1}^{T}\mathbb{E}_{(x,a)\sim d_h^{(t)}}\left[\delta_h^{(t)}(x,a)\right]}_{\text{on-policy average Bellman error}} = \underbrace{\sum_{t=1}^{T}\mathbb{E}_{(x,a)\sim d_h^{(t)}}\left[\delta_h^{(t)}(x,a)\mathbb{1}[t<\tau_h(x,a)]\right]}_{\text{burn-in phase}} + \underbrace{\sum_{t=1}^{T}\mathbb{E}_{(x,a)\sim d_h^{(t)}}\left[\delta_h^{(t)}(x,a)\mathbb{1}[t\geq\tau_h(x,a)]\right]}_{\text{stable phase}}.$$

We will not show that every state-action pair leaves the burn-in phase. Instead, we use coverability to argue that the contribution from pairs that have not left this phase is small on average. In particular, we use that $|\delta_h^{(t)}|\leq 1$ to bound

$$\sum_{t=1}^{T}\mathbb{E}_{(x,a)\sim d_h^{(t)}}\left[\delta_h^{(t)}(x,a)\mathbb{1}[t<\tau_h(x,a)]\right]\leq\sum_{x,a}\sum_{t<\tau_h(x,a)}d_h^{(t)}(x,a)=\sum_{x,a}\widetilde{d}_h^{(\tau_h(x,a))}(x,a)\leq 2C_{\mathsf{cov}}\sum_{x,a}\mu_h^{\star}(x,a)=2C_{\mathsf{cov}},$$

where the last inequality holds because

$$\widetilde{d}_h^{(\tau_h(x,a))}(x,a)=\widetilde{d}_h^{(\tau_h(x,a)-1)}(x,a)+d_h^{(\tau_h(x,a)-1)}(x,a)\leq 2C_{\mathsf{cov}}\cdot\mu_h^{\star}(x,a),$$

which follows from Eq. (4) and the definition of $\tau_h$.

For the stable phase, we apply change-of-measure as follows:

$$\sum_{t=1}^{T}\mathbb{E}_{(x,a)\sim d_h^{(t)}}\left[\delta_h^{(t)}(x,a)\mathbb{1}[t\geq\tau_h(x,a)]\right]$$

$$=\sum_{t=1}^{T}\sum_{x,a}d_h^{(t)}(x,a)\left(\frac{\widetilde{d}_h^{(t)}(x,a)}{\widetilde{d}_h^{(t)}(x,a)}\right)^{1/2}\delta_h^{(t)}(x,a)\mathbb{1}[t\geq\tau_h(x,a)]$$

$$\leq\underbrace{\sqrt{\sum_{t=1}^{T}\sum_{x,a}\frac{\left(\mathbb{1}[t\geq\tau_h(x,a)]d_h^{(t)}(x,a)\right)^2}{\widetilde{d}_h^{(t)}(x,a)}}}_{\texttt{(I)}:\text{ extrapolation error}}\cdot\underbrace{\sqrt{\sum_{t=1}^{T}\sum_{x,a}\widetilde{d}_h^{(t)}(x,a)\left(\delta_h^{(t)}(x,a)\right)^2}}_{\texttt{(II)}:\text{ in-sample }\textit{squared}\text{ Bellman error}}, \qquad (6)$$

where the last inequality is an application of Cauchy-Schwarz. Using part (II) of Eq. (5), we bound the in-sample error above by

$$\texttt{(II)}\leq O\left(\sqrt{\beta T}\right). \qquad (7)$$

**Bounding the extrapolation error using coverability.**    To proceed, we show that the extrapolation error (I) is controlled by coverability. We begin with a scalar variant of the standard elliptic potential lemma (Lattimore and Szepesvári, 2020); this result is proven in the sequel.

**Lemma 15** (Per-state-action elliptic potential lemma). *Let $d^{(1)},d^{(2)},...,d^{(T)}$ be an arbitrary sequence of distributions over a set $\mathcal{Z}$ (e.g., $\mathcal{Z}=\mathcal{X}\times\mathcal{A}$), and let $\mu\in\Delta(\mathcal{Z})$ be a distribution such that $d^{(t)}(z)/\mu(z)\leq C$ for all $(z,t)\in\mathcal{Z}\times[T]$. Then for all $z\in\mathcal{Z}$, we have*

$$\sum_{t=1}^{T}\frac{d^{(t)}(z)}{\sum_{i<t}d^{(i)}(z)+C\cdot\mu(z)}\leq O(\log(T)).$$

We bound the extrapolation error (I) by applying Lemma 15 on a *per-state basis*, then using coverability (and the equivalence to cumulative reachability) to argue that the potentials from different state-action pairs average out. Observe that by the definition of $\tau_h$, we have that for all $t \geq \tau_h(s,a)$, $\widetilde{d}_h^{(t)}(x,a) \geq C_{\mathsf{cov}}\mu_h^\star(x,a) \Rightarrow \widetilde{d}_h^{(t)}(x,a) \geq \frac{1}{2}(\widetilde{d}_h^{(t)}(x,a) + C_{\mathsf{cov}}\mu_h^\star(x,a))$, which allows us to bound term (I) of extrapolation error by

$$
\sum_{t=1}^T \sum_{x,a} \frac{\left(\mathbb{1}[t \geq \tau_h(x,a)] d_h^{(t)}(x,a)\right)^2}{\widetilde{d}_h^{(t)}(x,a)} \leq 2 \sum_{t=1}^T \sum_{x,a} \frac{d_h^{(t)}(x,a) \cdot d_h^{(t)}(x,a)}{\widetilde{d}_h^{(t)}(x,a) + C_{\mathsf{cov}} \cdot \mu_h^\star(x,a)}
$$

$$
\leq 2 \sum_{t=1}^T \sum_{x,a} \max_{t' \in [T]} d_h^{(t')}(x,a) \cdot \frac{d_h^{(t)}(x,a)}{\widetilde{d}_h^{(t)}(x,a) + C_{\mathsf{cov}} \cdot \mu_h^\star(x,a)}
$$

$$
\leq 2 \underbrace{\left(\sum_{x,a} \max_{t \in [T]} d_h^{(t)}(x,a)\right)}_{\leq C_{\mathsf{cov}} \text{ by Lemma 14}} \cdot \underbrace{\left(\max_{(s,a) \in \mathcal{S} \times \mathcal{A}} \sum_{t=1}^T \frac{d_h^{(t)}(x,a)}{\widetilde{d}_h^{(t)}(x,a) + C_{\mathsf{cov}} \cdot \mu_h^\star(x,a)}\right)}_{\leq O(\log(T)) \text{ by Lemma 15}}
$$

$$
\leq O(C_{\mathsf{cov}} \log(T)). \tag{8}
$$

To conclude, we substitute Eqs. (7) and (8) into Eq. (6), which gives

$$
\mathsf{Reg} \leq \sum_{h=1}^H \mathbb{E}_{(x,a) \sim d_h^{(t)}} \left[\delta_h^{(t)}(x,a)\right] \leq O\left(H\sqrt{C_{\mathsf{cov}} \cdot \beta T \log(T)}\right).
$$

$\square$

**Proof of Lemma 15.** Using the fact for any $u \in [0,1]$, $u \leq 2\log(1+u)$, we have

$$
\sum_{t=1}^T \frac{d^{(t)}(z)}{\sum_{i<t} d^{(i)}(z) + C \cdot \mu(x,a)} \leq 2 \sum_{t=1}^T \log\left(1 + \frac{d^{(t)}(x,a)}{\sum_{i<t} d^{(i)}(z) + C \cdot \mu(x,a)}\right)
$$
$$
\text{(since } d^{(t)}(x,a)/\mu(x,a) \leq C \ \forall t \in [T])
$$

$$
= 2 \sum_{t=1}^T \log\left(\frac{\sum_{i<t+1} d^{(i)}(z) + C \cdot \mu(x,a)}{\sum_{i<t} d^{(i)}(z) + C \cdot \mu(x,a)}\right)
$$

$$
= 2\log\left(\prod_{t=1}^T \frac{\sum_{i<t+1} d^{(i)}(z) + C \cdot \mu(x,a)}{\sum_{i<t} d^{(i)}(z) + C \cdot \mu(x,a)}\right)
$$

$$
= 2\log\left(\frac{\sum_{i=1}^T d^{(i)}(z) + C \cdot \mu(x,a)}{C \cdot \mu(x,a)}\right)
$$

$$
\leq 2\log(T+1). \qquad \text{(since } d^{(t)}(x,a)/\mu(x,a) \leq C \ \forall t \in [T])
$$

This completes the proof. $\square$

# E PROOFS AND ADDITIONAL DETAILS FROM SECTION 4

## E.1 ADDITIONAL DETAILS: OFFLINE RL

**Proposition 16** (Generalized concentrability is sufficient for offline RL)**.** *Given access to an offline data distribution $\mu$ satisfying generalized concentrability (Definition 4), if $\mathcal{F}$ satisfies Assumption 1, one can find an $\varepsilon$-optimal policy using* $\mathrm{poly}(\mathfrak{C}_{\mathsf{conc}}(\mu,\mathcal{F}), H, \log|\mathcal{F}|, \varepsilon^{-1})$ *samples.*

**Proof of Proposition 16.** Given an offline dataset $\mathcal{D} = \{\mathcal{D}_h\}_{h=1}^H$ with $n$ samples for each layer $h \in [H]$ under the distribution $\mu_h$, the MSBO algorithm (e.g., Xie and Jiang, 2020) produces a value function $\widehat{f} \in \mathcal{F}$ of the form

$$
\widehat{f} \leftarrow \operatorname*{argmin}_{f \in \mathcal{F}} \sum_{h=1}^H \left(\mathcal{L}_h(f_h, f_{h+1}) - \min_{f_h' \in \mathcal{F}_h} \mathcal{L}_h(f_h', f_{h+1})\right),
$$

$$\text{where} \quad \mathcal{L}_h(f,f') := \sum_{(x,a,r,x') \in \mathcal{D}_h} \left( f(x,a) - r - \max_{a' \in \mathcal{A}} f'(x',a') \right)^2, \forall f, f' \in \mathcal{F}.$$

By adapting the proof of Theorem 5 of Xie and Jiang (2020) (or Lemma 12), one can show that under Assumption 1, with probability at least $1 - \delta$, $\widehat{f}$ satisfies

$$\sum_{h=1}^{H} \mathbb{E}_{(x,a) \sim \mu_h} \left[ \left( (\widehat{f}_h(x,a) - \mathcal{T}_h \widehat{f}_{h+1})(x,a) \right)^2 \right] \leq H \cdot \frac{\log(|\mathcal{F}|/\delta)}{n}.$$

The result now follows by applying an adaptation of Xie and Jiang (2020, Corollary 4), which shows that for any $f \in \mathcal{F}$,

$$J(\pi^\star) - J(\pi_f) \leq 2 \max_{\pi \in \Pi} \sum_{h=1}^{H} \mathbb{E}_{(x,a) \sim d_h^\pi} [|f_h(x,a) - (\mathcal{T}_h f_{h+1})(x,a)|]$$

$$\leq 2 \sqrt{H \max_{\pi \in \Pi} \sum_{h=1}^{H} \mathbb{E}_{(x,a) \sim \mu_h} [(f_h(x,a) - (\mathcal{T}_h f_{h+1})(x,a))^2]}$$

$$\leq 2 \sqrt{H \mathfrak{C}_{\mathsf{conc}}(\mu, \mathcal{F}) \sum_{h=1}^{H} \mathbb{E}_{(x,a) \sim \mu_h} [(f_h(x,a) - (\mathcal{T}_h f_{h+1})(x,a))^2]} \quad \text{(by Definition 4)}$$

$$\leq 2H \sqrt{\frac{\mathfrak{C}_{\mathsf{conc}}(\mu, \mathcal{F}) \log(|\mathcal{F}|/\delta)}{n}}.$$

$\square$

### E.2 PROOFS FROM SECTION 4

**Proof of Theorem 3.** Assume without loss of generality that $H \leq \min\{\log_2(X), C\}$; if this does not hold, the result is obtained by applying the argument that follows with $H' = \min\{H, \lfloor \log_2(X) \rfloor, C\}$.

We consider a family of deterministic MDPs with horizon $H$. We use a layered state space $\mathcal{X} = \mathcal{X}_1 \cup \cdots \cup \mathcal{X}_H$, where only states in $\mathcal{X}_h$ are reachable at layer $h$. The state space is a binary tree of depth $H - 1$, which has $\sum_{h=0}^{\log_2(X)-1} 2^h = X - 1$ states. The are two actions, left and right, which determine whether the next state is the left or right successor in the tree.

For each MDP in the family, we allow a single action at a single leaf at $h = H$ to have reward $r_H = 1$, give reward 0 to all actions in all other states. For each such MDP, we use $(x_H^\star, a_H^\star)$ to denote the single state-action pair with $r = 1$. We also use $(x_h^\star, a_h^\star)$ for $h \in [H]$ to denote the unique path from $x_1$ to $(x_H^\star, a_H^\star)$. Note that the optimal policy is to follow this path, i.e.

$$d_h^{\pi^\star}(x,a) = \mathbb{1}[(x,a) = (x_h^\star, a_h^\star)].$$

We choose $\mathcal{F}_h$ to be the set of all possible indicator functions for a single state-action pair:

$$\mathcal{F}_h := \{f_h(x',a') = \mathbb{1}(x' = x, a' = a) \mid \forall (x,a) \in \mathcal{X}_h \times \mathcal{A}\}.$$

We define $\mathcal{F} = \mathcal{F}_1 \times \cdots \times \mathcal{F}_H$. Note that for each $h \in [H]$,

$$Q_h^\star(x_h,a_h) = \mathbb{1}(x_h = x_h^\star, a_h = a_h^\star) \in \mathcal{F}_h.$$

In addition, we have $\log |\mathcal{F}| \leq H \log(2X)$.

**Completeness.** We first verify that the construction satisfies completeness. Fix $f_h \in \mathcal{F}_h$, and let $f_h(x,a) = \mathbb{1}(x = x_{f,h}, a = a_{f,h})$ for some $(x_{f,h}, a_{f,h}) \in \mathcal{X}_h \times \mathcal{A}$. Then for any $(x_{h-1}, a_{h-1}) \in \mathcal{X}_{h-1} \times \mathcal{A}$, we consider two cases. First, if $x_{f,h}$ is not the unique successor of $(x_{h-1}, a_{h-1})$, then $(\mathcal{T}_{h-1} f_h)(x_{h-1}, a_{h-1}) = 0$. Otherwise,

$$(\mathcal{T}_{h-1} f_h)(x_{h-1}, a_{h-1}) = \sum_{x_h} \mathbb{P}(x_h \mid x_{h-1}, a_{h-1}) \max_{a_h} f(x_h, a_h)$$

$$= \mathbb{P}(x_{f,h} \mid x_{h-1}, a_{h-1}). \quad (\text{as } \max_{a_h} f(x_h, a_h) = \mathbb{1}(x_h = x_{f,h}))$$

$$= 1.$$

This means $\mathcal{T}_{h-1} f_h \in \mathcal{F}_{h-1}$, because there exists a single $(x_{h-1}, a_{h-1})$ pair in $\mathcal{X}_{h-1} \times \mathcal{A}$ such that $(\mathcal{T}_{h-1} f_h)(x_{h-1}, a_{h-1}) \neq 0$.

**Generalized coverability.** We now show that the construction satisfies generalized coverability. Fix an MDP in the family with optimal path $\{(x_h^\star, a_h^\star)\}_{h=1}^H$. We will show that for all $f = f_{1:H} \in \mathcal{F}$, if $f_{1:H} \neq Q_{1:H}^\star$, then there exists $h' \in [H]$, such that

$$\mathbb{E}_{d_{h'}^{\pi^\star}}\left[(f_h(x_{h'}, a_{h'}) - (\mathcal{T}_{h'} f_{h'+1})(x_{h'}, a_{h'}))^2\right] = (f_{h'}(x_{h'}^\star, a_{h'}^\star) - (\mathcal{T}_{h'} f_{h'+1})(x_{h'}^\star, a_{h'}^\star))^2 = 1. \quad (9)$$

From here, the result will follow by choosing $\mu_h = d_h^{\pi^\star} \ \forall h \in [H]$. Indeed, using the boundedness of $f_{1:H} \in \mathcal{F}$, we have

$$\sum_{h=1}^H \mathbb{E}_{d_h^\pi}\left[(f_h(x_h, a_h) - (\mathcal{T}_h f_{h+1})(x_h, a_h))^2\right] \leq H,$$

for all $\pi \in \Pi$, meaning that Eq. (9) implies that $\mathfrak{C}_{\mathsf{cov}}(\mu, \mathcal{F}) \leq H \leq C$ in this problem instance.

We proceed to prove Eq. (9). Based on the definition of $\mathcal{F}$, we know that $(f_h(x_h^\star, a_h^\star) - (\mathcal{T}_h f_{h+1})(x_h^\star, a_h^\star))^2 \in \{0, 1\}$ for all $h \in [H]$. Therefore, if we assume by contradiction that $f_{1:H} \neq Q_{1:H}^\star$ and there does not exist an $h' \in [H]$ that satisfies Eq. (9), we must have

$$f_h(x_h^\star, a_h^\star) = (\mathcal{T}_h f_{h+1})(x_h^\star, a_h^\star), \quad \forall h \in [H]. \quad (10)$$

By the condition Eq. (10), we have $(\mathcal{T}_H f_{H+1})(x_h^\star, a_H^\star) = R_H(x_H^\star, a_H^\star) = 1$, which implies that $f_h(x_h^\star, a_h^\star) = 1$ for all $h \in [H]$. From the construction of $\mathcal{F}$, we know $Q_{1:H}^\star$ is the only function with $Q_h^\star(x_h^\star, a_h^\star) = 1$ for all $h \in [H]$, which gives the desired contraction, and proves that such $h' \in [H]$ must exist, establishing Eq. (9).

**Lower bound on sample complexity.** A lower bound of $2^{\Omega(H)}$ samples to learn a $0.1$-optimal with probability $0.9$ follows from standard lower bounds for binary tree-structured MDPs (Krishnamurthy et al., 2016; Jiang et al., 2017) (recall that since there are $2^{H/2}$ leaves at layer $H$, and only one has non-zero reward, finding a policy with non-trivial regret is no easier than solving a multi-armed bandit problem with $2^{H/2}$ actions and binary rewards). $\qquad\square$

# F PROOFS AND ADDITIONAL RESULTS FROM SECTION 5

## F.1 ADDITIONAL DETAILS: SEQUENTIAL EXTRAPOLATION COEFFICIENT VERSUS BELLMAN-ELUDER DIMENSION

The discussion in Section 5 (in particular, Proposition 4 shows that Bellman-Eluder dimension and Bellman rank fail to capture coverability as a result of only considering average Bellman error rather than squared Bellman error. In light of this observation, a seemingly reasonable fix is to adapt the Bellman-Eluder dimension to consider squared Bellman error rather than average Bellman error. Consider the following variant.

**Definition 9** (Squared Bellman-Eluder dimension). *We define the Squared Bellman-Eluder dimension* $\dim_{\mathsf{BE}}^{\mathsf{sq}}(\mathcal{F}, \Pi, \varepsilon, h)$ *for layer* $h$ *is the largest* $d \in \mathbb{N}$ *such that there exist sequences* $\{d_h^{(1)}, d_h^{(2)}, ..., d_h^{(d)}\} \subseteq \mathfrak{D}_h^\Pi$ *and* $\{\delta_h^{(1)}, ..., \delta_h^{(d)}\} \subseteq \mathcal{F}_h - \mathcal{T}_h \mathcal{F}_{h+1}$ *such that for all* $t \in [d]$,

$$\left|\mathbb{E}_{d_h^{(t)}}[\delta_h^{(t)}]\right| > \varepsilon^{(t)}, \quad \text{and} \quad \sqrt{\sum_{i=1}^{t-1} \mathbb{E}_{d_h^{(i)}}\left[(\delta_h^{(t)})^2\right]} \leq \varepsilon^{(t)}, \quad (11)$$

*for* $\varepsilon^{(1)}, ..., \varepsilon^{(d)} \geq \varepsilon$. *We define* $\dim_{\mathsf{BE}}^{\mathsf{sq}}(\mathcal{F}, \Pi, \varepsilon) = \max_{h \in [H]} \dim_{\mathsf{BE}}^{\mathsf{sq}}(\mathcal{F}, \Pi, \varepsilon, h)$.

This definition is identical to Definition 6, except that the constraint $\sqrt{\sum_{i=1}^{t-1}(\mathbb{E}_{d_h^{(i)}}[\delta_h^{(t)}])^2} \leq \varepsilon^{(t)}$ in Definition 6 has been replaced by the constraint $\sqrt{\sum_{i=1}^{t-1}\mathbb{E}_{d_h^{(i)}}\left[(\delta_h^{(t)})^2\right]} \leq \varepsilon^{(t)}$, which uses squared Bellman error instead of average Bellman error. By adapting the analysis of Jin et al. (2021a) it is possible to show that this definition yields $\mathsf{Reg} \leq \widetilde{O}\left(H\sqrt{\inf_{\varepsilon > 0}\{\varepsilon^2 T + \dim_{\mathsf{BE}}^{\mathsf{sq}}(\mathcal{F}, \Pi, \varepsilon)\} \cdot T \log|\mathcal{F}|}\right)$. If one could show that $\dim_{\mathsf{BE}}^{\mathsf{sq}}(\mathcal{F}, \Pi, \varepsilon) \lesssim C_{\mathsf{cov}} \cdot \mathrm{polylog}(\varepsilon^{-1})$, this would recover Theorem 1. Unfortunately, it turns out that in general, one can have $\dim_{\mathsf{BE}}^{\mathsf{sq}}(\mathcal{F}, \Pi, \varepsilon) = \Omega(C_{\mathsf{cov}}/\varepsilon)$, which leads to suboptimal $T^{2/3}$-type regret using the result above. The following result shows that this guarantee cannot be improved without changing the complexity measure under consideration.

**Proposition 17.** *Fix $T \in \mathbb{N}$, and let $\varepsilon_T := T^{-1/3}$. There exist MDP class/policy class/value function class tuples $(\mathcal{M}_1, \Pi_1, \mathcal{F}_1)$ and $(\mathcal{M}_2, \Pi_2, \mathcal{F}_2)$ with the following properties.*

1. *All MDPs in $\mathcal{M}_1$ (resp. $\mathcal{M}_2$) satisfy Assumption 1 with respect to $\mathcal{F}_1$ (resp. $\mathcal{F}_2$). In addition, $\log|\mathcal{F}_1| = \log|\mathcal{F}_2| = \widetilde{O}(1)$.*

2. *For all MDPs in $\mathcal{M}_1$, we have $\dim^{\mathsf{sq}}_{\mathsf{BE}}(\mathcal{F}_1, \Pi_1, \varepsilon_T) \propto 1/\varepsilon_T$, and any algorithm must have $\mathbb{E}[\mathsf{Reg}] \geq \Omega(T^{2/3})$ for some MDP in the class*

3. *For all MDPs in $\mathcal{M}_2$, we also have $\dim^{\mathsf{sq}}_{\mathsf{BE}}(\mathcal{F}_2, \Pi_2, \varepsilon_T) \propto 1/\varepsilon_T$, yet $C_{\mathsf{cov}} = O(1)$ and GOLF attains $\mathbb{E}[\mathsf{Reg}] \leq \widetilde{O}(\sqrt{T})$.*

This result shows that there are two classes for which the optimal rate differs polynomially ($\Omega(T^{2/3})$ vs. $\widetilde{O}(\sqrt{T})$), yet the Bellman-Eluder dimension has the same size, and implies that the Bellman-Eluder dimension cannot provide rates better than $\Omega(T^{2/3})$ for classes with low coverability in general. Informally, the reason why Bellman-Eluder dimension fails capture the optimal rates for the problem instances in Proposition 17 is that the definition in Eq. (11) only checks whether the average Bellman error violates the threshold $\varepsilon$, and does not consider how far the error violates the threshold (e.g., $|\mathbb{E}_{d_h^{(t)}}[\delta_h^{(t)}]| > \varepsilon$ and $|\mathbb{E}_{d_h^{(t)}}[\delta_h^{(t)}]| > 1$ are counted the same).

In spite of this counterexample, it is possible to show that the Bellman-Eluder dimension with squared Bellman error is always bounded by the Sequential Extrapolation Coefficient up to a $\mathrm{poly}(\varepsilon^{-1})$ factor, and hence can always be bounded by coverability, albeit suboptimally.

**Proposition 18.** *Let $\mathcal{F}$ be a $[0,1]$-valued function class. For all $T \in \mathbb{N}$ and $\varepsilon > 0$, we have $\min\{\dim^{\mathsf{sq}}_{\mathsf{BE}}(\mathcal{F}, \Pi, \varepsilon), T\} \leq \frac{\mathsf{SEC}_{\mathsf{RL}}(\mathcal{F}, \Pi, T)}{\varepsilon^2}$.*

### F.1.1 Proofs from Additional Details

**Proof of Proposition 17.** Let the time horizon $T \in \mathbb{N}$ be fixed. We first construct the class $\mathcal{M}_1$ and verify that it satisfies the properties in the statement of Proposition 17, then do the same for $\mathcal{M}_2$.

**Class $\mathcal{M}_1$.** We choose $\mathcal{M}_1$ to be a class of bandit problems with $H = 1$. Let a parameter $\varepsilon_1 \in [0, 1/2]$ be fixed, and let $A := \varepsilon_1^{-1}$. We define $\mathcal{M}_1 = \{M^{(1)}, ..., M^{(A)}\}$, where for each $M^{(i)}$:

- The action space is $\mathcal{A} = \{1, ..., A\}$.
- The reward distribution for action $a \in \mathcal{A}$ in state $x_1$ is $\mathsf{Ber}(1/2 + \varepsilon_1 \mathbb{1}\{a = i\})$.

For each $i \in M^{(i)}$, the mean reward function is $f_1^{(i)}(x_1, \pi) = 1/2 + \varepsilon_1 \mathbb{1}\{a = i\}$. We define $\mathcal{F} = \{f^{(i)}\}_{i=1}^{A}$ and $\Pi = \{\pi_f \mid f \in \mathcal{F}\}$. Note that since $H = 1$, completeness of $\mathcal{F}$ is immediate.

*Lower bounding the Bellman-Eluder dimension.* Let $M^{(A)}$ be the underlying instance. We will lower bound the Bellman-Eluder dimension for layer $h = 1$. Consider the sequence $d_1^{(1)}, ..., d_1^{(A-1)}$, where $d_1^{(t)} := d_1^{\pi_f(t)}$ and $\delta_1^{(1)}, ..., \delta_1^{(A-1)}$, where $\delta_1^{(t)} := f_1^{(t)} - \mathcal{T}_1 f_2^{(t)} = f_1^{(t)} - f_1^{(A)}$ (recall that we adopt the convention $f_{H+1} = 0$). Observe that for each $t \in [A-1]$, we have

$$|\mathbb{E}_{d_1^{(t)}}[\delta_1^{(t)}](x_1, a_1)| = |f_1^{(t)}(x_1, t) - f_1^{(A)}(x_1, t)| = \varepsilon_1,$$

yet

$$\sum_{i < t} \mathbb{E}_{d_1^{(i)}}\left[(\delta_1^{(t)}(x_1, a_1))^2\right] = \varepsilon_1 \sum_{i < t} (f_1^{(t)}(x_1, i) - f_1^{(A)}(x_1, i))^2 = 0.$$

This certifies that $\dim^{\mathsf{sq}}_{\mathsf{BE}}(\mathcal{F}, \Pi, \varepsilon) \geq A - 1 \geq \varepsilon_1^{-1}/2$ for all $\varepsilon < \varepsilon_1$.

*Lower bounding regret.* A standard result (e.g., Lattimore and Szepesvári, 2020) is that for any family of multi-armed bandit instances of the form $\{M^{(1)}, ..., M^{(A)}\}$, where $M^{(i)}$ has Bernoulli rewards with mean $1/2 + \Delta \mathbb{1}\{a = i\}$ for $\Delta \leq 1/4$, any algorithm must have regret

$$\mathbb{E}[\mathsf{Reg}] \geq \Omega(1) \cdot \min\left\{\Delta T, \frac{A}{\Delta}\right\}$$

for some instance. We apply this result with the class $\mathcal{M}_1$, which has $\Delta = \varepsilon_1$ and $A = \varepsilon_1^{-1}$, which gives

$$\mathbb{E}[\mathsf{Reg}] \geq \Omega(1) \cdot \min\left\{\varepsilon_1 T, \frac{1}{\varepsilon_1^2}\right\}.$$

Choosing $\varepsilon_1 = \varepsilon_T = T^{-1/3}$ yields $\mathbb{E}[\mathsf{Reg}] \geq \Omega(T^{2/3})$ whenever $T$ is greater than an absolute constant.

**Class $\mathcal{M}_2$.** Let a parameter $\varepsilon_2 \in [0, 1/2]$ be fixed, and let $A := \varepsilon_2^{-1}$ (we assume without loss of generality that $\varepsilon_2^{-1} \in \mathbb{N}$). We define $\mathcal{M}_2 = \{M^{(1)}, ..., M^{(A)}\}$, where each MDP $M^{(i)}$ is as defined follows:

- We have $H = 2$, and there is a layered state space $\mathcal{X} = \mathcal{X}_1 \times \mathcal{X}_2$, where $\mathcal{X}_1 = \{x_1\}$ and $\mathcal{X}_2 = \{y, z\}$.

- The action space is $\mathcal{A} = \{1, ..., A\}$.

- $x_1$ is the deterministic initial state. Regardless of the action, we transition to $z$ with probability $1 - \varepsilon_2$ and $y$ with probability $\varepsilon_2$.

- For each MDP $M^{(i)}$ all actions have zero reward in states $x_1$ and $z$. For state $y$, action $i$ has reward $1$ and all other actions have reward $0$.

We let $f^{(i)}$ denote the optimal $Q$-function for $M^{(i)}$, which has:

- $f_1^{(i)}(x_1, \cdot) = \varepsilon_2$ and $f_2^{(i)}(z, \cdot) = 0$.

- $f_2^{(i)}(y, a) = \mathbb{1}\{a = i\}$.

We define $\mathcal{F} = \{f^{(i)}\}_{i \in [A]}$; it is clear that this class satisfies completeness. We define $\Pi = \{\pi_f \mid f \in \mathcal{F}\}$; for states where there are multiple optimal actions (i.e., $f_h(x, a) = f_h(x, a')$), we take $\pi_{f,h}(x)$ to be the optimal with the least index, which implies that $\pi_{f,1}(x_1) = \pi_{f,2}(z) = 1$ for all $f \in \mathcal{F}$.

*Verifying coverability.* We choose $\mu_1(x, a) = \mathbb{1}\{x = x_1, a = 1\}$. We choose $\mu_2(z, 1) = \frac{1}{2}$ and $\mu_2(y, a) = \frac{1}{2A}$ for all $a \in \mathcal{A}$. It is immediate that coverability is satisfied with constant $1$ for $h = 1$. For $h = 2$, we have that for all $\pi \in \Pi$,

$$\frac{d_2^\pi(z, 1)}{\mu_2(z, 1)} = \frac{1 - \varepsilon_2}{1/2} \leq 2$$

and

$$\frac{d_2^\pi(y, a)}{\mu_2(y, a)} \leq \frac{\varepsilon_2}{\mu_2(y, a)} \leq 2A\varepsilon_2 \leq 2.$$

Hence, we have $C_{\mathsf{cov}} \leq 2$; note that this holds for any choice of $\varepsilon_2$.

*Lower bounding the Bellman-Eluder dimension.* Let $M^{(A)}$ be the underlying MDP. We will lower bound the Bellman-Eluder dimension for layer $h = 2$. Consider the sequence $d_2^{(1)}, ..., d_2^{(A-1)}$, where $d_2^{(t)} := d_2^{\pi_{f^{(t)}}}$ and $\delta_2^{(1)}, ..., \delta_2^{(A-1)}$, where $\delta_2^{(t)} := f_2^{(t)} - \mathcal{T}_2 f_3^{(t)} = f_2^{(t)} - f_2^{(A)}$ (recall that we adopt the convention $f_{H+1} = 0$). Observe that for each $t \in [A-1]$, we have

$$|\mathbb{E}_{d_2^{(t)}}[\delta_2^{(t)}](x_2, a_2)| = \varepsilon_2 |f_2^{(t)}(y, t) - f_2^{(A)}(y, t)| = \varepsilon_2,$$

yet

$$\sum_{i < t} \mathbb{E}_{d_2^{(i)}}\left[(\delta_2^{(t)}(x_2, a_2))^2\right] = \varepsilon_2 \sum_{i < t}(f_2^{(t)}(y, i) - f_2^{(A)}(y, i))^2 = 0.$$

This certifies that $\dim_{\mathsf{BE}}^{\mathsf{sq}}(\mathcal{F}, \Pi, \varepsilon, 2) \geq A - 1 \geq \varepsilon_2^{-1}/2$ for all $\varepsilon < \varepsilon_2$.

*Upper bound on regret.* To conclude, we set $\varepsilon_2 = \varepsilon_T = 1/T^{-1/3}$. With this choice, we have $\dim_{\mathsf{BE}}^{\mathsf{sq}}(\mathcal{F}_2, \Pi_2, \varepsilon_T) \geq \Omega(\varepsilon_T^{-1})$. Since the construction satisfies completeness (Assumption 1) and has $C_{\mathsf{cov}} \leq 2$ and $H = 2$, Theorem 1 yields

$$\mathsf{Reg} \leq O(\sqrt{T \log(|\mathcal{F}|T/\delta)}) = O(\sqrt{T \log(T/(\varepsilon_2 \delta))}) = \widetilde{O}(\sqrt{T \log(1/\delta)}).$$

$\square$

**Proof of Proposition 18.** Fix $h \in [H]$, and $n \in \mathbb{N}$, and consider sequences $\{d_h^{(1)}, d_h^{(2)}, ..., d_h^{(n)}\}$ and $\{\delta_h^{(1)}, \delta_h^{(2)}, ..., \delta_h^{(n)}\}$ that satisfy Eq. (11) (that is, the sequences witness the value of $\dim_{\mathsf{BE}}(\mathcal{F}, \Pi, \varepsilon, h)$). Then

$$\dim_{\mathsf{BE}}^{\mathsf{sq}}(\mathcal{F}, \Pi, \varepsilon, h) \leq \sum_{t=1}^n \frac{\mathbb{E}_{d_h^{(t)}}\left[\delta_h^{(t)}\right]^2}{\left(\varepsilon^{(t)}\right)^2}$$

$$\leq \sum_{t=1}^{n} \left(1+(\varepsilon^{(t)})^2\right) \cdot \frac{\mathbb{E}_{d_h^{(t)}}\left[\delta_h^{(t)}\right]^2}{(\varepsilon^{(t)})^2\left(1+\sum_{i=1}^{t-1}\mathbb{E}_{d_h^{(i)}}[\delta_h^{(t)}]^2\right)}$$

$$\text{(by } \textstyle\sum_{i=1}^{t-1}\mathbb{E}_{d_h^{(i)}}[\delta_h^{(t)}]^2 \leq (\varepsilon^{(t)})^2)$$

$$\leq \sum_{t=1}^{n} \frac{1+(\varepsilon^{(t)})^2}{(\varepsilon^{(t)})^2} \frac{\mathbb{E}_{d_h^{(t)}}\left[\delta_h^{(t)}\right]^2}{1+\sum_{i=1}^{t-1}\mathbb{E}_{d_h^{(i)}}[\delta_h^{(t)}]^2}$$

$$\leq \sum_{t=1}^{n} \frac{2}{(\varepsilon^{(t)})^2} \frac{\mathbb{E}_{d_h^{(t)}}\left[\delta_h^{(t)}\right]^2}{1+\sum_{i=1}^{t-1}\mathbb{E}_{d_h^{(i)}}[\delta_h^{(t)}]^2} \qquad \text{(by } \varepsilon^{(t)} < |\mathbb{E}_{d_h^{(t)}}[\delta_h^{(t)}]| \leq 1)$$

$$\leq \frac{1}{\varepsilon^2} \sum_{t=1}^{n} \frac{\mathbb{E}_{d_h^{(t)}}\left[\delta_h^{(t)}\right]^2}{1 \vee \sum_{i=1}^{t-1}\mathbb{E}_{d_h^{(i)}}[\delta_h^{(t)}]^2} \qquad \text{(by } \varepsilon^{(t)} \geq \varepsilon)$$

$$\leq \frac{\mathsf{SEC}_{\mathsf{RL}}(\mathcal{F},\Pi,n)}{\varepsilon^2}.$$

This implies for any $T > 0$,

$$\min\{\mathsf{dim}_{\mathsf{BE}}^{\mathsf{sq}}(\mathcal{F},\Pi,\varepsilon),T\} \leq \frac{\mathsf{SEC}_{\mathsf{RL}}(\mathcal{F},\Pi,T)}{\varepsilon^2}.$$

$\square$

### F.2 Proofs from Section 5

**Proof of Proposition 4.** We present a counterexample for both $Q$-type and $V$-type Bellman-Eluder dimension. We recall that the $V$-type Bellman-Eluder dimension is defined by replacing $\mathcal{F}_h - \mathcal{T}_h\mathcal{F}_{h+1}$ with $V_{\mathcal{F}_h-\mathcal{T}_h\mathcal{F}_{h+1}}$ and $\mathfrak{D}_h^{\Pi}$ with $\mathfrak{D}_{h,x}^{\Pi}$ in Definition 6, where $V_{\mathcal{F}_h-\mathcal{T}_h\mathcal{F}_{h+1}} := \{(f_h - \mathcal{T}_h f_{h+1})(\cdot,\pi_{f,h}) : f \in \mathcal{F}\} \subset (\mathcal{X} \to \mathbb{R})$ and $\mathfrak{D}_{h,x}^{\Pi} := \{d_h^{\pi}(\cdot) : \pi \in \Pi\} \subset \Delta(\mathcal{X})$; see Appendix F.3.1 or Jin et al. (2021a) for more background on $V$-type Bellman-Eluder dimension.

**$V$-type Bellman-Eluder dimension.** The hard instance for $V$-type Bellman-Eluder dimension is based on the construction of Efroni et al. (2022a, Proposition B.1), which shows that for any $d = 2^i$ ($i \in \mathbb{N}$), there exists an exogenous MDP (ExoMDP) with $|\mathcal{S}| = 3$ endogenous states, $|\mathcal{A}| = 2$, $H = 2$, and $d$ exogenous factors, with the following properties:[9]

1. There exists a function class $\mathcal{F}$ such that $Q^\star \in \mathcal{F}$ and $|\mathcal{F}| = d$. In addition for all $f \in \mathcal{F}$ with $f \neq Q^\star$, $\pi_f$ is $1/8$-suboptimal.

2. For all $f, f' \in \mathcal{F} \setminus Q^\star$, we have (note that $H = 2$)

$$\mathbb{E}_{x \sim d_2^{\pi_{f'}}, a \sim \pi_{f,2}}[f_2(x,a) - R_2(x,a)] = \frac{1}{2}\mathbb{1}\{f = f'\}. \tag{12}$$

3. $C_{\mathsf{cov}} \leq 6$; this is a consequence of Proposition 8 and the fact that the ExoMDP model in Efroni et al. (2022a) is a special case of the Ex-BMDP model in Section 3.2.

This means that if we take $\{f^{(1)}, f^{(2)}, ..., f^{(d-1)}\}$ to be any ordering of the set of functions in $\mathcal{F} \setminus \{Q^\star\}$, then set $\delta_h^{(i)} := f_h^{(i)} - \mathcal{T}_h f_{h+1}^{(i)}$ and $d_h^{(t)} := d_h^{\pi_{f^{(t)}}}$, we have that for all $t \in [d-1]$,

$$\left|\mathbb{E}_{x \sim d_2^{(t)}, a \sim \pi_{f^{(t)},2}}[\delta_2^{(t)}]\right| = \frac{1}{2}, \quad \text{and} \quad \sqrt{\sum_{i=1}^{t-1}\left(\mathbb{E}_{x \sim d_2^{(i)}, a \sim \pi_{f^{(t)},2}}[\delta_2^{(t)}]\right)^2} = 0.$$

This implies that the $V$-type Bellman-Eluder dimension $\mathsf{dim}_{\mathsf{BE-v}}(\mathcal{F},\Pi_{\mathcal{F}},\varepsilon)$ is at least $d-1$ for all $\varepsilon \leq 1/2$. It is straightforward to verify that this construction in Efroni et al. (2022a) satisfies Assumption 1 (completeness), because functions in the class have $f_1 = \mathcal{T}_2 f_2$ (that is, zero Bellman error at $h=1$). As a result, since $H = 2$, completeness for this construction is implied by $Q^\star \in \mathcal{F}$.

---

[9]Technically, the construction in Efroni et al. (2022a) has a stochastic initial state with known distribution. This can be embedded in our framework, which has a deterministic initial state, by lifting the horizon from 2 to 3.

$Q$**-type Bellman-Eluder dimension.** The construction above immediately extends to $Q$-type. This is because in the construction, the value of $R_2(x,\cdot)$ and $f_2(x,\cdot)$ depends only on $x$ (i.e., is independent of the action) for all $f \in \mathcal{F}$ (cf. Efroni et al., 2022a, Proposition B.1). Therefore, for any $f,g \in \mathcal{F}$, we have,

$$\mathbb{E}_{x\sim d_2^{\pi_f}, a\sim \pi_{g,2}}[g_2(x,a) - R_2(x,a)] = \mathbb{E}_{(x,a)\sim d_2^{\pi_f}}[g_2(x,a) - R_2(x,a)]. \tag{13}$$

This implies that the $Q$-type Bellman residual matrix

$$\left\{ \mathbb{E}_{(x,a)\sim d_2^{\pi_{f'}}}[f_2(x,a) - R_2(x,a)] \right\}_{f,f'\in\mathcal{F}\setminus\{Q^\star\}}$$

embeds the scaled identity matrix and, via the same argument as for $V$-type above, immediately implies that $\dim_{\mathsf{BE}}(\mathcal{F},\Pi,\varepsilon) \geq d-1$ for all $\varepsilon \leq 1/2$. As before, we have $C_{\mathsf{cov}} \leq 6$, and $\mathcal{F}$ is complete. $\qquad\square$

**Proposition 19.** *For any $d\in\mathbb{N}$, there exists an MDP $M$ with $H=2$ and $|\mathcal{A}|=2$, a policy class $\Pi$ with $|\Pi|=d$, and a value function class $\mathcal{F}$ with $|\mathcal{F}|=d$ satisfying completeness, such that $C_{\mathsf{cov}}=O(1)$, yet* OLIVE *(Jiang et al., 2017) requires at least $\Omega(d)$ trajectories to return a $0.1$-optimal policy.*

**Proof of Proposition 19.** We now show that that OLIVE, a canonical average-Bellman-error-based hypothesis elimination algorithm, also suffers from the lower bound in the construction from Proposition 4. By Eq. (12) (V-type OLIVE) and Eq. (13) (Q-type OLIVE), we know that any sub-optimal hypothesis $f\in\mathcal{F}\setminus Q^\star$ cannot be eliminated until $\pi_f$ is executed. On the other hand, the construction ensures $\mathbb{E}[\max_a f(s_1,a)]=7/8$ whereas $J(\pi^\star)=3/4$. This means OLIVE will enumerate over $\mathcal{F}\setminus Q^\star$ before finding a $0.1$-optimal policy for this instance, and hence suffers from complexity of $\Omega(d)$ ($|\mathcal{F}|=d$). $\qquad\square$

**Proof of Theorem 5.** As in Theorem 1, as a consequence of completeness (Assumption 1), the construction of $\mathcal{F}^{(t)}$, and Lemma 12, we have that with probability at least $1-\delta$, for all $t\in[T]$:

$$\text{(i) } Q^\star \in \mathcal{F}^{(t)}, \quad \text{and} \quad \text{(ii) } \sum_{x,a}\widetilde{d}_h^{(t)}(x,a)\big(\delta_h^{(t)}(x,a)\big)^2 \leq O(\beta),$$

and whenever this event holds,

$$\mathsf{Reg} \leq \sum_{t=1}^{T}\Big(f_1^{(t)}(x_1,\pi_{f_1^{(t)},1}(x_1)) - J(\pi^{(t)})\Big) = \sum_{t=1}^{T}\sum_{h=1}^{H}\mathbb{E}_{(x,a)\sim d_h^{(t)}}\big[\underbrace{f_h^{(t)}(x,a) - (\mathcal{T}_h f_{h+1}^{(t)})(x,a)}_{=:\delta_h^{(t)}(x,a)}\big].$$

To proceed, we have that for all $h\in[H]$,

$$\sum_{t=1}^{T}\mathbb{E}_{d_h^{(t)}}\big[\delta_h^{(t)}\big] = \sum_{t=1}^{T}\Big(\mathbb{E}_{d_h^{(t)}}\big[\delta_h^{(t)}\big]\Big)\left(\frac{1\vee\sum_{i=1}^{t-1}\mathbb{E}_{d_h^{(i)}}\big[(\delta_h^{(t)})^2\big]}{1\vee\sum_{i=1}^{t-1}\mathbb{E}_{d_h^{(i)}}\big[(\delta_h^{(t)})^2\big]}\right)^{1/2}$$

$$\leq \sqrt{\sum_{t=1}^{T}\frac{\mathbb{E}_{d_h^{(t)}}\big[\delta_h^{(t)}\big]^2}{1\vee\sum_{i=1}^{t-1}\mathbb{E}_{d_h^{(i)}}\big[(\delta_h^{(t)})^2\big]}}\sqrt{\sum_{t=1}^{T}\left(1\vee\sum_{i=1}^{t-1}\mathbb{E}_{d_h^{(i)}}\big[(\delta_h^{(t)})^2\big]\right)}$$

$$\text{(by Cauchy-Schwarz inequality)}$$

$$\leq \sqrt{\sum_{t=1}^{T}\frac{\mathbb{E}_{d_h^{(t)}}\big[\delta_h^{(t)}\big]^2}{1\vee\sum_{i=1}^{t-1}\mathbb{E}_{d_h^{(i)}}\big[(\delta_h^{(t)})^2\big]}}\sqrt{\beta T}$$

$$\leq \sqrt{\mathsf{SEC}_{\mathsf{RL}}(\mathcal{F},\Pi,T)\cdot\beta T}. \qquad\qquad \text{(by Definition 13)}$$

Therefore, we obtain

$$\mathsf{Reg} \leq H\sqrt{\mathsf{SEC}_{\mathsf{RL}}(\mathcal{F},\Pi,T)\cdot\beta T}.$$

Plugging in the choice for $\beta$ completes the proof. $\qquad\square$

**Proof of Proposition 6.** We prove a more general result. Consider a set of distributions $\mathfrak{D} \subset \Delta(\mathcal{Z})$, and a set of test functions $\Psi \subset (\mathcal{Z} \to [0,1])$. We define a generalized form of coverability with respect to $\mathfrak{D}$ by

$$C_{\mathsf{cov}}(\mathfrak{D}) := \inf_{\mu \in \Delta(\mathcal{Z})} \sup_{d \in \mathfrak{D}} \left\| \frac{d}{\mu} \right\|_{\infty}. \tag{14}$$

We will show that, for any $T > 0$,

$$\mathsf{SEC}(\Psi, \mathcal{D}, T) \lesssim C_{\mathsf{cov}}(\mathfrak{D}) \log(T),$$

which is implies Proposition 6.

Going forward, we fix an arbitrary sequence $\{d^{(1)}, d^{(2)}, ..., d^{(T)}\} \subset \mathfrak{D}$ as well as an arbitrary sequence of $\{\psi^{(1)}, \psi^{(2)}, ..., \psi^{(T)}\} \subset \Psi$. Following Eq. (4), we define

$$\mu^{\star} := \operatorname*{argmin}_{\mu \in \Delta(\mathcal{Z})} \sup_{d \in \mathfrak{D}} \left\| \frac{d}{\mu} \right\|_{\infty}. \tag{15}$$

In addition, define $\widetilde{d}^{(t)} = \sum_{i < t} d^{(t)}$.

For each $z \in \mathcal{Z}$, let

$$\tau(z) := \min\left\{ t \;\middle|\; \sum_{i=1}^{t-1} d^{(i)}(z) \geq C_{\mathsf{cov}} \mu^{\star}(z) \right\}. \tag{16}$$

We decompose $\mathbb{E}_{d^{(t)}}[\psi^{(t)}]$ as

$$\mathbb{E}_{d^{(t)}}\left[\psi^{(t)}\right] = \mathbb{E}_{d^{(t)}}\left[\psi^{(t)}(z)\mathbb{1}[t < \tau(z)]\right] + \mathbb{E}_{d^{(t)}}\left[\psi^{(t)}(z)\mathbb{1}[t \geq \tau(z)]\right].$$

Then,

$$\sum_{t=1}^{T} \frac{\mathbb{E}_{d^{(t)}}\left[\psi^{(t)}\right]^2}{1 \vee \sum_{i=1}^{t-1} \mathbb{E}_{d^{(i)}}\left[(\psi^{(t)})^2\right]}$$
$$\lesssim \underbrace{\sum_{t=1}^{T} \frac{\mathbb{E}_{d^{(t)}}\left[\psi^{(t)}(z)\mathbb{1}[t < \tau(z)]\right]^2}{1 \vee \sum_{i=1}^{t-1} \mathbb{E}_{d^{(i)}}\left[(\psi^{(t)})^2\right]}}_{(\mathtt{I})} + \underbrace{\sum_{t=1}^{T} \frac{\mathbb{E}_{d^{(t)}}\left[\psi^{(t)}(z)\mathbb{1}[t \geq \tau(z)]\right]^2}{1 \vee \sum_{i=1}^{t-1} \mathbb{E}_{d^{(i)}}\left[(\psi^{(t)})^2\right]}}_{(\mathtt{II})}, \tag{17}$$

where we use $a \lesssim b$ as shorthand for $a \leq O(b)$.

We first bound the term (I),

$$\begin{aligned}
(\mathtt{I}) &\leq \sum_{t=1}^{T} \mathbb{E}_{d^{(t)}}\left[\psi^{(t)}(z)\mathbb{1}[t < \tau(z)]\right]^2 \\
&\leq \sum_{t=1}^{T} \mathbb{E}_{d^{(t)}}\left[\mathbb{1}[t < \tau(z)]\right]^2 && \text{(by } \psi(\cdot) \in [0,1], \forall \psi \in \Psi) \\
&\leq \sum_{t=1}^{T} \mathbb{E}_{d^{(t)}}\left[\mathbb{1}[t < \tau(z)]\right] && \text{(by } \mathbb{E}_{d^{(t)}}\left[\mathbb{1}[t < \tau(z)]\right] \leq 1) \\
&= \sum_{z \in \mathcal{Z}} \sum_{t=1}^{T} d_h^{(t)}(z)\mathbb{1}[t < \tau(z)] \\
&= \sum_{z \in \mathcal{Z}} \left(\widetilde{d}^{(\tau(z)-1)}(z) + d^{(\tau(z)-1)}(z)\right) \\
&\stackrel{(a)}{\leq} \sum_{z \in \mathcal{Z}} 2C_{\mathsf{cov}}(\mathfrak{D})\mu^{\star}(z) \\
&\leq C_{\mathsf{cov}}(\mathfrak{D}), \tag{18}
\end{aligned}$$

where (a) follows because $\widetilde{d}^{(\tau(z)-1)}(z), d^{(\tau(z)-1)}(z) \leq C_{\mathsf{cov}}(\mathfrak{D})\mu^\star(z)$, for all $z \in \mathcal{Z}$, as a consequence of Eqs. (15) and (16).

We now turn to the term (II). First, observe that

$$\sum_{z \in \mathcal{Z}} \mathbb{1}[t \geq \tau(z)] d^{(t)}(z) \psi^{(t)}(z)$$

$$= \sum_{z \in \mathcal{Z}} \mathbb{1}[t \geq \tau(z)] d^{(t)}(z) \left( \frac{\sum_{i=1}^{t-1} d^{(i)}(z)}{\sum_{i=1}^{t-1} d^{(i)}(z)} \right)^{1/2} \psi^{(t)}(z)$$

$$\leq \sqrt{\sum_{z \in \mathcal{Z}} \frac{\mathbb{1}[t \geq \tau(z)](d^{(t)}(z))^2}{\sum_{i=1}^{t-1} d^{(i)}(z)}} \sqrt{\sum_{i=1}^{t-1} \mathbb{E}_{d^{(i)}} \left[ (\psi^{(t)})^2 \right]}. \qquad \text{(by Cauchy-Schwarz inequality)}$$

By rearranging this inequality, we have

$$\text{(II)} \leq \sum_{t=1}^{T} \sum_{z \in \mathcal{Z}} \frac{\mathbb{1}[t \geq \tau(z)] \left( d_h^{(t)}(z) \right)^2}{\sum_{i=1}^{t-1} d^{(i)}(z)} \qquad \text{(defining } 0/0 = 0\text{)}$$

$$\leq 2 \sum_{t=1}^{T} \sum_{z \in \mathcal{Z}} \frac{\mathbb{1}[t \geq \tau(z)](d^{(t)}(z))^2}{C_{\mathsf{cov}} \cdot \mu^\star(z) + \sum_{i=1}^{t-1} d^{(i)}(z)} \qquad \text{(by Eq. (16))}$$

$$\lesssim \sum_{t=1}^{T} \sum_{z \in \mathcal{Z}} \frac{(d^{(t)}(z))^2}{C_{\mathsf{cov}} \cdot \mu^\star(z) + \sum_{i=1}^{t-1} d^{(i)}(z)}$$

$$\leq \sum_{t=1}^{T} \sum_{z \in \mathcal{Z}} \left( \max_{i \leq T} d^{(i)}(z) \right) \frac{d^{(t)}(z)}{\sum_{i=1}^{t-1} d^{(i)}(z) + C_{\mathsf{cov}} \cdot \mu^\star(z)}$$

$$\leq C_{\mathsf{cov}}(\mathfrak{D}_h) \sum_{z \in \mathcal{Z}} \mu^\star(z) \sum_{t=1}^{T} \frac{d^{(t)}(z)}{\sum_{i=1}^{t-1} d^{(i)}(z) + C_{\mathsf{cov}} \cdot \mu^\star(z)} \qquad \text{(by Eq. (14))}$$

$$\lesssim C_{\mathsf{cov}}(\mathfrak{D}) \sum_{z \in \mathcal{Z}} \mu^\star(z) \log(T) \qquad \text{(by Lemma 15)}$$

$$= C_{\mathsf{cov}}(\mathfrak{D}) \log(T). \qquad (19)$$

Substituting Eqs. (18) and (19) into Eq. (17), we obtain

$$\mathsf{SEC}(\Psi, \mathcal{D}, T) \lesssim C_{\mathsf{cov}}(\mathfrak{D}) \log(T).$$

$\square$

**Proof of Proposition 7.** This proof provides a slightly more general result. Consider a set of distributions $\mathfrak{D} \subset \Delta(\mathcal{Z})$ and a set of test functions $\Psi \subset (\mathcal{Z} \to [0,1])$ be given. We consider an abstract version of the Bellman-Eluder dimension with respect to $\mathfrak{D}$ and $\Psi$. We define $\dim_{\mathsf{BE}}(\Psi, \mathfrak{D}, \varepsilon)$ is the largest $d \in \mathbb{N}$ such that there exist sequences $\{d^{(1)}, d^{(2)}, ..., d^{(d)}\} \subset \mathfrak{D}$ and $\{\psi^{(1)}, \psi^{(2)}, ..., \psi^{(d)}\} \subset \Psi$ such that for all $t \in [d]$, [10]

$$\left| \mathbb{E}_{d^{(t)}}[\psi^{(t)}] \right| > \varepsilon^{(t)}, \quad \text{and} \quad \sqrt{\sum_{i=1}^{t-1} \left( \mathbb{E}_{d^{(i)}}[\psi^{(t)}] \right)^2} \leq \varepsilon^{(t)}, \qquad (20)$$

for $\varepsilon^{(1)}, ..., \varepsilon^{(d)} \geq \varepsilon$. We will show that, for any all $T \in \mathbb{N}$,

$$\mathsf{SEC}(\Psi, \mathcal{D}, T) \lesssim \inf_{\varepsilon > 0} \left\{ \varepsilon^2 T + \dim_{\mathsf{BE}}(\Psi, \mathfrak{D}, \varepsilon) \right\} \cdot \log(T),$$

which immediately implies Proposition 7.

---

[10]This definition coincides with distributional Eluder dimension (see, e.g., Jin et al., 2021a), which only differs from Bellman-Eluder dimension on the notation of test function. We overload the notation for $\dim_{\mathsf{BE}}$ over this proof for simplicity.

**A generalized definition of $\varepsilon$-dependent sequence.** In what follows, we rely on a slightly different notion of an $\varepsilon$-*(in)dependent sequence* from the one given in Jin et al. (2021a, Definition 6) and Russo and Van Roy (2013). We provide background on both definitions below.

**$\varepsilon$-(in)dependent sequence (e.g., Jin et al., 2021a, Definition 6).** A distribution $\nu \in \mathfrak{D}$ is $\varepsilon$-dependent on a sequence $\{\nu^{(1)},...,\nu^{(k)}\} \subseteq \mathfrak{D}$ if: When $|\mathbb{E}_\nu[\psi]| > \varepsilon$ for some $\psi \in \Psi$, we also have $\sum_{i=1}^k (\mathbb{E}_{\nu^{(i)}}[\psi])^2 > \varepsilon^2$. Otherwise, $\nu$ is $\varepsilon$-independent if this does not hold.

**Generalized $\varepsilon$-(in)dependent sequence.** A distribution $\nu \in \mathfrak{D}$ is (generalized) $\varepsilon$-dependent on a sequence $\{\nu^{(1)}, ... , \nu^{(k)}\} \subseteq \mathfrak{D}$ if: for all $\varepsilon' \geq \varepsilon$, if $|\mathbb{E}_\nu[\psi]| > \varepsilon'$ for some $\psi \in \Psi$, we also have $\sum_{i=1}^k (\mathbb{E}_{\nu^{(i)}}[\psi])^2 > \varepsilon'^2$. We say that $\nu$ is (generalized) $\varepsilon$-independent if this does not hold, i.e., for some $\varepsilon' \geq \varepsilon$, it has $|\mathbb{E}_\nu[\psi]| > \varepsilon'$ but $\sum_{i=1}^k (\mathbb{E}_{\nu^{(i)}}[\psi])^2 \leq \varepsilon'^2$.

The generalized definition above naturally induces a new implication (which the original definition may not have): *If $\varepsilon' \geq \varepsilon$, then $\varepsilon$-dependent sequence $\Rightarrow \varepsilon'$-dependent sequence, or in other words, $\varepsilon'$-independent sequence $\Rightarrow \varepsilon$-independent sequence.*

The definition of the *distributional Eluder dimension* (see Eq. (20)) can be written in two equivalent ways using original and generalized definition for a $\varepsilon$-independent sequence: $\dim_{\mathsf{BE}}(\Psi, \mathfrak{D}, \varepsilon)$ is the largest $d \in \mathbb{N}$ such that there exists a sequence $\{d^{(1)}, d^{(2)},...,d^{(d)}\} \subset \mathfrak{D}$ such that for all $t \in [d]$:

(i) $d^{(t)}$ is $\varepsilon'$-independent of $\{d^{(1)}, d^{(2)},...,d^{(t-1)}\}$ for some $\varepsilon' \geq \varepsilon$.

(ii) $d^{(t)}$ is (*generalized*) $\varepsilon$-independent of $\{d^{(1)}, d^{(2)},...,d^{(t-1)}\}$ $\Longleftarrow$[by the implication above]$\Longrightarrow$ $d^{(t)}$ is (*generalized*) $\varepsilon'$-independent of $\{d^{(1)}, d^{(2)},...,d^{(t-1)}\}$ for some $\varepsilon' \geq \varepsilon$.

This indicates that the distributional Eluder dimension can be equivalently written in terms of generalized independent sequences. Going forward, we only use the generalized $\varepsilon$-(in)dependent definition, and omit the word generalized.

**Setup.** Let us use $\dim_{\mathsf{BE}}(\varepsilon)$ as shorthand for $\dim_{\mathsf{BE}}(\Psi, \mathfrak{D}, \varepsilon)$. By Eq. (20), we know $\dim_{\mathsf{BE}}(\varepsilon)$ also upper bounds the length of sequences $\{d^{(1)}, d^{(2)},...,d^{(d)}\} \subset \mathfrak{D}$ and $\{\psi^{(1)}, \psi^{(2)},...,\psi^{(d)}\} \subset \Psi$ such that for all $t \in [d]$,

$$|\mathbb{E}_{d^{(t)}}[\delta^{(t)}]| > \varepsilon^{(t)}, \quad \text{and} \quad \sqrt{\sum_{i=1}^{t-1} \mathbb{E}_{d^{(i)}}[(\psi^{(t)})^2]} \leq \varepsilon^{(t)},$$

for $\varepsilon^{(1)},...,\varepsilon^{(d)} \geq \varepsilon$ (note that the square is inside the expectation which is different from Eq. (20)).

Now, for any $\{d^{(1)}, d^{(2)}, ... , d^{(T)}\} \subset \mathfrak{D}$ and $\{\psi^{(1)}, \psi^{(2)}, ... , \psi^{(T)}\} \subset \Psi$, we define $\beta^{(t)} := \sum_{i=1}^{t-1} \mathbb{E}_{d^{(i)}}[(\psi^{(t)})^2]$. We will study the sequence

$$\left\{ \frac{\mathbb{E}_{d^{(1)}}[\psi^{(1)}]^2}{1 \vee \beta^{(1)}}, \frac{\mathbb{E}_{d^{(2)}}[\psi^{(2)}]^2}{1 \vee \beta^{(3)}},..., \frac{\mathbb{E}_{d^{(T)}}[\psi^{(T)}]^2}{1 \vee \beta^{(T)}} \right\}. \tag{21}$$

Fix a parameter $\alpha > 0$, whose value will be specified later. For the remainder of the proof, we use $L^{(t)}$ to denote the number of disjoint $\alpha\sqrt{1 \vee \beta_h^{(t)}}$-dependent subsequences of $d^{(t)}$ in $\{d^{(1)}, d^{(2)},...,d^{(t-1)}\}$, for each $t \in [T]$.

**Step 1.** Suppose the $t$-th term of Eq. (21) is greater than $\alpha^2$, so that $|\mathbb{E}_{d^{(t)}}[\psi^{(t)}]| > \alpha\sqrt{1 \vee \beta^{(t)}}$. From the definition of $L^{(t)}$, we know there have at least $L^{(t)}$ disjoint subsequences of $\{d^{(1)},...,d^{(t-1)}\}$ (denoted by $\mathfrak{S}^{(1)},...,\mathfrak{S}^{(L^{(t)})}$), such that

$$\sum_{i=1}^{L^{(t)}} \sum_{\nu \in \mathfrak{S}^{(i)}} (\mathbb{E}_\nu[\psi^{(t)}])^2 \geq (1 \vee \beta^{(t)})\alpha^2. \tag{22}$$

On the other hand, by the definition of $\beta_h^{(t)}$, we have

$$\sum_{i=1}^{L^{(t)}} \sum_{\nu \in \mathfrak{S}^{(i)}} (\mathbb{E}_\nu[\psi^{(t)}])^2 \leq \sum_{i=1}^{t-1} \mathbb{E}_{d^{(i)}}[(\psi^{(t)})^2] \leq \beta^{(t)}. \tag{23}$$

Therefore, combining Eqs. (22) and (23) we obtain that, if $|\mathbb{E}_{d^{(t)}}[\psi^{(t)}]| > \alpha\sqrt{1 \vee \beta^{(t)}}$ for some $t \in [T]$,

$$\beta^{(t)} \geq L^{(t)}(1 \vee \beta^{(t)})\alpha^2 \Longrightarrow L^{(t)} \leq \frac{1}{\alpha^2}. \tag{24}$$

**Step 2.** On the other hand, let $\{i_1, i_2, ..., i_\kappa\}$ be the longest subsequence of $[T]$, where

$$\frac{\mathbb{E}_{d^{(i_j)}}[\delta^{(i_j)}]^2}{1 \vee \beta^{(i_j)}} > \alpha^2, \forall j \in [\kappa].$$

For compactness, we use $\{\nu^{(1)}, \nu^{(2)}, ..., \nu^{(\kappa)}\}$ abbreviate $\{d^{(i_1)}, d^{(i_2)}, ..., d^{(i_\kappa)}\}$. We now argue that there exists $j^\star \in [\kappa]$, such that for $\nu^{(j^\star)}$, there must exist at least

$$\underline{L}^\star \geq \left\lfloor \frac{\kappa}{\mathsf{dim}_{\mathsf{BE}}(\alpha) + 1} \right\rfloor \geq \frac{\kappa}{\mathsf{dim}_{\mathsf{BE}}(\alpha) + 1} - 1 \tag{25}$$

$\alpha$-dependent disjoint subsequences in $\{\nu^{(1)}, \nu^{(2)}, ..., \nu^{(j^\star - 1)}\}$ (the actual number of disjoint subsequences is denoted by $\underline{L}^\star$). This is because we can construct such disjoint subsequences by the following procedure:

$\langle 1 \rangle$ For $j \in [\underline{L}^\star]$, $\mathfrak{S}^{(j)} \leftarrow \{\nu^{(j)}\}$. Then, set $j \leftarrow \underline{L}^\star + 1$.

$\langle 2 \rangle$ If $\nu^{(j)}$ is $\alpha$-dependent on $\mathfrak{S}^{(1)}, ..., \mathfrak{S}^{(\underline{L}^\star)}$, terminate the procedure (goal achieved).

$\langle 3 \rangle$ Otherwise, we know $\nu^{(j)}$ is $\alpha$-independent on at least one of $\mathfrak{S}^{(1)}, ..., \mathfrak{S}^{(\underline{L}^\star)}$ (denoted by $\mathfrak{S}^\star$). Update $\mathfrak{S}^\star \leftarrow \mathfrak{S}^\star \bigcup \{\nu^{(j)}\}$, $j \leftarrow j + 1$, and go to $\langle 2 \rangle$.

From the definition of $\mathsf{dim}_{\mathsf{BE}}(\alpha)$, we know if $|\mathfrak{S}^{(i)}| \geq \mathsf{dim}_{\mathsf{BE}}(\alpha) + 1$, any $\nu \in \mathfrak{D}_h$ must be $\alpha$-dependent on $\mathfrak{S}^{(i)}$ (for each $i \in [\underline{L}^\star]$). Therefore, such a procedure must terminate before or on $j^{(\max)} = \underline{L}^\star \mathsf{dim}_{\mathsf{BE}}(\alpha) + \underline{L}^\star$. Thus, if $j^{(\max)} \leq \kappa$, termination in $\langle 2 \rangle$ must happen. This only requires $\underline{L}^\star$ to satisfy

$$\underline{L}^\star \mathsf{dim}_{\mathsf{BE}}(\alpha) + \underline{L}^\star \leq \kappa \quad \implies \quad \underline{L}^\star \leq \frac{\kappa}{\mathsf{dim}_{\mathsf{BE}}(\alpha) + 1}.$$

That is, as long as $\underline{L}^\star \leq \left\lfloor \frac{\kappa}{\mathsf{dim}_{\mathsf{BE}}(\alpha) + 1} \right\rfloor$, the termination in $\langle 2 \rangle$ must happen for some $j^\star \leq \kappa$.

**Step 3.** As we discussed at the beginning, $\alpha$-dependence implies $\alpha'$-dependence for all $\alpha' \geq \alpha$. This means the $\underline{L}^\star$ in Step 2 lower bounds $\max_{t \in [T]} L^{(t)}$ in Step 1, because $\{d^{(i_1)}, d^{(i_2)}, ..., d^{(i_\kappa)}\}$ is a subset of $\{d^{(1)}, d^{(2)}, ..., d^{(i_\kappa)}\}$. Thus, combining Eqs. (24) and (25), we can obtain that,

$$\frac{1}{\alpha^2} \geq \max_{t \in [T]} L^{(t)} \geq \underline{L}^\star \geq \frac{\kappa}{\mathsf{dim}_{\mathsf{BE}}(\alpha) + 1} - 1.$$

This implies that

$$\kappa \leq \left(1 + \frac{1}{\alpha^2}\right)(\mathsf{dim}_{\mathsf{BE}}(\alpha) + 1) \leq \frac{3\mathsf{dim}_{\mathsf{BE}}(\alpha)}{\alpha^2} + 1. \qquad (\text{suppose } \alpha \leq 1)$$

As a consequence, for any $\varepsilon \in (0, 1]$, by setting $\alpha = \sqrt{\varepsilon}$,

$$\sum_{t=1}^{T} \mathbb{1}\left(\frac{\mathbb{E}_{d^{(t)}}[\psi^{(t)}]^2}{1 \vee \beta^{(t)}} > \varepsilon\right) \leq \frac{3\mathsf{dim}_{\mathsf{BE}}(\sqrt{\varepsilon})}{\varepsilon} + 1. \tag{26}$$

**Step 4.** Let $e^{(1)} \geq e^{(2)} \geq \cdots \geq e^{(T)}$ denote the sequence in Eq. (21) reordered in a decreasing fashion. For any parameter $w \in (0, 1]$ to be specified later, we have

$$\sum_{t=1}^{T} \frac{\mathbb{E}_{d^{(t)}}[\psi^{(t)}]^2}{1 \vee \sum_{i=1}^{t-1} \mathbb{E}_{d^{(i)}}[(\psi^{(t)})^2]} = \sum_{t=1}^{T} e^{(t)}$$

$$\leq Tw + \sum_{t=1}^{T} e^{(t)} \mathbb{1}(e^{(t)} > w).$$

Observe that for any $t \in [T]$ such that $e^{(t)} > w$, if $2\eta \geq e^{(t)} > \eta \geq w$, we have

$$t \leq \sum_{i=1}^{T} \mathbb{1}(e_i > \eta)$$

$$\leq \frac{3}{\eta}\mathsf{dim_{BE}}(\sqrt{\eta})+1 \qquad\qquad \text{(by Eq. (26))}$$

$$\leq \frac{3}{\eta}\mathsf{dim_{BE}}(\sqrt{w})+1$$

$$\implies \eta \leq \frac{3d}{t-1} \qquad\qquad (\text{define } d := \mathsf{dim_{BE}}(\sqrt{w}))$$

$$\implies e^{(t)} \leq \min\left(\frac{6d}{t-1}, 1\right). \qquad\qquad (2\eta \geq e^{(t)} > \eta)$$

Therefore,

$$\sum_{t=1}^{T} e^{(t)}\mathbb{1}(e^{(t)} > w) \leq d + \sum_{t=d+1}^{T} \frac{6d}{t-1}$$

$$\leq d + 6d\log(T).$$

$$\implies \sum_{t=1}^{T} e^{(t)} \leq Tw + \mathsf{dim_{BE}}(\sqrt{w}) + 6\mathsf{dim_{BE}}(\sqrt{w})\log(T).$$

Selecting $w = \varepsilon^2$ implies

$$\mathsf{SEC}(T) \lesssim \inf_{\varepsilon > 0}\left\{\varepsilon^2 T + \mathsf{dim_{BE}}(\varepsilon)\right\}\cdot\log(T).$$

This completes the proof. $\qquad\qquad\qquad\qquad\qquad\qquad\qquad\qquad\qquad\qquad\qquad$ □

## F.3 DISCUSSION: RELATIONSHIP TO ADDITIONAL COMPLEXITY MEASURES

### F.3.1 SEQUENTIAL EXTRAPOLATION COEFFICIENT: $Q$-TYPE VERSUS $V$-TYPE

The Sequential Extrapolation Coefficient, as defined in Definition 8), can be thought of as a generalization of $Q$-type Bellman-Eluder dimension (Jin et al., 2021a). In this section we sketch how one can adapt Sequential Extrapolation Coefficient so as to generalize $V$-type Bellman-Eluder dimension instead. Note that $V$-type Bellman-Eluder dimension subsumes the original notion of Bellman rank from Jiang et al. (2017).

We define the $V$-type Sequential Extrapolation Coefficient for RL as follows.

**Definition 10** (Sequential Extrapolation Coefficient for RL, $V$-type). *For each $h \in [H]$, let $\mathfrak{D}_{h,x}^{\Pi} := \{d_h^\pi(\cdot) : \pi \in \Pi\} \subset \Delta(\mathcal{X})$ and $V_{\mathcal{F}_h - \mathcal{T}_h \mathcal{F}_{h+1}} := \{(f_h - \mathcal{T}_h f_{h+1})(\cdot, \pi_{f,h}) : f \in \mathcal{F}\} \subset (\mathcal{X} \to \mathbb{R})$. Then we define,*

$$\mathsf{SEC_{RL\text{-}v}}(\mathcal{F},\Pi,T) := \max_{h \in [H]} \mathsf{SEC}(V_{\mathcal{F}_h - \mathcal{T}_h \mathcal{F}_{h+1}}, \mathfrak{D}_{h,x}^{\Pi}, T).$$

We recall that the $V$-type Bellman-Eluder dimension $\mathsf{dim_{BE\text{-}v}}(\mathcal{F},\Pi,\varepsilon)$ is defined analogously, by replacing $\mathcal{F}_h - \mathcal{T}_h \mathcal{F}_{h+1} \to V_{\mathcal{F}_h - \mathcal{T}_h \mathcal{F}_{h+1}}$ and $\mathfrak{D}_h^{\Pi} \to \mathfrak{D}_{h,x}^{\Pi}$ in Definition 6.

Lastly, we give a $V$-type generalization of Definition 2 (i.e., coverability w.r.t. state only), for a policy class $\Pi$ as follows:

$$C_{\mathsf{cov\text{-}v}} := \inf_{\mu_1,\dots,\mu_H \in \Delta(\mathcal{X})} \sup_{\pi \in \Pi, h \in [H]} \left\|\frac{d_h^\pi}{\mu_h}\right\|_\infty. \qquad (27)$$

As a simple implication, we have $C_{\mathsf{cov\text{-}v}} \leq C_{\mathsf{cov}} \leq C_{\mathsf{cov\text{-}v}} \cdot |\mathcal{A}|$.

Note that the $V$-type variants of sequential extrapolation coefficient, Bellman-Eluder dimension, and coverability differ from their $Q$-type counterparts only in the choices for the distribution and test function sets. Since our proofs for Propositions 6 and 7 hold for arbitrary distributions and test function sets, we immediately obtain the following $V$-type extensions of Propositions 6 and 7.

**Proposition 20** (Coverability $\implies$ SEC, $V$-type). *Let $C_{\mathsf{cov\text{-}v}}$ be the $V$-type coverability coefficient (Eq. (27)) with policy class $\Pi$. Then for any value function class $\mathcal{F}$, $\mathsf{SEC_{RL\text{-}v}}(\mathcal{F},\Pi,T) \leq O(C_{\mathsf{cov\text{-}v}} \cdot \log(T)).$*

**Proposition 21** (Bellman-Eluder dimension $\implies$ SEC, $V$-type). *Suppose* $\dim_{\mathsf{BE\text{-}v}}(\mathcal{F},\Pi,\varepsilon)$ *be the* $V$-*type Bellman-Eluder dimension with function class* $\mathcal{F}$ *and policy* $\Pi$, *then*

$$\mathsf{SEC}_{\mathsf{RL\text{-}v}}(\mathcal{F},\Pi,T) \leq O\left(\inf_{\varepsilon>0}\left\{\varepsilon^2 T + \dim_{\mathsf{BE\text{-}v}}(\mathcal{F},\Pi,\varepsilon)\right\}\cdot\log(T)\right).$$

As shown in Jin et al. (2021a), GOLF (Algorithm 1) can be extended to $V$-type by simply replacing Line 3 in Algorithm 1 with sampling $(s_h,a_h,r_h,s_{h+1})\sim d_h^{(t)}\times\pi_{\mathsf{unif}}$ ($s_h\sim d_h^{(t)}$ and $a_h\sim\mathsf{unif}(\mathcal{A})$) each $h\in[H]$. By slightly modifying the proof of Theorem 5 one can obtain similar sample complexity guarantees based on the $V$-type Sequential Extrapolation Coefficient. We omit the details here, since the only differences are 1) a $V$-type analog of Lemma 12 (provided by Jin et al., 2021a, Lemma 44); and 2) trivially upper bounding the quantity $\mathbb{E}_{d_h^{(i)}\times\pi_h^{(t)}}[(\delta_h^{(t)})^2]$ (used in $\mathsf{SEC}_{\mathsf{RL\text{-}v}}$) by $|\mathcal{A}|\cdot\mathbb{E}_{d_h^{(i)}\times\pi_{\mathsf{unif}}}[(\delta_h^{(t)})^2]$ (controlled by in-sample error). Note, however, that due to the uniform exploration, this algorithm leads to a sample complexity guarantee of the form

$$J(\pi^\star)-J(\bar{\pi})\leq O\left(H\sqrt{\frac{\mathsf{SEC}_{\mathsf{RL\text{-}v}}(\mathcal{F},\Pi,T)|\mathcal{A}|\log(TH|\mathcal{F}|/\delta)}{T}}\right),$$

but not a regret bound.

### F.3.2 CONNECTION TO BILINEAR CLASSES

The Bilinear class framework (Du et al., 2021) generalizes the notion of Bellman rank (Jiang et al., 2017), which captures various more structural conditions via an additional class of *discrepancy functions*. In this section we sketch how one can generalize the sequential extrapolation coefficient (SEC) further by allowing for the use of general discrepancy functions to form confidence sets and estimate Bellman residuals, in the vein of Bilinear classes.

**Definition 11** (Gen-SEC). *Let* $\mathcal{Z}$ *be an abstract set. Let* $\Psi\subset(\mathcal{Z}\to\mathbb{R})$ *be a* function class, *and let* $\mathfrak{D}_\Psi(:=\{d_\psi:\psi\in\Psi\}),\mathfrak{P}_\Psi(:=\{p_\psi:\psi\in\Psi\})\subset\Delta(\mathcal{Z})$ *be two corresponding distribution classes, and* $\mathfrak{L}_\Psi(:=\{\ell_\psi:\psi\in\Psi\})\subset(\mathcal{Z}\to\mathbb{R})$ *be a corresponding discrepancy function class. The* Gen-SEC *for length* $T$ *is given by*

$$\mathsf{SEC}^{\mathsf{gen}}(\Psi,\mathfrak{D}_\Psi,\mathfrak{P}_\Psi,\mathfrak{L}_\Psi,T):=\sup_{\psi^{(1)},\ldots,\psi^{(T)}\in\Psi}\left\{\sum_{t=1}^T\frac{\mathbb{E}_{d_{\psi^{(t)}}}[\psi^{(t)}]^2}{1\vee\sum_{i=1}^{t-1}\mathbb{E}_{p_{\psi^{(i)}}}[\ell_{\psi^{(t)}}^2]}\right\}. \tag{28}$$

To apply the generalized SEC to reinforcement learning, one can set (for each level $h$) $\Psi=\mathcal{F}_h-\mathcal{T}_h\mathcal{F}_{h+1}$, $\mathfrak{D}_\Psi=\{d_h^\pi(\cdot,\cdot):\pi\in\Pi_\mathcal{F}\}$, and $\mathfrak{P}_\Psi=\{(d_h^\pi\times\pi_{\mathsf{est},\psi_h})(\cdot,\cdot):\pi\in\Pi_\mathcal{F}\}$, where $(d\times\pi)(x,a):=d(x)\pi(a\,|\,x)$ (for any $d\in\Delta(\mathcal{X})$, $\pi\in(\mathcal{X}\to\Delta(\mathcal{A}))$ and $(x,a)\in\mathcal{X}\times\mathcal{A}$), and $\pi_{\mathsf{est},\psi_h}$ denotes the estimation policy depending on $\psi_h$ (e.g., greedy policy w.r.t. $\psi_h$ or uniformly random policy over $\mathcal{A}$). The discrepancy function class $\mathfrak{L}_\Psi$ can be selected according to the original Bilinear rank for covering various structural conditions, and setting $\mathfrak{L}_\Psi=\Psi$ recovers the original SEC.

By combining GOLF and Theorem 5 with the approach from Du et al. (2021), one can provide sample complexity guarantees that scale with the Gen-SEC. We omit the details, but the basic idea is to form the confidence set using the discrepancy function class $\mathfrak{L}_\Psi$ rather than working with squared Bellman error.

**Bounding the generalized** SEC **by bilinear rank.** In what follows, we show that the abstract version of the Generalized SEC in Eq. (28) can be bounded by an abstract generalization of the notion of Bilinear rank from Du et al. (2021).

**Definition 12** (Bilinear rank, finite dimension (Du et al., 2021)). *Let* $\mathcal{Z}$ *be an abstract set. Let* $\Psi\subset(\mathcal{Z}\to\mathbb{R})$ *be a* function class, *and let* $\mathfrak{D}_\Psi(:=\{d_\psi:\psi\in\Psi\}),\mathfrak{P}_\Psi(:=\{p_\psi:\psi\in\Psi\})\subset\Delta(\mathcal{Z})$ *be two corresponding distribution classes, and* $\mathfrak{L}_\Psi(:=\{\ell_\psi:\psi\in\Psi\})\subset(\mathcal{Z}\to\mathbb{R})$ *be a corresponding discrepancy function class. The class* $\Psi$ *is said to have Bilinear rank* $d$ *if there exists* $\psi^\star\in\Psi$ *and functions* $X,W\subset(\Psi\to\mathbb{R}^d)$ *such that 1)* $\sum_{\psi\in\Psi}\|X(\psi)\|_2\leq 1$ *and* $\sum_{\psi\in\Psi}\|W(\psi)\|_2\leq B_W$, *and 2)*

$$\mathbb{E}_{d_\psi}[\psi]\leq|\langle W(\psi)-W(\psi^\star),X(\psi)\rangle|\quad\forall\psi\in\Psi,$$
$$\mathbb{E}_{p_\psi}[\ell_{\psi'}]=|\langle W(\psi')-W(\psi^\star),X(\psi)\rangle|\quad\forall\psi,\psi'\in\Psi.$$

*We define* $\dim_{\mathsf{bi}}(\Psi,\mathfrak{D}_\Psi,\mathfrak{P}_\Psi,\mathfrak{L}_\Psi)$ *as the least dimension* $d$ *for which this property holds.*

**Proposition 22** (Bilinear rank $\implies$ Gen-SEC). *Suppose* $\mathsf{SEC}^{\mathsf{gen}}(\Psi, \mathfrak{D}_\Psi, \mathfrak{P}_\Psi, \mathfrak{L}_\Psi, T)$ *and* $\dim_{\mathsf{bi}}(\Psi, \mathfrak{D}_\Psi, \mathfrak{P}_\Psi, \mathfrak{L}_\Psi)$ *be the gen-SEC and Bilinear rank defined in Definitions 11 and 12 with respect to function class $\Psi$, distribution classes $\mathfrak{D}_\Psi$ and $\mathfrak{D}_\Psi$, and discrepancy function class $\mathfrak{L}_\Psi$. Then we have,*

$$\mathsf{SEC}^{\mathsf{gen}}(\Psi, \mathfrak{D}_\Psi, \mathfrak{P}_\Psi, \mathfrak{L}_\Psi, T) \lesssim \dim_{\mathsf{bi}}(\Psi, \mathfrak{D}_\Psi, \mathfrak{P}_\Psi, \mathfrak{L}_\Psi) \log\left(1 + \frac{4B_W^2 T}{d}\right).$$

**Proof of Proposition 22.** Throughout the proof, we use $d^{(t)}$, $p^{(t)}$ and $\ell^{(t)}$ as the shorthands of $d_{\psi^{(t)}}$, $p_{\psi^{(t)}}$ and $\ell_{\psi^{(t)}}$. We study the quantity,

$$\sum_{t=1}^T \frac{\mathbb{E}_{d^{(t)}}[\psi^{(t)}]^2}{1 \vee \sum_{i=1}^{t-1}\mathbb{E}_{p^{(i)}}[(\ell^{(t)})^2]} \leq 2\sum_{t=1}^T \frac{\mathbb{E}_{d^{(t)}}[\psi^{(t)}]^2}{1 + \sum_{i=1}^{t-1}\mathbb{E}_{p^{(i)}}[\ell^{(t)}]^2}.$$

By Definition 12, we have

$$\mathbb{E}_{d^{(t)}}[\psi^{(t)}]^2 \leq |\langle W(\psi^{(t)}) - W(\psi^\star), X(\psi^{(t)})\rangle|^2,$$

and

$$\begin{aligned}
1 + \sum_{i=1}^{t-1}\mathbb{E}_{p^{(i)}}[\ell^{(t)}]^2 &= 1 + \sum_{i=1}^{t-1}|\langle W(\psi^{(t)}) - W(\psi^\star), X(\psi^{(i)})\rangle|^2 \\
&\geq (W(\psi^{(t)}) - W(\psi^\star))^\top \Sigma_t (W(\psi^{(t)}) - W(\psi^\star)) \\
&= \|W(\psi^{(t)}) - W(\psi^\star)\|_{\Sigma_t}^2,
\end{aligned}$$

where $\Sigma_t := \frac{1}{4B_W^2}\mathbf{I} + \sum_{i=1}^{t-1} X(\psi^{(i)})X(\psi^{(i)})^\top$.

We bound

$$\begin{aligned}
\mathbb{E}_{d^{(t)}}[\psi^{(t)}]^2 &\leq |\langle W(\psi^{(t)}) - W(\psi^\star), X(\psi^{(t)})\rangle|^2 \\
&\leq \|W(\psi^{(t)}) - W(\psi^\star)\|_{\Sigma_t}^2 \cdot \|X(\psi^{(t)})\|_{\Sigma_t^{-1}}^2,
\end{aligned}$$

which implies

$$\begin{aligned}
\sum_{t=1}^T \frac{\mathbb{E}_{d^{(t)}}[\psi^{(t)}]^2}{1 + \sum_{i=1}^{t-1}\mathbb{E}_{p^{(i)}}[\ell^{(t)}]^2} &\leq \sum_{t=1}^T 1 \wedge \|X(\psi^{(t)})\|_{\Sigma_t^{-1}}^2 \\
&\leq 2\log\left(\frac{\det(\Sigma_T)}{\det(\Sigma_1)}\right) \\
&\leq 2\dim_{\mathsf{bi}}(\Psi, \mathfrak{D}_\Psi, \mathfrak{P}_\Psi, \mathfrak{L}_\Psi) \log\left(1 + \frac{4B_W^2 T}{d}\right),
\end{aligned}$$

where the last two inequalities follow from the elliptical potential lemma (Lattimore and Szepesvári, 2020, Lemma 19.4). Putting everything together, we obtain

$$\mathsf{SEC}^{\mathsf{gen}}(\Psi, \mathfrak{D}_\Psi, \mathfrak{P}_\Psi, \mathfrak{L}_\Psi, T) \leq 4\dim_{\mathsf{bi}}(\Psi, \mathfrak{D}_\Psi, \mathfrak{P}_\Psi, \mathfrak{L}_\Psi) \log\left(1 + \frac{4B_W^2 T}{d}\right).$$

$\square$

## G   EXTENSION: REWARD-FREE EXPLORATION

Reward-free exploration investigates is a problem where 1) the learning agent interacts with an environment without rewards, aiming to gather information so that 2) in a subsequent offline phase, the information collected can be used to learn near-optimal policies for a wide range of possible reward functions (Jin et al., 2020a; Zhang et al., 2020; Wang et al., 2020a; Zanette et al., 2020b; Chen et al., 2022). This section provides a reward-free extension of our main results, and gives sample complexity bounds based on coverability for a reward-free extension of GOLF.

**Function approximation.** We assume access to a value function class $\mathcal{F}$, which is used for the offline optimization, and a function class, $\mathcal{G}$, which is used for the reward-free exploration phase. Following the normalized reward assumption, we assume $g_h \in \mathcal{X} \times \mathcal{A} \to [0,1], \forall (g,h) \in \mathcal{G} \times [H]$.

We define $\mathcal{P}_h$ as be the "zero-reward" Bellman operator for horizon $h \in [H]$. That is, for any $g_h \in \mathcal{G}_h$ and any $h \in [H]$,

$$(\mathcal{P}_h g_{h+1})(x_h, a_h) := \sum_{x'} \mathbb{P}_h(x_{h+1} \mid x_h, a_h) \max_{a_{h+1} \in \mathcal{A}} g_{h+1}(x_{h+1}, a_{h+1}).$$

We let $R$ denote the *target reward function* used in the offline phase, which is not known to the algorithm in the offline exploration phase. We make the following assumption.

**Assumption 2** (Reward-free completeness). *Let $\mathcal{T}_{1:H}$ be the Bellman operator with the target reward function $R$, and $\mathcal{F}$ be the function class used to optimize the target reward function. Then for all $h \in [H]$*

*(a)* $\mathcal{P}_h \mathcal{G}_{h+1} \in \mathcal{G}_h$ *for all* $g_{h+1} \in \mathcal{G}_{h+1}$

*(b)* $\mathcal{F}_h - \mathcal{T}_h \mathcal{F}_{h+1} \subseteq \mathcal{G}_h - \mathcal{P}_h \mathcal{G}_{h+1}$.

Analogous to Assumption 1, Assumption 2(a) is used to control the squared Bellman error with zero reward. Assumption 2(b) guarantees that the class of test functions of interest for the reward-free exploration phase ($\mathcal{G}_h - \mathcal{P}_h \mathcal{G}_{h+1}$ for layer $h \in [H]$, see Algorithm 2) is sufficiently rich relative to the relevant class of test functions for the offline phase ($\mathcal{F}_h - \mathcal{T}_h \mathcal{F}_{h+1}$ for layer $h \in [H]$, see Algorithm 3). Without loss of generality, we assume that $|\mathcal{G}| = \max\{|\mathcal{F}|, |\mathcal{G}|\}$.

**Reward-free Sequential Extrapolation Coefficient.** The main guarantees for this section are stated in terms of a reward-free variant of the sequential extrapolation coefficient, which we define as follows.

**Definition 13** (Sequential Extrapolation Coefficient for Reward-Free RL). *For each $h \in [H]$, let $\mathfrak{D}_h^{\Pi_{\mathcal{G}}} := \{d_h^{\pi} : \pi \in \Pi_{\mathcal{G}}\}$ and $\mathcal{G}_h - \mathcal{P}_h \mathcal{G}_{h+1} := \{g_h - \mathcal{P}_h g_{h+1} : g \in \mathcal{G}\}$. Then we define,*

$$\mathsf{SEC}_{\mathsf{RL,rf}}(\mathcal{G}, \Pi_{\mathcal{G}}, T) := \max_{h \in [H]} \mathsf{SEC}(\mathcal{G}_h - \mathcal{P}_h \mathcal{G}_{h+1}, \mathfrak{D}_h^{\Pi_{\mathcal{G}}}, T).$$

Using the same arguments (and same proofs) as Section 5.2, the reward-free variant of sequential extrapolation coefficient can be shown to subsume coverability (as well as reward-free counterpart of the Bellman-Eluder dimension, which we omit).

## G.1 Algorithm and Theoretical Analysis

---

**Algorithm 2** Reward-Free Exploration with GOLF

---

**input:** Function class for reward-free exploration $\mathcal{G}$.

**initialize:** $\mathcal{D}_{h,\mathsf{rf}}^{(0)} \leftarrow \varnothing, \forall h \in [H]. \, \mathcal{G}^{(0)} \leftarrow \mathcal{G}$.

1: **for** episode $t = 1, 2, ..., T$ **do**
2:      Select policy $\pi^{(t)} \leftarrow \pi_{g^{(t)}}$, where $g^{(t)} = \mathrm{argmax}_{g \in \mathcal{G}^{(t-1)}} g(x_1, \pi_{g,1})$.
3:      Execute $\pi^{(t)}$ for one episode and obtain $\{x_1^{(t)}, a_1^{(t)}, x_2^{(t)}, ..., x_H^{(t)}, a_H^{(t)}, x_{H+1}^{(t)}\}$.
4:      Update historical data $\mathcal{D}_{h,\mathsf{rf}}^{(t)} \leftarrow \mathcal{D}_{h,\mathsf{rf}}^{(t-1)} \bigcup \{(x_h^{(t)}, a_h^{(t)}, x_{h+1}^{(t)})\}, \forall h \in [H]$.
5:      Compute confidence set:

$$\mathcal{G}^{(t)} \leftarrow \left\{ g \in \mathcal{G} : \mathcal{L}_{h,\mathsf{rf}}^{(t)}(g_h, g_{h+1}) - \min_{g_h' \in \mathcal{G}_h} \mathcal{L}_{h,\mathsf{rf}}^{(t)}(g_h', g_{h+1}) \leq \beta_{\mathsf{rf}}, \forall h \in [H] \right\}, \quad (29)$$

$$\text{where} \quad \mathcal{L}_{h,\mathsf{rf}}^{(t)}(g, g') := \sum_{(x,a,x') \in \mathcal{D}_{h,\mathsf{rf}}^{(t)}} \left[ \left( g(x,a) - \max_{a' \in \mathcal{A}} g'(x', a') \right)^2 \right], \forall g, g' \in \mathcal{G}.$$

6: Select $t_\star \leftarrow \mathrm{argmin}_{t \in [T]} g_1^{(t)}(x_1, \pi_1^{(t)})$.
7: Return data $\mathcal{D}_{h,\mathsf{rf}}^{(t_\star - 1)}, \forall h \in [H]$.

---

---

**Algorithm 3** Offline GOLF with Exploration Data and Target Reward

**input:**
- Target reward function, $R$.
- Function class $\mathcal{F}$ for offline RL.
- Exploration data from Algorithm 2, denoted by $\mathcal{D}_{h,\mathsf{rf}}$, $\forall h \in [H]$.
  1: Compute confidence set:

$$\mathcal{F}^{(\mathsf{off})} \leftarrow \left\{ f \in \mathcal{F} : \mathcal{L}_h^{(\mathsf{off})}(f_h, f_{h+1}) - \min_{f_h' \in \mathcal{F}_h} \mathcal{L}_h^{(\mathsf{off})}(f_h', f_{h+1}) \leq \beta_{\mathsf{off}}, \forall h \in [H] \right\}, \qquad (30)$$

$$\text{where} \quad \mathcal{L}_h^{(\mathsf{off})}(f, f') := \sum_{(x,a,x') \in \mathcal{D}_{h,\mathsf{rf}}} \left[ \left( f(x,a) - R(x,a) - \max_{a' \in \mathcal{A}} f'(x', a') \right)^2 \right], \forall f, f' \in \mathcal{F}.$$

  2: Return $\widehat{\pi} \leftarrow \pi_{\widehat{f}}$, where $\widehat{f} = \arg\max_{f \in \mathcal{F}^{(\mathsf{off})}} f(x_1, \pi_{f,1})$.

---

Recall that the key ideas in GOLF are: 1) using optimism to relate regret to on-policy average Bellman error; 2) using squared Bellman error to construct a confidence set, which ensures optimism. In the reward-free setting, one can apply these ideas by running GOLF (Algorithm 2) with rewards set to zero. Intuitively, this strategy ensures exploration because the algorithm must explore to rule out test functions in $\mathcal{G}$. However, a-priori it is unclear whether running some standard offline RL algorithms on the exploration data produced by this strategy should lead to a near-optimal policy, especially given that the PAC guarantee of GOLF relies on outputting a uniform mixture of all historical policies (see, e.g., Corollary 2).

To address such issues, one can imagine that, if we know which is the best over all historical policies (say, $\pi^{(t_\star)}$ for some $t_\star$), could running one-step GOLF on the exploration data at $t_\star$ (Algorithm 3) guarantee to find a good policy? Note that, for the original GOLF algorithm (in the known-reward case), running so directly reproduces $\pi^{(t_\star)}$. Although knowing which is the best over all historical policies seems impossible in the known-reward case, thanks to the reward-free nature, we will show that the value of $g(x_1, \pi_{g,1})$ directly captures "how bad is $g$" (akin to the regret in the known-reward case), which allow us to find the best step over the reward-free exploration phase.

The following result provides a sample complexity guarantee for this strategy.

**Theorem 23.** *Under Assumptions 1 and 2, there exists an absolute constants $c_1$ and $c_2$ such that for any $\delta \in (0,1]$ and $T \in \mathbb{N}_+$, if we choose $\beta_{\mathsf{off}} = c_1 \cdot \log(^{TH|\mathcal{G}|}/\delta)$ and $\beta_{\mathsf{rf}} = (c_1 + c_2) \cdot \log(^{TH|\mathcal{G}|}/\delta)$ in Algorithms 2 and 3, then with probability at least $1 - \delta$, the policy $\widehat{\pi}$ output by Algorithm 3 has*

$$J(\pi^\star) - J(\widehat{\pi}) \leq O\left( H \sqrt{\frac{\mathsf{SEC}_{\mathsf{RL,rf}}(\mathcal{G}, \Pi_\mathcal{G}, T) \log(^{TH|\mathcal{G}|}/\delta)}{T}} \right).$$

We defer the proof to Appendix G.2. We also introduce the following two lemmas, which are key to adapting the known-reward results to the reward-free case.

**Lemma 24** (Reward-free exploration overestimates regret)**.** *For any $f \in \mathcal{F}$, let $g$ be defined as $g_h = f_h - \mathcal{T}_h f_{h+1} + \mathcal{P}_h g_{h+1}$, $\forall h \in [H]$. Then for any $(x,a,h) \in \mathcal{X} \times \mathcal{A} \times [H]$, we have $g_h(x,a) \geq f_h(x,a) - Q_h^{\pi_f}(x,a)$.*

Since the Q-function for all policies in the zero-reward case are zero, Lemma 24 guarantees that, regret in the reward-free exploration phase—$(g_1(x_1, \pi_{g,1}) - 0)$ always upper bounds its counterpart of the offline phase—$(f_1(x_1, \pi_{f,1}) - Q_h^{\pi_f}(x, \pi_{f,1}))$. Equipped with the optimism argument, we can show that if $g_1(x_1, \pi_{g,1})$ is small, its corresponding $\pi_f$ (the $f$ with $f_h - \mathcal{T}_h f_{h+1} = g_h - \mathcal{P}_h g_{h+1}$, $\forall h \in [H]$) also has small regret.

**Lemma 25** (Reward-free exploration has larger confidence set)**.** *Suppose Assumption 2 holds and under the same conditions as Theorem 23. For any $f \in \mathcal{F}^{(\mathsf{off})}$ (defined in Eq. (30)), there must exist $g \in \mathcal{G}^{(t_\star - 1)}$ (defined in Eq. (29)), such that $f_h - \mathcal{T}_h f_{h+1} = g_h - \mathcal{P}_h g_{h+1}$, $\forall h \in [H]$.*

Lemma 25 ensures that the reward-free version space $\mathcal{G}^{(t_\star - 1)}$ subsumes the offline version space $\mathcal{F}^{(\mathsf{off})}$. Thus, we can use the metrics during reward-free exploration to upper bound that of the offline phase.

### G.1.1  RELATED WORK

Our approach adapts techniques for reward-free exploration in nonlinear RL introduced in Chen et al. (2022). In what follows, we discuss the connection to this work in greater detail. We focus on the $Q$-type results of Chen et al. (2022), but similar arguments are likely apply to the $V$-type.

Briefly, Chen et al. (2022) extends the OLIVE algorithm to the reward-free setting by using the idea of online exploration with zero rewards. The most important difference here is that, as discussed in Section 5, since OLIVE only considers average Bellman residuals, it cannot capture coverability. Beyond this difference, let us compare the completeness assumptions in Assumption 2 to those made in Chen et al. (2022). We will show that the completeness assumption used by Chen et al. (2022, Assumption 2) is a sufficient condition for ours (Assumptions 1 and 2). In our notation, Chen et al. (2022), use $\mathcal{F} \coloneqq \Psi + R \coloneqq \{\psi_{1:H}(\cdot,\cdot) + R_{1:H}(\cdot,\cdot) : \psi \in \Psi\}$ for some function class $\Psi$ during offline phase, and select $\mathcal{G} \coloneqq \Psi - \Psi \coloneqq \{\psi_{1:H}(\cdot,\cdot) - \psi'_{1:H}(\cdot,\cdot) : \psi, \psi' \in \Psi\}$ for the reward-free exploration phase. Thus for any $h \in [H]$, we have: For Assumption 1 and Assumption 2(a):

$$
\begin{aligned}
\mathcal{T}_h \mathcal{F}_{h+1} &= R_h + \mathcal{P}_h(\Psi_{h+1} + R_{h+1}) \\
&\subseteq R_h + \Psi_h = \mathcal{F}_h. &&\text{(by Chen et al. (2022, Assumption 2))} \\
\mathcal{P}_h \mathcal{G}_{h+1} &= \mathcal{P}_h(\Psi_{h+1} - \Psi_{h+1}) \\
&\subseteq \Psi_{h+1} - \Psi_{h+1} = \mathcal{G}_h. &&\text{(by Chen et al. (2022, Assumption 2))}
\end{aligned}
$$

For Assumption 2(b):

$$
\begin{aligned}
\mathcal{F}_h - \mathcal{T}_h \mathcal{F}_{h+1} &= \Psi_h + R_h - R_h - \mathcal{P}_h(\Psi_{h+1} + R_{h+1}) \\
&= \Psi_h - \mathcal{P}_h(\Psi_{h+1} + R_{h+1}) \\
&\subseteq \Psi_h - \Psi_h. &&\text{(by Chen et al. (2022, Assumption 2))} \\
\mathcal{G}_h - \mathcal{P}_h \mathcal{G}_{h+1} &= \Psi_h - \Psi_h - \mathcal{P}_h(\Psi_{h+1} - \Psi_{h+1}) \\
&\supseteq \Psi_h - \Psi_h. &&(0 \in \Psi_{h+1} - \Psi_{h+1}) \\
\implies \mathcal{F}_h - \mathcal{T}_h \mathcal{F}_{h+1} &\subseteq \mathcal{G}_h - \mathcal{P}_h \mathcal{G}_{h+1}.
\end{aligned}
$$

### G.2  PROOFS

We first present the following form of Freedman's inequality for martingales (e.g., Agarwal et al., 2014).

**Lemma 26** (Freedman's Inequality). *Let $\{X^{(1)}, X^{(2)}, ..., X^{(T)}\}$ be a real-valued martingale difference sequence adapted to a filtration $\{\mathscr{F}^{(1)}, \mathscr{F}^{(2)}, ..., \mathscr{F}^{(T)}\}$ (i.e., $\mathbb{E}[X^{(t)} \mid \mathscr{F}^{(t-1)}] = 0$, $\forall t \in [T]$). If $|X^{(t)}| \le R$ almost surely for all $t \in [T]$, then for any $\eta \in (0, 1/R)$, with probability at least $1 - \delta$,*

$$
\sum_{t=1}^{T} X^{(t)} \le \eta \sum_{t=1}^{T} \mathbb{E}\big[(X^{(t)})^2 \mid \mathscr{F}^{(t-1)}\big] + \frac{\log(1/\delta)}{\eta}.
$$

We now provide proofs from Appendix G.1.

**Proof of Theorem 23.** Over this section, the test function class is selected as

$$
\delta_{h,\mathsf{rf}}^{(t)}(x_h, a_h) \coloneqq g_h^{(t)}(x_h, a_h) - (\mathcal{P}_h g_{h+1}^{(t)})(x_h, a_h), \forall (h,t) \in [H] \times [T].
$$

By Theorem 5 (setting reward to be zero and replacing everything regarding $\mathcal{F}$ to $\mathcal{G}$), we have

$$
\sum_{t=1}^{T} \sum_{h=1}^{H} \mathbb{E}_{d_h^{(t)}}\left[\delta_{h,\mathsf{rf}}^{(t)}\right] \le H\sqrt{T\mathsf{SEC}_{\mathsf{RL,rf}}(\mathcal{G}, \Pi_{\mathcal{G}}, T)\beta_{\mathsf{rf}}}. \tag{31}
$$

For any $(h,t) \in [H] \times [T]$, we have,

$$
\begin{aligned}
\mathbb{E}_{d_h^{(t)}}\big[(\mathcal{P}_h g_{h+1}^{(t)})(x_h, a_h)\big] &= \mathbb{E}_{d_h^{(t)}}\left[\sum_{x'} \mathbb{P}_h(x_{h+1} \mid x_h, a_h) \max_{a_{h+1} \in \mathcal{A}} g_{h+1}^{(t)}(x_{h+1}, a_{h+1})\right] \\
&= \mathbb{E}_{d_h^{(t)}}\left[\sum_{x'} \mathbb{P}_h(x_{h+1} \mid x_h, a_h) g_{h+1}^{(t)}(x_{h+1}, \pi_h^{(t)})\right]
\end{aligned}
$$

($\pi^{(t)}$ is the greedy policy of $g^{(t)}$)

$$= \mathbb{E}_{d_{h+1}^{(t)}} \left[ g_{h+1}^{(t)}(x_{h+1}, a_{h+1}) \right]. \tag{32}$$

Therefore, we know

$$\sum_{h=1}^{H} \mathbb{E}_{d_h^{(t)}} \left[ \delta_{h,\mathsf{rf}}^{(t)} \right] = \sum_{h=1}^{H} \mathbb{E}_{d_h^{(t)}} \left[ g_h^{(t)} - \mathcal{P}_h g_h^{(t)} \right]$$

$$= \sum_{h=1}^{H} \left( \mathbb{E}_{d_h^{(t)}} \left[ g_h^{(t)} \right] - \mathbb{E}_{d_{h+1}^{(t)}} \left[ g_{h+1}^{(t)} \right] \right) \qquad \text{(by Eq. (32))}$$

$$= \mathbb{E}_{d_1^{(t)}} \left[ g_1^{(t)} \right]$$

$$= g_1^{(t)}(x_1, \pi_1^{(t)}). \tag{33}$$

Now, since

$$t_\star := \operatorname*{argmin}_{t \in [T]} g_1^{(t)}(x_1, \pi_1^{(t)}),$$

then,

$$g_1^{(t_\star)}(x_1, \pi^{(t_\star)}) = \frac{1}{T} \sum_{t=1}^{T} g_1^{(t)}(x_1, \pi_1^{(t)})$$

$$= \frac{1}{T} \sum_{t=1}^{T} \sum_{h=1}^{H} \mathbb{E}_{d_h^{(t)}} \left[ \delta_{h,\mathsf{rf}}^{(t)} \right] \qquad \text{(by Eq. (33))}$$

$$\leq H \sqrt{\frac{\mathsf{SEC}_{\mathsf{RL,rf}}(\mathcal{G}, \Pi_{\mathcal{G}}, T) \beta_{\mathsf{rf}}}{T}}, \tag{34}$$

where the last inequality follows from Eq. (31).

By Lemma 25, we know there exists a $\widehat{g} \in \mathcal{G}^{(t_\star - 1)}$, such that $\widehat{f}_h - \mathcal{T}_h \widehat{f}_{h+1} = \widehat{g}_h - \mathcal{P}_h \widehat{g}_{h+1}, \forall h \in [H]$. In addition, we can obtain

$$J(\pi^\star) - J(\pi_{\widehat{f}}) \leq \widehat{f}(x_1, \pi_{\widehat{f}, 1}) - J(\pi_{\widehat{f}}) \qquad \text{(by Lemma 12)}$$

$$\leq \widehat{g}(x_1, \pi_{\widehat{g}, 1}). \qquad \text{(by Lemma 24)}$$

Therefore, we have

$$J(\pi^\star) - J(\pi_{\widehat{f}}) \leq \widehat{g}(x_1, \pi_{\widehat{g}, 1})$$

$$\leq g_1^{(t_\star)}(x_1, \pi_1^{(t_\star)})$$

$$\leq H \sqrt{\frac{\mathsf{SEC}_{\mathsf{RL,rf}}(\mathcal{G}, \Pi_{\mathcal{G}}, T) \beta_{\mathsf{rf}}}{T}}. \qquad \text{(by Eq. (34))}$$

Plugging back the selection of $\beta_{\mathsf{rf}}$ completes the proof. $\qquad \square$

**Proof of Lemma 24.** We establish the proof by induction. For $h = H$, the the inductive hypothesis holds because $g_H = f_H - R_H = f_H - Q_H^{\pi_f}$.

Suppose the inductive hypothesis holds at $h+1$, we have for any $x \in \mathcal{X}$,

$$g_{h+1}(x, a) \geq f_{h+1}(x, a) - Q_{h+1}^{\pi_f}(x, a), \forall a \in \mathcal{A}.$$

$$\implies g_{h+1}(x, \pi_{f, h+1}) \geq f_{h+1}(x, \pi_{f, h+1}) - Q_{h+1}^{\pi_f}(x, \pi_{f, h+1}).$$

$$\implies \max_{a \in \mathcal{A}} g_{h+1}(x, a) \geq f_{h+1}(x, \pi_{f, h+1}) - Q_{h+1}^{\pi_f}(x, \pi_{f, h+1}).$$

$$\implies g_{h+1}(x, \pi_{g, h+1}) \geq f_{h+1}(x, \pi_{f, h+1}) - V_{h+1}^{\pi_f}(x). \tag{35}$$

Then, as $g_h = f_h - \mathcal{T}_h f_{h+1} + \mathcal{P}_h g_{h+1}$, we have for any $(x, a) \in \mathcal{X} \times \mathcal{A}$,

$$g_h(x, a) = f_h(x, a) - R_h(x, a) - \mathbb{E}_{x'|x, a} \left[ \max_{a' \in \mathcal{A}} f_{h+1}(x', a') \right] + \mathbb{E}_{x'|x, a} \left[ \max_{a' \in \mathcal{A}} g_{h+1}(x', a') \right]$$

$$
\begin{aligned}
&= f_h(x,a) - R_h(x,a) + \mathbb{E}_{x'|x,a}\left[\max_{a'\in\mathcal{A}} g_{h+1}(x',a') - \max_{a'\in\mathcal{A}} f_{h+1}(x',a')\right] \\
&= f_h(x,a) - R_h(x,a) + \mathbb{E}_{x'|x,a}[g_{h+1}(x',\pi_{g,h+1}) - f_{h+1}(x',\pi_{f,h+1})] \\
&\geq f_h(x,a) - R_h(x,a) + \mathbb{E}_{x'|x,a}\left[-V_{h+1}^{\pi_f}(x')\right] &&\text{(by Eq. (35))} \\
&= f_h(x,a) - \left(R_h(x,a) + \mathbb{E}_{x'|x,a}\left[V_{h+1}^{\pi_f}(x')\right]\right) \\
&= f_h(x,a) - Q_h^{\pi_f}(x,a).
\end{aligned}
$$

Therefore, we prove that the inductive hypothesis also holds at $h$ using the inductive hypothesis at $h+1$. This completes the proof. $\qquad\square$

**Proof of Lemma 25.** Over this proof, we use $d_h^{(t)}$ as the shorthand of $d_h^{\pi^{(t)}}$. The proof of this lemma consists of two parts.

(i) There exists a radius $\beta_1$, such that for any $g\in\mathcal{G}$, if such $g$ satisfies

$$
\sum_{t=1}^{t_\star-1}\mathbb{E}_{d_h^{(t)}}\left[(g_h - \mathcal{P}_h g_{h+1})^2\right]\leq\beta_1,\ \forall h\in[H]
$$

then $g\in\mathcal{G}^{(t_\star-1)}$.

(ii) There exists another radius $\beta_2$, where $\beta_2\leq\beta_1$. For any $f\in\mathcal{F}^{\mathrm{off}}$, we have

$$
\sum_{t=1}^{t_\star-1}\mathbb{E}_{d_h^{(t)}}\left[(f_h - \mathcal{T}_h f_{h+1})^2\right]\leq\beta_2,\ \forall h\in[H].
$$

**Proof of part (i).** For any $(t,h,g)\in[T]\times[H]\times\mathcal{G}$, let $Y_h^{(t)}(g)$ be defined as

$$
Y_h^{(t)}(g) := \left(g_h(x_h^{(t)},a_h^{(t)}) - g_{h+1}(x_{h+1}^{(t)},\pi_{g,1})\right)^2 - \left((\mathcal{P}_h g_{h+1})(x_h^{(t)},a_h^{(t)}) - g_{h+1}(x_{h+1}^{(t)},\pi_{g,1})\right)^2.
$$

Also, let $\mathscr{F}_h^{(t)}$ be the filtration induced by $\{x_1^{(i)},a_1^{(i)},x_2^{(i)},a_2^{(i)},...,x_H^{(i)}\}_{i=1}^t$, and we then have

$$
\mathbb{E}\left[Y_h^{(t)}(g)\,\big|\,\mathscr{F}_h^{(t-1)}\right] = \mathbb{E}_{d_h^{(t)}}\left[(g_h - \mathcal{P}_h g_{h+1})^2\right] \tag{36}
$$

and

$$
\mathbb{V}\left[Y_h^{(t)}(g)\,\big|\,\mathscr{F}_h^{(t-1)}\right]\leq\mathbb{E}\left[\left(Y_h^{(t)}(g)\right)^2\,\Big|\,\mathscr{F}_h^{(t-1)}\right]\leq 2\mathbb{E}\left[Y_h^{(t)}(g)\,\big|\,\mathscr{F}_h^{(t-1)}\right] = 2\mathbb{E}_{d_h^{(t)}}\left[(g_h - \mathcal{P}_h g_{h+1})^2\right].
$$

Now, let $\bar{Y}_h^{(t)}(g) := Y_h^{(t)}(g) - \mathbb{E}\left[Y_h^{(t)}(g)\,\big|\,\mathscr{F}_h^{(t-1)}\right]$, so that $\left\{\bar{Y}_h^{(t)}(g)\right\}_{t=1}^T$ is a martingale difference sequence adapts to the filtration $\left\{\mathscr{F}_h^{(t)}\right\}_{t=1}^T$, and $|\bar{Y}_h^{(t)}(g)|\leq 2$ almost surely. Then, by applying Lemma 26 with a union bound, we have for any $(h,g)\in[H]\times\mathcal{G}$ and any $\eta\in(0,1/2)$, with probability at least $1-\delta$,

$$
\begin{aligned}
\sum_{t=1}^{t_\star-1}\bar{Y}_h^{(t)}(g) &\leq \eta\sum_{t=1}^{t_\star-1}\mathbb{E}\left[\left(\bar{Y}_h^{(t)}(g)\right)^2\,\Big|\,\mathscr{F}_h^{(t-1)}\right] + \frac{\log(H|\mathcal{G}|/\delta)}{\eta} \\
&\leq \eta\sum_{t=1}^{t_\star-1}\mathbb{E}\left[\left(Y_h^{(t)}(g)\right)^2\,\Big|\,\mathscr{F}_h^{(t-1)}\right] + \frac{\log(H|\mathcal{G}|/\delta)}{\eta} \\
&\qquad\qquad\qquad\qquad\text{(variance is bounded by the second moment)} \\
&\leq \eta\sum_{t=1}^{t_\star-1}\mathbb{E}\left[Y_h^{(t)}(g)\,\big|\,\mathscr{F}_h^{(t-1)}\right] + \frac{\log(H|\mathcal{G}|/\delta)}{\eta}. \qquad (|Y_h^{(t)}(g)|\leq 1\text{ by its definition}) \\
\implies \sum_{t=1}^{t_\star-1}Y_h^{(t)}(g) &\leq \eta\sum_{t=1}^{t_\star-1}\mathbb{E}\left[Y_h^{(t)}(g)\,\big|\,\mathscr{F}_h^{(t-1)}\right] + \frac{\log(H|\mathcal{G}|/\delta)}{\eta} + \sum_{t=1}^{t_\star-1}\mathbb{E}\left[Y_h^{(t)}(g)\,\big|\,\mathscr{F}_h^{(t-1)}\right] \\
&\qquad\qquad\qquad\qquad\text{(by the definition of }\bar{Y}_h^{(t)}(g))
\end{aligned}
$$

$$= (1+\eta)\sum_{t=1}^{t_\star-1}\mathbb{E}\big[Y_h^{(t)}(g) \,\big|\, \mathscr{F}_h^{(t-1)}\big] + \frac{\log(H|\mathcal{G}|/\delta)}{\eta}. \tag{37}$$

If some $g \in \mathcal{G}$ satisfies

$$\underbrace{\sum_{t=1}^{t_\star-1}\mathbb{E}_{d_h^{(t)}}\Big[\big(g_h - \mathcal{P}_h g_{h+1}\big)^2\Big]}_{=\sum_{t=1}^{t_\star-1}\mathbb{E}\big[Y_h^{(t)}(g)\,\big|\,\mathscr{F}_h^{(t-1)}\big]\text{ by Eq. (36)}} \le \beta_1, \ \forall h \in [H],$$

then by Eq. (37), we have for any $h \in [H]$

$$\sum_{t=1}^{t_\star-1} Y_h^{(t)}(g) \le (1+\eta)\beta_1 + \frac{\log(H|\mathcal{G}|/\delta)}{\eta}$$

$$\le 3\big(\beta_1 + \log(H|\mathcal{G}|/\delta)\big). \qquad\qquad \text{(e.g., by picking } \eta = 1/3)$$

So we only need to guarantee

$$3 \cdot \big(\beta_1 + \log(H|\mathcal{G}|/\delta)\big) \le \beta_{\mathsf{rf}}$$

$$\implies \beta_1 \le \frac{\beta_{\mathsf{rf}}}{3} - \log(H|\mathcal{G}|/\delta). \tag{38}$$

**Proof of part (ii).** Similar to (i), for any $(t,h,f) \in [T] \times [H] \times \mathcal{F}$, let $X_h^{(t)}(f)$ be defined as

$$X_h^{(t)}(f) := \big(f_h(x_h^{(t)},a_h^{(t)}) - R(x_h^{(t)},a_h^{(t)}) - f_{h+1}(x_{h+1}^{(t)},\Pi_{\mathcal{G}})\big)^2$$
$$- \big((\mathcal{T}_h f_{h+1})(x_h^{(t)},a_h^{(t)}) - R(x_h^{(t)},a_h^{(t)}) - f_{h+1}(x_{h+1}^{(t)},\pi_{f,h+1})\big)^2.$$

Also let $\bar{X}_h^{(t)}(f) := \mathbb{E}\big[X_h^{(t)}(f)\,\big|\,\mathscr{F}_h^{(t-1)}\big] - X_h^{(t)}(f)$, so that $\big\{\bar{X}_h^{(t)}(f)\big\}_{t=1}^T$ is a martingale difference sequence adapts to the filtration $\big\{\mathscr{F}_h^{(t)}\big\}_{t=1}^T$, and $|\bar{X}_h^{(t)}(f)| \le 2$ almost surely.

Thus, by same arguments as Eqs. (36) and (37) (as well as applying Lemma 26), we have

$$\mathbb{E}\big[X_h^{(t)}(f)\,\big|\,\mathscr{F}_h^{(t)}\big] = \mathbb{E}_{d_h^{(t)}}\Big[(f_h - \mathcal{T}_h f_{h+1})^2\Big] \tag{39}$$

and for any $(h,f) \in [H] \times \mathcal{F}$ and any $\eta \in (0,1/2)$, with probability at least $1-\delta$,

$$\sum_{t=1}^{t_\star-1}\mathbb{E}\big[X_h^{(t)}(f)\,\big|\,\mathscr{F}_h^{(t-1)}\big] \le \eta\sum_{t=1}^{t_\star-1}\mathbb{E}\big[X_h^{(t)}(f)\,\big|\,\mathscr{F}_h^{(t-1)}\big] + \frac{\log(H|\mathcal{F}|/\delta)}{\eta} + \sum_{t=1}^{t_\star-1}X_h^{(t)}(f)$$

$$\implies (1-\eta)\sum_{t=1}^{t_\star-1}\mathbb{E}\big[X_h^{(t)}(f)\,\big|\,\mathscr{F}_h^{(t-1)}\big] \le \frac{\log(H|\mathcal{F}|/\delta)}{\eta} + \sum_{t=1}^{t_\star-1}X_h^{(t)}(f). \tag{40}$$

Therefore, if $f \in \mathcal{F}^{(\mathsf{off})}$, we have

$$\sum_{t=1}^{t_\star-1}X_h^{(t)}(f) = \sum_{t=1}^{t_\star-1}\big(f_h(x_h^{(t)},a_h^{(t)}) - f_{h+1}(x_{h+1}^{(t)},\pi_{f,h+1})\big)^2 - \sum_{t=1}^{t_\star-1}\big((\mathcal{T}_h f_{h+1})(x_h^{(t)},a_h^{(t)}) - f_{h+1}(x_{h+1}^{(t)},\pi_{f,h+1})\big)^2$$

$$\le \sum_{t=1}^{t_\star-1}\big(f_h(x_h^{(t)},a_h^{(t)}) - f_{h+1}(x_{h+1}^{(t)},\pi_{f,h+1})\big)^2 - \min_{f_h'\in\mathcal{F}_h}\sum_{t=1}^{t_\star-1}\big(f_h'(x_h^{(t)},a_h^{(t)}) - f_{h+1}(x_{h+1}^{(t)},\pi_{f,h+1})\big)^2$$

$$\le \mathcal{L}_h^{(\mathsf{off})}(f_h, f_{h+1}^{(t_\star)}) - \min_{f_h'\in\mathcal{F}_h}\mathcal{L}_h^{(\mathsf{off})}(f_h', f_{h+1}^{(t_\star)})$$

$$\le \beta_{\mathsf{off}}. \tag{41}$$

We then combine Eqs. (39) to (41) and obtain

$$\sum_{t=1}^{t_\star-1}\mathbb{E}_{d_h^{(t)}}\Big[(f_h - \mathcal{T}_h f_{h+1})^2\Big] = \sum_{t=1}^{t_\star-1}\mathbb{E}\big[X_h^{(t)}(f)\,\big|\,\mathscr{F}_h^{(t-1)}\big] \qquad\qquad \text{(by Eq. (39))}$$

$$\leq \frac{\log(H|\mathcal{F}|/\delta)}{(1-\eta)\eta} + \frac{1}{1-\eta} \sum_{t=1}^{t_\star-1} X_h^{(t)}(f) \qquad \text{(by Eq. (40))}$$

$$\leq \frac{\log(H|\mathcal{F}|/\delta)}{(1-\eta)\eta} + \frac{1}{1-\eta} \beta_{\mathsf{off}} \qquad \text{(by Eq. (41))}$$

$$\leq \underbrace{5\log(H|\mathcal{F}|/\delta) + 2\beta_{\mathsf{off}}}_{=:\beta_2}. \qquad \text{(by e.g., setting } \eta = 1/3)$$

So we only need to guarantee

$$5\log(H|\mathcal{F}|/\delta) + 2\beta_{\mathsf{off}} = \beta_2 \leq \beta_1. \tag{42}$$

**Putting everything together.** By Eqs. (38) and (42), we know we only need the following inequality to hold:

$$5\log(H|\mathcal{F}|/\delta) + 2\beta_{\mathsf{off}} \leq \frac{\beta_{\mathsf{rf}}}{3} - \log(H|\mathcal{G}|/\delta).$$
$$\implies \beta_{\mathsf{rf}} \geq 6\beta_{\mathsf{off}} + 18\log(H|\mathcal{G}|/\delta).$$

This is satisfied via the condition of Theorem 23.

Combining (i) and (ii), we can simply obtain for any $h \in [H]$,

$$\left\{ f_h - \mathcal{T}_h f_{h+1} : f \in \mathcal{F}^{\mathsf{off}} \right\} \subseteq \left\{ f_h - \mathcal{T}_h f_{h+1} : \sum_{t=1}^{t_\star-1} \mathbb{E}_{d_h^{(t)}} \left[ (f_h - \mathcal{T}_h f_{h+1})^2 \right] \leq \beta_2, \forall h \in [H], f \in \mathcal{F} \right\}$$
$$\text{(by (ii))}$$

$$\subseteq \left\{ g_h - \mathcal{P}_h f_{h+1} : \sum_{t=1}^{t_\star-1} \mathbb{E}_{d_h^{(t)}} \left[ (g_h - \mathcal{P}_h f_{h+1})^2 \right] \leq \beta_1, \forall h \in [H], g \in \mathcal{G} \right\}$$
$$\text{(by Assumption 2 and } \beta_2 \leq \beta_1)$$

$$\subseteq \left\{ g_h - \mathcal{P}_h g_{h+1} : g \in \mathcal{G}^{(t_\star-1)} \right\}. \qquad \text{(by (i))}$$

This completes the proof. $\qquad\square$

