# OpenReview forum: "The Role of Coverage in Online Reinforcement Learning"
_ICLR.cc/2023/Conference — ICLR 2023 notable top 5%_

### Official Review · Reviewer_PRRf · 2022-10-24

**Confidence:** 4
**Correctness:** 4
**Technical Novelty And Significance:** 3
**Empirical Novelty And Significance:** 3
**Recommendation:** 8

**Clarity, Quality, Novelty And Reproducibility:**

The paper is clearly written. Please refer to weakness section for specific questions.

**Strength And Weaknesses:**

*Strengths*
- The paper is well written.
- The authors have addressed a crucial challenge of sample efficiency in reinforcement learning and the paper is quite relevant to the community.
- The authors have introduced the concept of coverability with taking motivation from the idea of concentrability from offline RL which is quite interesting.

*Questions to Authors*

- It would have been nice to show the effect of coverability coefficient via empirical studies.

- Is it possible to have coverability coefficient =0 ?

- How difficult is it to construct the confidence sets in practice? Is the approach feasible?

- Is it possible to extend this framework to continuous state-action spaces?


**Summary Of The Paper:**

The authors have introduced the concept of coverability, and related it to the sample efficiency in the online reinforcement learning. This idea is very interesting and the paper is well written with detailed description of everything.

**Summary Of The Review:**

N/A

---

> ### Author Response · Authors · 2022-11-13
> **Authors response**
>
> We thank the reviewer for appreciating the contributions of our work, and for your positive comments. We answer your questions below:
>
> ------
>
> > It would have been nice to show the effect of coverability coefficient via empirical studies.
>
> While we agree that it would be interesting to explore these concepts empirically, actually doing so presents difficulties that are not just in this work, but in the line of research on online RL more broadly: there have been a number of important complexity measures for online RL with function approximation (e.g., Bellman rank and Bellman-eluder dimension discussed in Section 5) proposed in recent years, but no experimental works have been done to study their behavior empirically; this is largely due to computationally issues involved with implementing these algorithms (e.g., Dann et al.'18). We agree that this is an important direction for future research.
>
> ------
>
> > Is it possible to have coverability coefficient = 0 ?
>
> By definition, the lowest possible value for the coefficient is 1. This can be seen via *cumulative reachability*, an equivalent definition of coverability (see Lemma 14 on p18), defined as $\max_{h }\sum_{x,a} \sup_{\pi \in \Pi} d_h^{\pi}(x,a)$. It is easy to observe that if there is only one policy in the policy class $\Pi$, cumulative reachability (which is equal to coverability) is 1, and having more policies in the policy class will only make the cumulative reachability larger.
>
> Also, we suspect that by “coverability coefficient = 0”, the reviewer might mean “how bad can the coverability get”? Apologies if we misinterpreted your question, but just in case, let us emphasize that for the coverability coefficient, **the lower the better**, as our regret bounds grow larger when the coverability coefficient grows large. As we discuss below Definition 2, the coverability coefficient is always upper bounded by the number of state-action pairs. Another trivial upper bound (which we can include in the revision) is $|\Pi|$, i.e., the number of candidate policies, which can be achieved by setting $\mu_h$ as the uniform mixture of $d_h^\pi$ for all $\pi$ in the policy class. However, both bounds are very loose (we want to avoid dependence on the number of states and actions, and only want to pay logarithmic dependence on the size of policy class). This is consistent with the well-known fact that structural assumptions (beyond, e.g., access to a small policy class or value function class) are necessary to make online RL with function approximation tractable. In our case, assuming that the coverability coefficient is small serves as such a structural assumption.
>
> ------
>
> > How difficult is it to construct the confidence sets in practice? Is the approach feasible?
>
> Often, confidence-set-based algorithms can be viewed as running a constrained optimization, and can be turned into a regularized optimization problem for practical implementation; see the paper “Bellman-consistent pessimism for offline reinforcement learning” for an example of how this can be done in the context of offline RL. That said, for online RL with general function approximation, there are currently no computationally efficient algorithms that use “global” confidence sets of the type we consider. For example, the Bellman-eluder dimension paper (Jin et al.' 21), which proposed the GOLF algorithm we incorporate, mentioned that “the algorithms proposed in [a number of confidence-set-based works] and this paper are all computationally inefficient in general”. See also Dann et al.'18 for discussion of computational barriers. We emphasize that the main contribution of our work, which is to highlight the novel connection between the learnability conditions of online and offline RL, is *statistical* in nature rather than computational. Understanding when it is possible to develop a computationally tractable implementation of confidence-set-based algorithms for general function approximation is a long-standing issue for online RL, and is out of the scope of our work, though we hope to investigate in the future.
>
> ------
>
> > Is it possible to extend this framework to continuous state-action spaces?
>
> Note that our guarantees incur **no explicit dependence on the size of the state-action space**, so there is no technical obstacle to handling continuous state-action spaces other than using appropriate measure-theoretic notation. We avoid such a treatment only because it would make the paper more lengthy and less readable to a general audience. Many contemporary works in online RL with function approximation (including those cited and discussed in our paper) restrict to discrete state-action spaces for exactly the same reason: the algorithms and the insights are effectively applicable to the continuous case, but the measure-theoretic setup required to properly handle the continuous spaces does not really add to the key insights and messages of these works.

---

### Official Review · Reviewer_p7d7 · 2022-10-25

**Confidence:** 3
**Correctness:** 3
**Technical Novelty And Significance:** 2
**Empirical Novelty And Significance:** Not applicable
**Recommendation:** 5

**Clarity, Quality, Novelty And Reproducibility:**

The paper is well written and the problem the paper considers is important and novel.

**Strength And Weaknesses:**

Strengths:
- The paper addresses an important setting of RL: online RL with access to ofﬂine data
- The paper is generally well written

Weaknesses:
- However, I have some concerns about the novelty of the main contribution of the paper.  A very recent result under review of ICLR 2023 [1] shows that under the Concentrability assumption as Definition 1 (similar to the coverability assumption as the paper proposed), it is enough to achieve an "optimal" offline RL, without any interaction with the environment. These results seem correct to me. Then,
 - If we can access offline data and assume that offline data has good coverage as stated in Definition 2 (lower bound of Concentrability assumption), so do we really need the online RL?
 - If we really still need the online RL, then it seems that the coverability condition on offline data that the paper proposed is not enough weak to keep a sample-efficient online RL.

[1] https://openreview.net/forum?id=ZsvWb6mJnMv&referrer=%5BReviewer%20Console%5D(%2Fgroup%3Fid%3DICLR.cc%2F2023%2FConference%2FReviewers%23assigned-papers)

**Summary Of The Paper:**

This paper considers a setting of RL where the learner has access to ofﬂine data, yet also has the ability to online interact with the environment. The main contribution of the paper is to show that under a data distribution with good coverage, we can enable sample-efficient online RL. The paper introduces a new notation, called coverability which is weaker than the concentrability coefficient in offline RL and shows that it is enough for sample-efficient online RL. In addition, the paper also shows (1) several weaker notions of coverage which are sufficient for ofﬂine RL, but insufficient for online RL; (2) existing complexity measures for online RL such as Bellman rank and Bellman-Eluder dimension are insufficient for sample-efficient online RL.



**Summary Of The Review:**

Due to my concerns about the main contributions of this paper, I am toward rejecting it.  If the authors can address my concern, I am willing to increase my score.

---

> ### Author Response · Authors · 2022-11-13
> **Clarification of the main point of the paper—we do not assume access to offline data or an offline data distribution**
>
> Unfortunately, the reviewer seems to have misunderstood the setup of our paper: The review suggests that our learning algorithm performs online RL “[with] access to offline data” and therefore we should compare our sample complexity guarantees to existing offline RL algorithms and analyses. This is not the case. The key feature of our results is that we assume **the existence** of an “offline distribution” with good concentrability, but such a distribution is **not available to the learner** in the setting we consider. The learner knows nothing about this distribution, and in particular does not have access to any data sampled from the distribution. The point of our work is to show that **existence** of this distribution, even though it is not known to the learner, can be viewed as a structural assumption which imposes regularity on the MDP dynamics (and suffices for sample-efficient online RL). We were concerned that this might a common point of confusion about our work, so we went to great lengths to highlight this point as much as possible in the submission; for example:
> 1. abstract: “... even when the agent does not know said distribution”
> 2. Algorithm 1: does not mention or refer to offline distribution or data
> 3. Section 1.1: “...coverability (that is, mere existence of a distribution with good concentrability) is sufficient for sample-efficient online exploration, even when the learner has no prior knowledge of this distribution.”
> 4. Page 4:
> “Note that if the learner were given access to data from the distribution $\mu$ that achieves the value of $C_\mathsf{cov}$, it would be possible to simply appeal to offline RL methods such as FQI, but since the learner has no prior knowledge of $\mu$, this question is non-trivial, and requires deliberate exploration.”
> 5. Page 5:
> “While coverability implies that there exists a distribution $\mu$ for which the concentrability coefficient $C_\mathsf{conc}$ is bounded, Algorithm 1 has no prior knowledge of this distribution.”
>
> While our work provides new conceptual bridges between offline and online RL, the fact that the learner doesn't have access to the offline distribution (or data from the distribution) makes existing algorithms and analyses for offline RL inapplicable, as we consider a purely online setup.
>
> We hope this addresses your concern. If there are any further questions we can clarify, please feel free to let us know.

---

### Official Review · Reviewer_LEoz · 2022-11-01

**Confidence:** 3
**Correctness:** 4
**Technical Novelty And Significance:** 3
**Empirical Novelty And Significance:** Not applicable
**Recommendation:** 8

**Clarity, Quality, Novelty And Reproducibility:**

The overall writing is clear, easy to follow. I didn't read the entire appendix in detail and only checked the proofs of some main results, which is easy to understand and seems to be correct.

A very small number of typos appear in the text, e.g., "ands" in page 1, "offline and offline", "can achieved" in page 2. There might be more in the later sections but overall the impact is minimal and doesn't affect the understanding of the paper.

**Strength And Weaknesses:**

Strength:

1. The topic is important and the idea is novel. Both offline RL and online RL are important topics for the community and the paper serves as a bridge of the two formulations. In particular, the idea that the notion in offline RL can facilitate online exploration is novel and may serve as an inspiring point for future researches.

2. The story is complete. The authors compare their notion with existing notations for both online and offline RL, which addresses the necessity and novelty of their findings. The example of block MDP is also a good point, which shows how the authors' findings can actually result in a sharper sample complexity, compared with previous works.

Weakness:

1. Part of the technical contribution is limited, since the algorithm and some of the analysis are directly copied from existing works. However, it shouldn't be considered as a big problem since the paper itself is mainly focused on the analysis instead of the algorithm design, and the paper does show its own novelty in the regret analysis.


**Summary Of The Paper:**

The paper introduces a new notion of coverability, and shows its sufficiency for efficient online reinforcement learning. This new notion is compared with existing concepts from both offline and online RL. For offline RL, the authors show other weaker notions of coverage are not sufficient for efficient exploration. For online RL, they show their notion are independent from existing complexity measures, including Bellman rank and Bellman-Eluder dimension, and provide a new complexity measure that unifies these notions.

**Summary Of The Review:**

The work is novel and provides impressive results, based on simple but inspiring observations. I would suggest it be accepted.

---

> ### Author Response · Authors · 2022-11-13
> **Authors response**
>
> We thank the reviewer for your positive comments! We will be sure to incorporate your suggestions.
>
> The reviewer mentioned that  “part of the technical contributions are limited” (because the algorithm and some lemmas used in the analysis are standard), but also pointed out that “the paper does show its own novelty in the regret analysis”. We believe that our paper should be evaluated based on the conceptual and technical novelty it brings to the table. While some of our results build on existing techniques, we combine them with many new technical and conceptual insights to tell a story that we believe is novel, and is of broad interest. It seems as though the reviewer agrees with this point given the favorable score, but we wanted to clarify this to make sure that other reviewers and the AC are on the same page.

---

### Decision · Program_Chairs · 2023-01-20

**Decision:**

Accept: notable-top-5%

**Justification For Why Not Higher Score:**

N/A.

**Justification For Why Not Lower Score:**

I believe this paper and the ideas it presents are of broad interest to the RL community at large.

**Metareview: Summary, Strengths And Weaknesses:**

This paper established a new complexity measure for online RL with nonlinear function approximations based on the coverage conditions commonly studied in offline RL. Conceptually, this paper draws an interesting connection between online and offline RL. It also shows that the coverage condition is not captured by existing complexity notions such as the Bellman-Eluder dimension and therefore contributes new territories of provably solvable RL instances. I vote for acceptance of this paper.

**Note From Pc:**

if the above contains the word "oral" or "spotlight" please see: "oral" presentation means -> notable-top-5% and "spotlight" means -> notable-top-25%. As stated in our emails, we are disassociating presentation type from AC recommendations